# WAVE: Weighted Autoregressive Varying Gate for Time Series Forecasting

**Jiecheng Lu** [1]   **Xu Han** [2]   **Yan Sun** [1]   **Shihao Yang** [1]

## Abstract

We propose a Weighted Autoregressive Varying gatE (WAVE) attention mechanism equipped with both Autoregressive (AR) and Moving-average (MA) components. It can adapt to various attention mechanisms, enhancing and decoupling their ability to capture long-range and local temporal patterns in time series data. In this paper, we first demonstrate that, for the time series forecasting (TSF) task, the previously overlooked decoder-only autoregressive Transformer model can achieve results comparable to the best baselines when appropriate tokenization and training methods are applied. Moreover, inspired by the ARMA model from statistics and recent advances in linear attention, we introduce the full ARMA structure into existing autoregressive attention mechanisms. By using an indirect MA weight generation method, we incorporate the MA term while maintaining the time complexity and parameter size of the underlying efficient attention models. We further explore how indirect parameter generation can produce implicit MA weights that align with the modeling requirements for local temporal impacts. Experimental results show that WAVE attention that incorporates the ARMA structure consistently improves the performance of various AR attentions on TSF tasks, achieving state-of-the-art results. The code implementation is available at the following link.

## 1. Introduction

In recent years, autoregressive (AR) decoder-only Transformer-based models (Vaswani, 2017; Radford, 2018) have been widely used in sequence modeling tasks across fields such as NLP (Brown et al., 2020; Touvron et al., 2023), CV (Chen et al., 2020; Esser et al., 2021; Chang et al., 2022;

Liu et al., 2024a), and audio (Borsos et al., 2023). This structure is well-suited for various sequential generation and prediction tasks. However, in typical sequence modeling tasks like time series forecasting (TSF), there has been less exploration of this architecture compared to other structures. Most of the best-performing recent TSF models are encoder-only Transformers (Liu et al., 2024b; Nie et al., 2022), MLPs (Das et al., 2023; Lu et al., 2024), or even linear models (Zeng et al., 2023; Xu et al., 2024). The few relevant discussions mainly focus on using pretrained autoregressive LLMs or similar structures for few-shot and zero-shot prediction (Gruver et al., 2023; Jin et al., 2024; Das et al., 2024; Liu et al., 2024c), with little research directly evaluating their TSF performance in end-to-end training. Therefore, this paper will first briefly demonstrate that with appropriate tokenization and training methods, a basic AR Transformer is enough to achieve results comparable to the state-of-the-art (SOTA) baselines, as shown in Fig. 1.

Recently, efficient linear autoregressive attention variants have been explored and developed (Katharopoulos et al., 2020; Hua et al., 2022), reducing the time complexity of standard softmax attention from $O(N^2)$ to $O(N)$. Researchers have found that adding a gating decay factor or a similar exponential moving average (EMA) structure to AR structure, as in gated linear attention (Ma et al., 2022; Yang et al., 2024), enhances linear attention's ability to model local patterns and improves performance. The success of these approaches inspired us to introduce a more comprehensive full autoregressive moving-average (ARMA) structure into existing AR attention mechanisms and explore the performance of these Transformers in TSF.

In TSF models, EMA, connecting back to the historic work of Holt-Winters (Winters, 1960; Holt, 2004), focuses on smoothed local data, which improves the modeling of short-term fluctuations but reduces the ability to capture long-term information. In contrast, ARMA, connecting back to the historic work of Box-Jenkins (Box et al., 1974), a classic structure in TSF, considers both historical data and the cumulative impact of prediction errors. This allows it to handle and decouple long-term and short-term effects, significantly improving forecasting performance on data with complicated temporal patterns.

We propose the Weighted Autoregressive Varying gatE

---

[1]Georgia Institute of Technology [2]AWS. Correspondence to: Jiecheng Lu <jlu414@gatech.edu>, Shihao Yang <shihao.yang@isye.gatech.edu>.

*Proceedings of the 42nd International Conference on Machine Learning*, Vancouver, Canada. PMLR 267, 2025. Copyright 2025 by the author(s).

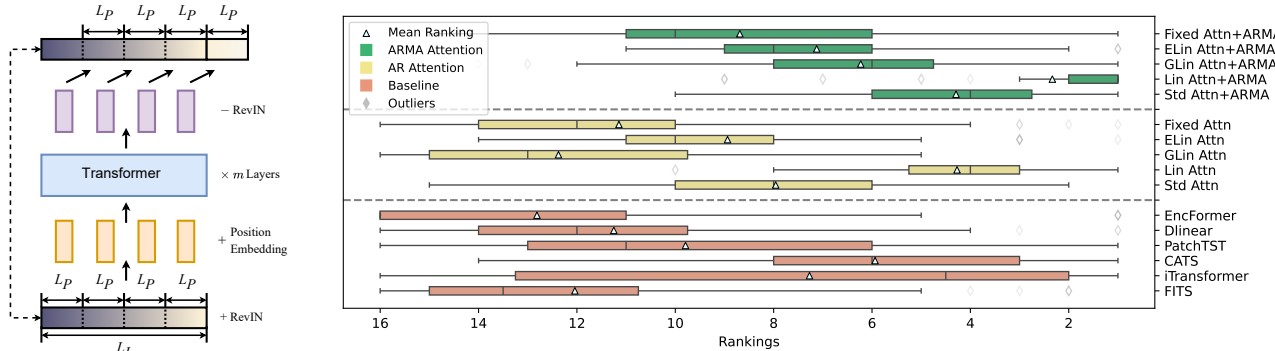

*Figure 1.* (Left: a) Overall architecture of our decoder Transformer for TSF. (Right: b) Box plots of performance rankings from 48 sub-experiments across 12 datasets. Green represents WAVE Transformers, yellow AR Transformers, and red the baselines, with triangles indicating mean rankings. AR Transformers perform comparably to baselines, while WAVE Transformers significantly outperform their AR counterparts. See Table and for more details.

(WAVE) attention mechanism equipped with ARMA structure, which integrates a moving-average (MA) term into various existing AR attention mechanisms. Our method improves the TSF performance of AR Transformers without significantly increasing computational costs, maintaining $O(N)$ time complexity and original parameter size. We design an indirect MA weight generation to obtain the MA output without explicitly computing the MA attention matrix, preserving efficiency of linear attentions. We explore specific techniques for generating implicit MA weights to ensure proper decoupling and handling of short-term effects. Extensive experiments and visualization analyses demonstrate that ARMA balances long- and short-term dependencies, significantly improving AR Transformers and achieving state-of-the-art TSF results.

The main contributions of this paper can be summarized as follows:

a) We demonstrate that, with appropriate tokenization and preprocessing methods, an AR Transformer is enough to achieve the level of existing SOTA baselines. Furthermore, the introduction of WAVE attention enables the decoder-only Transformer to outperform SOTA baselines.

b) We propose the WAVE attention mechanism, which introduces an MA term into existing AR attention without increasing time complexity or parameter size. By adding the MA term to various AR attention mechanisms, the resulting WAVE Transformers significantly improve forecasting performance compared to their AR counterparts.

c) We design an indirect MA weight generation method that is computationally efficient while ensuring that the implicit MA weights effectively capture the important short-term effects in TSF, allowing the AR term to focus more on long-term and cyclic patterns.

## 2. Method

### 2.1. Time series forecasting

In Time Series Forecasting (TSF), the goal is to predict the future part in a multivariate time series $\mathbf{S} \in \mathbb{R}^{L \times C}$, where $L$ is the length of the series, and $C$ is the number of channels or input series. The time series is divided into historical input $\mathbf{S}_I \in \mathbb{R}^{L_I \times C}$, and future data $\mathbf{S}_P \in \mathbb{R}^{L_P \times C}$, where $L = L_I + L_P$, and $L_I$ and $L_P$ represent the lengths of the input and forecasting periods, respectively. The objective is to learn a mapping function $f : \mathbb{R}^{L_I \times C} \to \mathbb{R}^{L_P \times C}$ that predicts the future values $\widehat{\mathbf{S}}_P = f(\mathbf{S}_I)$, given the historical input $\mathbf{S}_I$.

### 2.2. Appropriate tokenization for autoregressive forecasting

Recently, most time series forecasting research utilizes encoder-decoder or encoder-only Transformers for TSF (Li et al., 2019b; Zhou et al., 2021; Wu et al., 2021; Nie et al., 2022; Liu et al., 2024b), with limited focus on end-to-end decoder-only autoregressive Transformer because of error accumulation issue. For long-term forecasts, The autoregressive Transformers requires iteratively doing one-step prediction, leading to error accumulation and higher MSE compared to non-autoregressive models that generate the entire forecast at once.

To prevent error accumulation, we use an autoregressive Transformer (Fig. 1) that treats one-step prediction as the complete forecast. Inspired by PatchTST (Nie et al., 2022), we adopt a channel-independent approach, predicting each series separately and applying RevIN (Kim et al., 2022) to each. For an input series of length $L_I$ in $\mathbf{S}_I$, we apply non-overlapping patches with a patch size $L_P$, dividing the input into $N = \frac{L_I + P}{L_P}$ patches, where $P$ is zero-padding for divisibility. This ensures that each out-of-sample prediction token covers the entire forecasting length $L_P$, thereby

avoiding error accumulation[1].

Fig. 1 shows that autoregressive Transformers using this method can achieve performance comparable to existing SOTA models. Additionally, decoder-based architectures may have significant advantages in extended lookback length and varying output horizon, highlighting their potential.

## 2.3. Preliminaries: decoder-only Transformer

We use a GPT-2–style decoder-only Transformer (Radford et al., 2019) for autoregressive TSF. Token patches of length $L_P$ are linearly projected to $d$-dimensional vectors and combined with learnable positional embeddings to form the input sequence $\mathbf{X} \in \mathbb{R}^{N \times d}$, where each token is $\boldsymbol{x}_t \in \mathbb{R}^{1 \times d}$. Each of the $m$ Transformer layers applies layer normalization $\text{LN}(\cdot)$, attention $\text{Attn}(\cdot)$, and a channel-wise MLP $\text{MLP}(\cdot)$. With single-head softmax attention, a Transformer layer is defined as:

$$\text{Attn}(\mathbf{X}) = \text{softmax}\left(\mathbf{M} \odot (\mathbf{Q}\mathbf{K}^\top)\right) \mathbf{V}\mathbf{W}_o,$$
$$\text{with } \mathbf{Q}, \mathbf{K}, \mathbf{V} = \mathbf{X}\mathbf{W}_q, \mathbf{X}\mathbf{W}_k, \mathbf{X}\mathbf{W}_v \quad (1)$$
$$\mathbf{X} := \mathbf{X} + \text{Attn}(\text{LN}(\mathbf{X})), \text{ then } \mathbf{X} := \mathbf{X} + \text{MLP}(\text{LN}(\mathbf{X}))$$

where $\mathbf{W}_q, \mathbf{W}_k, \mathbf{W}_v, \mathbf{W}_o \in \mathbb{R}^{d \times d}$ are the projection matrices for the query, key, value, and output, respectively, and $\mathbf{M} \in \mathbb{R}^{N \times N}$ is the causal mask, defined as $\mathbf{M}_{ij} = 1\{i \geq j\} - \infty \cdot 1\{i < j\}$.

## 2.4. Preliminaries: efficient linear attention mechanisms

Recent autoregressive efficient attention mechanisms reduce computational complexity from $O(N^2)$ to $O(N)$ by avoiding the explicit calculation of the $N \times N$ attention matrix (Katharopoulos et al., 2020; Choromanski et al., 2021; Hua et al., 2022; Sun et al., 2023). Most of them can be reformulated as parallel linear RNNs with identity or diagonal state updates. Although these efficient attentions do not outperform standard softmax attention for large models, they achieve comparable results on smaller tasks (Katharopoulos et al., 2020; Choromanski et al., 2021). This paper investigates integrating these mechanisms into TSF and shows that adding a moving-average term significantly improves their performance. We begin by expressing the recurrent form of standard softmax attention. For a single head without output projection, let $\boldsymbol{q}_t, \boldsymbol{k}_t, \boldsymbol{v}_t$ be the vectors at step $t$ from $\mathbf{Q}, \mathbf{K}, \mathbf{V}$. The output $\boldsymbol{o}_t$ is given by: $\boldsymbol{o}_t = \frac{\sum_{i=1}^{t} \exp(\boldsymbol{q}_t \boldsymbol{k}_i^\top) \boldsymbol{v}_i}{\sum_{i=1}^{t} \exp(\boldsymbol{q}_t \boldsymbol{k}_i^\top)}$.

**Linear attention**  Linear Attention replaces the $\exp(\boldsymbol{q}_t \boldsymbol{k}_i^\top)$

---

[1] Please note that, consistent with previous studies on long-term time series forecasting, this paper focuses on one-step prediction for the next long time period. Specifically, we predict the next token of length $L_P$. Readers can view AR tokenization as PatchTST-style tokenization with an added autoregressive loss.

term in standard attention with a kernel function $k(\boldsymbol{x}, \boldsymbol{y}) = \langle \phi(\boldsymbol{x}), \phi(\boldsymbol{y}) \rangle$, resulting in $\phi(\boldsymbol{q}_t)\phi(\boldsymbol{k}_i)$ (Katharopoulos et al., 2020). This change reduces the time complexity from $O(N^2)$ to $O(N)$ by eliminating the need to compute the full $N \times N$ attention matrix. Instead, it computes $\phi(\boldsymbol{k}_i)^\top \boldsymbol{v}_i$ for each $i$ and aggregates over $N$. Various kernel functions have been explored, with identity kernels without denominators performing well enough (Mao, 2022; Qin et al., 2022; Sun et al., 2023; Yang et al., 2024). In this setup, Linear attention can be viewed as an RNN with a hidden state matrix $\boldsymbol{k}_i^\top \boldsymbol{v}_i \in \mathbb{R}^{d \times d}$ that updates using the identity function. The output at each step is: $\boldsymbol{o}_t = \boldsymbol{q}_t \sum_{i=1}^{t} \boldsymbol{k}_i^\top \boldsymbol{v}_i$.

**Element-wise linear attention**  In multi-head linear attention with $h$ heads, we handle $h$ hidden state matrices of size $\frac{d}{h} \times \frac{d}{h}$. When $h = d$, this simplifies to $h$ scalar hidden states, effectively transforming linear attention into a linear RNN with a $d$-dimensional hidden state vector $\phi(\boldsymbol{k}_i) \odot \boldsymbol{v}_i$ and enabling element-wise computations of $\boldsymbol{q}, \boldsymbol{k}, \boldsymbol{v}$. This approach, also known as the Attention Free Transformer (AFT) (Zhai et al., 2021), is favored for its simplicity and efficiency in recent works (Peng et al., 2023). We adopt the structure in AFT, where $\sigma(\cdot)$ is the sigmoid function, and the output at each step is: $\boldsymbol{o}_t = \sigma(\boldsymbol{q}_t) \odot \frac{\sum_{i=1}^{t} \exp(\boldsymbol{k}_i) \odot \boldsymbol{v}_i}{\sum_{i=1}^{t} \exp(\boldsymbol{k}_i)}$.

**Gated linear attention**  Recent studies have explored adding a forget gate, commonly used in traditional RNNs, to linear attention, allowing autoregressive models to forget past information and focus on local patterns (Mao, 2022; Sun et al., 2023; Qin et al., 2024; Yang et al., 2024). We implement a simple gating mechanism where each input $\boldsymbol{x}_t$ is converted into a scalar between [0, 1] and expanded into a forget matrix $\mathbf{G}_i$ matching the shape of $\boldsymbol{k}_i \boldsymbol{v}_i$. With gating parameters $\mathbf{W}_g \in \mathbb{R}^{d \times 1}$, the output at each step is: $\boldsymbol{o}_t = \boldsymbol{q}_t \sum_{i=1}^{t} \mathbf{G}_i \odot \boldsymbol{k}_i^\top \boldsymbol{v}_i$, $\mathbf{G}_i = \prod_{k=1}^{i} \sigma(\boldsymbol{x}_k \mathbf{W}_g) \mathbf{1}^\top \mathbf{1}$.

**Fixed Attention**  We additionally explore an autoregressive structure with fixed, data-independent weights $w_{t,i}$, replacing the dynamically generated attention weights $\phi(\boldsymbol{q}_t)\phi(\boldsymbol{k}_i)$. Without dynamic parameter generation, this becomes a linear layer with a causal mask $\mathbf{M}$ rather than a true attention mechanism. We use this structure to examine the effect of adding a moving-average term. This autoregressive causal linear layer is expressed as: $\boldsymbol{o}_t = \sum_{i=1}^{t} w_{t,i} \boldsymbol{v}_i$.

## 2.5. Inspiration: decoupling the short-term impact

In sequence modeling tasks like NLP, context tokens closer to the output token typically carry higher importance. As a result, a gating mechanism with exponential decay in gated linear attention can significantly enhance the performance by assigning greater weights to nearby tokens. Additionally, NLP tasks require retrieval of long-term information. Even though the decay factor reduces the weights for long-term tokens, the AR weights' ability to capture long-term de-

pendencies allows for effective retrieval within moderately sized lookback windows.

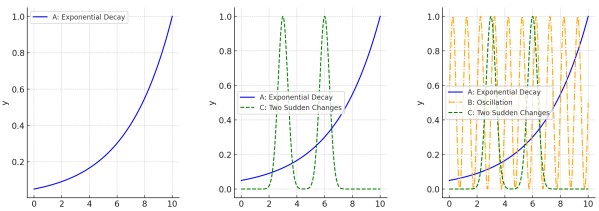

*Figure 2.* Visualization of different effects with exponential decay strategies and their challenges in gated linear attention. (Left: a) Pure exponential decay strategy in gated linear attention; (Mid: b) Exponential decay facing challenges in capturing long-term dependencies; (Right: c) Exponential decay facing challenges in capturing periodic dependencies

However, applying exponential decay to the gated linear attention AR weights may not align with the needs of TSF, which often involves stable periodic patterns alongside short-term impacts. TSF data frequently exhibit seasonal effects that differ from the transient long-term effects in NLP. These seasonal effects are stable and persist across the temporal dimension without decaying. As shown in Figure 2, in scenarios involving seasonal effects, sudden changes, and local effects, relying solely on exponential decay in gated linear attention is not the most suitable approach for modeling TSF. Decoupling local effects into short-term MA weights, allowing the AR term to focus on modeling the seasonal and long-term effects it handles best, would be a better solution for TSF.

## 2.6. WAVE attention mechanism

In the attention mechanisms above, the next-step prediction at time $t$ is a weighted sum of all previous values $\boldsymbol{v}_i \in \mathbb{R}^{1 \times d}$, with weights $\mathbf{w}_{t,i} \in \mathbb{R}^{1 \times d}$ derived from interactions between $\boldsymbol{q}_t$ and $\boldsymbol{k}_i$. Naturally, we can write these attention

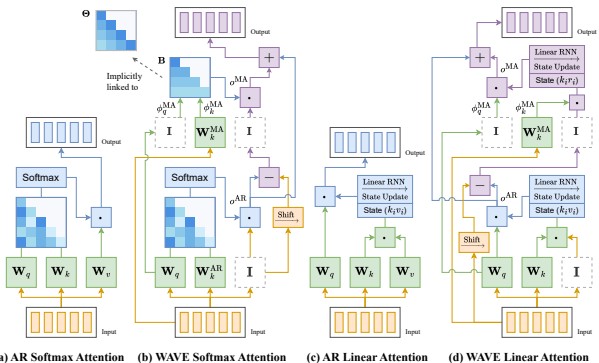

*Figure 3.* WAVE attention structure with the indirect MA weight generation method applied to softmax and linear attention. See Table 1 for more calculation details.

mechanisms in an AR model structure:

$$\boldsymbol{v}_{t+1} = \boldsymbol{o}_t^{\mathrm{AR}} + \boldsymbol{r}_t = \sum_{i=1}^{t} \mathbf{w}_{t,i} \odot \boldsymbol{v}_i + \boldsymbol{r}_t,$$

where $\boldsymbol{r}_t$ is the AR error. In an ARMA model, the MA term captures short-term fluctuations, allowing the AR component to focus on long-term dependencies. Let $\boldsymbol{\epsilon}_t$ be the error after introducing the MA term and $\boldsymbol{\theta}_{t-1,j}$ the MA weights generated by some attention mechanism. We expand the AR error $\boldsymbol{r}_t$ into an MA form and extend the model to an ARMA structure as:

$$
\begin{aligned}
\boldsymbol{v}_{t+1} &= \boldsymbol{o}_t^{\mathrm{AR}} + \boldsymbol{o}_t^{\mathrm{MA}} + \boldsymbol{\epsilon}_t \\
&= \sum_{i=1}^{t} \mathbf{w}_{t,i} \odot \boldsymbol{v}_i + \sum_{j=1}^{t-1} \boldsymbol{\theta}_{t-1,j} \odot \boldsymbol{\epsilon}_j + \boldsymbol{\epsilon}_t, \\
\boldsymbol{r}_t &= \sum_{j=1}^{t-1} \boldsymbol{\theta}_{t-1,j} \odot \boldsymbol{\epsilon}_j + \boldsymbol{\epsilon}_t
\end{aligned}
\tag{2}
$$

The structure of the MA output $\boldsymbol{o}_t^{\mathrm{MA}} = \sum_{j=1}^{t-1} \boldsymbol{\theta}_{t-1,j} \odot \boldsymbol{\epsilon}_j$ resembles the AR term and could potentially be computed using an attention mechanism. For simplicity, we consider a single channel of the $d$-dimensional space, with other channels an be handled in parallel. We express the matrix form of the $\boldsymbol{r}_t$ in Eq. (2) for one channel as:

$$
\begin{pmatrix} r_1 \\ r_2 \\ \vdots \\ r_t \end{pmatrix} =
\begin{pmatrix}
0 & 0 & \cdots & 0 & 0 \\
\theta_{1,1} & 0 & \cdots & 0 & 0 \\
\theta_{2,1} & \theta_{2,2} & \cdots & 0 & 0 \\
\vdots & \vdots & \ddots & \vdots & \vdots \\
\theta_{t-1,1} & \theta_{t-1,2} & \cdots & \theta_{t-1,t-1} & 0
\end{pmatrix}
\begin{pmatrix} \epsilon_1 \\ \epsilon_2 \\ \vdots \\ \epsilon_t \end{pmatrix} +
\begin{pmatrix} \epsilon_1 \\ \epsilon_2 \\ \vdots \\ \epsilon_t \end{pmatrix}
\tag{3}
$$

$$\mathbf{r} = (\mathbf{I} + \boldsymbol{\Theta})\boldsymbol{\epsilon}, \quad \boldsymbol{\epsilon} = (\mathbf{I} + \boldsymbol{\Theta})^{-1}\mathbf{r},$$

where $\boldsymbol{\Theta}$ is a strictly lower triangular matrix of MA weights for this channel. Once the attention mechanism determines $\boldsymbol{o}_t^{\mathrm{AR}}$ and $\boldsymbol{\theta}_{t-1,j}$, we can calculate $\boldsymbol{r}_j = \boldsymbol{v}_{j+1} - \boldsymbol{o}_j^{\mathrm{AR}}$ (token shifting) for all $j \leq N-1$ and determine $\boldsymbol{\epsilon}_j$ via matrix inversion. Substituting these back into Eq. (2) yields the final WAVE attention output $\boldsymbol{o}_t^{\mathrm{AR}} + \boldsymbol{o}_t^{\mathrm{MA}}$ for step $t$.

However, computing $\boldsymbol{o}_t^{\mathrm{MA}}$ requires inverting $\mathbf{I} + \boldsymbol{\Theta}$, which involves calculating all $\boldsymbol{\theta}_{t-1,j}$ in the $N \times N$ matrix. This increases the complexity of linear attentions back to $O(N^2)$ and may also cause training instability. To maintain linear time complexity, we need a method to compute $\boldsymbol{o}_t^{\mathrm{MA}}$ without explicitly calculating all $\boldsymbol{\theta}_{t-1,j}$ values.

## 2.7. Indirect MA weight generation

We need an approach that can leverage linear attention's efficiency to compute $\boldsymbol{o}_t^{\mathrm{MA}}$ without the costly $\boldsymbol{\Theta}^{N \times N}$ matrix operations. Instead of separately calculating attention weights to determine $\boldsymbol{\epsilon}_j$ as value input and recomputing the whole MA output, we aim to use a linear RNN to collect all keys and values at once. We observe from Eq. (3) that

*Table 1.* Summary of WAVE attention for various attention mechanisms, detailing the calculation methods for AR output and MA output, where $\boldsymbol{r}_j = \boldsymbol{v}_{j+1} - \boldsymbol{o}_j^{\mathrm{AR}}$.

| Model | AR term output $\boldsymbol{o}_t^{\mathrm{AR}}$ | Indirect MA term output $\boldsymbol{o}_t^{\mathrm{MA}}$ |
|---|---|---|
| Standard Softmax Attention (Std Attn) | $\frac{\sum_{i=1}^{t} \exp(\boldsymbol{q}_t(\boldsymbol{k}_i^{\mathrm{AR}})^\top)\boldsymbol{v}_i}{\sum_{i=1}^{t} \exp(\boldsymbol{q}_t(\boldsymbol{k}_i^{\mathrm{AR}})^\top)}$ | $\sum_{j=1}^{t-1} \phi_q^{\mathrm{MA}}(\boldsymbol{q}_{t-1})\phi_k^{\mathrm{MA}}(\boldsymbol{k}_j^{\mathrm{MA}})^\top \boldsymbol{r}_j$ |
| Linear Attention (Lin Attn) | $\boldsymbol{q}_t \sum_{i=1}^{t}(\boldsymbol{k}_i^{\mathrm{AR}})^\top \boldsymbol{v}_i$ | $\phi_q^{\mathrm{MA}}(\boldsymbol{q}_{t-1}) \sum_{j=1}^{t-1} \phi_k^{\mathrm{MA}}(\boldsymbol{k}_j^{\mathrm{MA}})^\top \boldsymbol{r}_j$ |
| Element-wise Linear Attention (ELin Attn) | $\sigma(\boldsymbol{q}_t) \odot \frac{\sum_{i=1}^{t} \exp(\boldsymbol{k}_i^{\mathrm{AR}}) \odot \boldsymbol{v}_i}{\sum_{i=1}^{t} \exp(\boldsymbol{k}_i^{\mathrm{AR}})}$ | $\phi_q^{\mathrm{MA}}(\boldsymbol{q}_{t-1}) \odot \sum_{j=1}^{t-1} \phi_k^{\mathrm{MA}}(\boldsymbol{k}_j^{\mathrm{MA}}) \odot \boldsymbol{r}_j$ |
| Gated Linear Attention (GLin Attn) | $\boldsymbol{q}_t \sum_{i=1}^{t} \mathbf{G}_i \odot (\boldsymbol{k}_i^{\mathrm{AR}})^\top \boldsymbol{v}_i$ | $\phi_q^{\mathrm{MA}}(\boldsymbol{q}_{t-1}) \sum_{j=1}^{t-1} \phi_k^{\mathrm{MA}}(\boldsymbol{k}_j^{\mathrm{MA}})^\top \boldsymbol{r}_j$ |
| Fixed Attention (Fixed Attn) | $\sum_{i=1}^{t} w_{t,i}^{\mathrm{AR}} \boldsymbol{v}_i$ | $\phi_q^{\mathrm{MA}}(\boldsymbol{w}_{t-1}^{\mathrm{MA,q}}) \sum_{j=1}^{t-1} \phi_k^{\mathrm{MA}}(\boldsymbol{w}_j^{\mathrm{MA,k}})^\top \boldsymbol{r}_j$ |

there is already a sequential relationship between $\boldsymbol{r}_j$ and $\boldsymbol{\epsilon}_j$, and $\boldsymbol{r}_j$ can be computed directly once $\boldsymbol{o}_t^{\mathrm{AR}}$ is determined. Therefore, we implicitly compute the MA weights of $\boldsymbol{\epsilon}_j$ by using $\boldsymbol{r}_j$ as value input for the MA component instead of $\boldsymbol{\epsilon}_j$. Let $\boldsymbol{\beta}_{t-1,j}$ denote the generated attention weights corresponding to $\boldsymbol{r}_j$ at step $t$, and let $\boldsymbol{\theta}_{t-1,j}$ here be the implicit MA weights hiddenly linked to the generated $\boldsymbol{\beta}_{t-1,j}$. Based on Eq. (3), for each channel, we establish:

$$\sum_{j=1}^{t-1} \boldsymbol{\beta}_{t-1,j} \odot \boldsymbol{r}_j = \sum_{j=1}^{t-1} \boldsymbol{\theta}_{t-1,j} \odot \boldsymbol{\epsilon}_j \Leftrightarrow \mathbf{B}\boldsymbol{r} = \boldsymbol{\Theta}\boldsymbol{\epsilon}$$

$$\mathbf{B} = \boldsymbol{\Theta} \cdot (\mathbf{I} + \boldsymbol{\Theta})^{-1}, \quad \boldsymbol{\Theta} = \mathbf{B} \cdot (\mathbf{I} - \mathbf{B})^{-1} \qquad (4)$$

With $\boldsymbol{\Theta} = \mathbf{B} \cdot (\mathbf{I} - \mathbf{B})^{-1}$, as long as the indirectly generated $\boldsymbol{\Theta}$ accurately reflects the characteristics of the MA weights we want, we can use $\sum_{j=1}^{t-1} \boldsymbol{\beta}_{t-1,j} \odot \boldsymbol{r}_j$ as $\boldsymbol{o}_t^{\mathrm{MA}}$. Since $\boldsymbol{r}_j$ is known after computing $\boldsymbol{o}_t^{\mathrm{AR}}$, linear attention can be used to compute $\boldsymbol{o}_t^{\mathrm{MA}}$ without increasing the time complexity. To ensure the implicitly generated $\boldsymbol{\Theta}$ from $\mathbf{B}$ captures the desired MA properties, we must carefully design how $\mathbf{B}$ is generated. The invertibility of $(\mathbf{I} - \mathbf{B})^{-1}$ is guaranteed since $\mathbf{B}$ is strictly lower triangular. To efficiently compute the generated weights, we use the $\boldsymbol{\beta}_{t-1,j} = \phi_q^{\mathrm{MA}}(\boldsymbol{q}_{t-1}^{\mathrm{MA}})\phi_k^{\mathrm{MA}}(\boldsymbol{k}_j^{\mathrm{MA}})$ to generate $\mathbf{B}$, similar to linear attention. Previous dynamic ARMA models in statistics often update MA weights based on observations (Grenier, 1983; Azrak & Mélard, 2006), so we derive $\boldsymbol{q}_{t-1}^{\mathrm{MA}}$ and $\boldsymbol{k}_j^{\mathrm{MA}}$ by multiplying the attention input $\boldsymbol{x}_{[\cdot]}$ with $\mathbf{W}_q^{\mathrm{MA}}$ and $\mathbf{W}_k^{\mathrm{MA}}$. Now, the effectiveness of MA weights lies in selecting the most suitable functions $\phi_q^{\mathrm{MA}}(\cdot)$ and $\phi_k^{\mathrm{MA}}(\cdot)$.

## 2.8. Selection of $\phi(\cdot)$ and characteristics of implicit MA weights

The MA term models short-term effects and local temporal relationships, so we want the implicit $\boldsymbol{\Theta}$ to follow a pattern where elements near the diagonal have larger absolute values, and those farther away gradually decrease. The expanded form of $\boldsymbol{\Theta}$ is given by $\boldsymbol{\Theta} = \mathbf{B} \cdot (\mathbf{I} - \mathbf{B})^{-1} = \mathbf{B} + \mathbf{B}^2 + \mathbf{B}^3 + \cdots$. The elements along the diagonal direction in $\mathbf{B}$ continually accumulate as products into the elements below them in $\boldsymbol{\Theta}$. Since $\mathbf{B}$ is strictly lower trian-

gular, the elements of the subdiagonal in $\boldsymbol{\Theta}$ remain constant, while the elements further down progressively accumulate additional terms formed by the product of different $\beta_{[\cdot]}$ elements above. Assuming $\beta_{[\cdot]}$ follows a distribution and simplifying by setting each $\beta_{[\cdot]}$ to the distribution mean $b$, the elements of $\boldsymbol{\Theta}$ can be expressed as:

$$\theta_{ij} = b(1+b)^{i-j-1}, \quad \text{where } i > j \qquad (5)$$

This simplification offers valuable insights. To prevent longer-term errors from having a larger impact, we aim to avoid large absolute values accumulating in $\boldsymbol{\Theta}$ far from the diagonal. We also want $\theta_\cdot$ to decay steadily as it moves away from the diagonal. Therefore, constraining $\beta_\cdot$ between -1 and 0, with a preference of smaller absolute values, is a practical approach.

We tested various activation function combinations for $\phi_q^{\mathrm{MA}}(\cdot)$ and $\phi_k^{\mathrm{MA}}(\cdot)$ to generate $\boldsymbol{\beta}_{t-1,j} = \phi_q^{\mathrm{MA}}(\boldsymbol{q}_{t-1}^{\mathrm{MA}})\phi_k^{\mathrm{MA}}(\boldsymbol{k}_j^{\mathrm{MA}})$ values, as shown in Fig. 4. We used the sigmoid function $\phi_k^{\mathrm{MA}}(\boldsymbol{k}_j^{\mathrm{MA}}) = \sigma(\alpha\boldsymbol{k}_j^{\mathrm{MA}}/\sqrt{d})$ to obtain values between 0 and 1, where $\alpha = 0.05$ [1] and $\sqrt{d}$ are scaling factors to maintain small absolute values. Then, we selected a function $\phi_q^{\mathrm{MA}}(\cdot)$ to make the product negative. We ultimately chose $\phi_q^{\mathrm{MA}}(\boldsymbol{q}_t^{\mathrm{MA}}) = -\mathrm{LeakyReLU}(-\boldsymbol{q}_t^{\mathrm{MA}}/\sqrt{d})$ with a negative slope of 0.02. The inner negative sign maintains directional consistency (for later parameter sharing), and the outer negative sign encourages a negative output.

Fig. 4 shows that LeakyReLU provides a balanced lag weight pattern. Unlike ReLU and Sigmoid, which only output values of the same sign, LeakyReLU offers some relaxation while keeping most values negative. This adds flexibility by enabling the desired negative smoothing effect

---

[1]In the key activation, $\alpha$ controls the variance of each row in the $\mathbf{B}$ matrix, indirectly influencing the amount of long-term information (lower left) in the MA weights $\boldsymbol{\Theta}$. Increasing $\alpha$ would make the MA weights focus more on modeling long-term information. However, since we want the AR weights to handle the long-term component, we set $\alpha$ to a relatively small value. This explains why the rows of the $\mathbf{B}$ matrix appear smooth in the visualization. Refer to Fig. 7 for more details on $\alpha$, and see Fig. 8 for the effects of reversed positive $\phi_q$.

*Table 2.* Summary of main TSF results with forecasting horizons $L_P \in \{12, 24, 48, 96\}$ and $L_I = 512$. See Table 8 for the original results. Averages of test set MSE for each model on each dataset are presented. Average rankings (AvgRank) of each model, along with the count of first-place rankings (#Top1), are also included.

| Model | Pure AR / WAVE Transformer | | | | | | | | | | Baseline | | | | | |
| | Std Attn | Std Attn +ARMA | Lin Attn | Lin Attn +ARMA | GLin Attn | GLin Attn +ARMA | ELin Attn | ELin Attn +ARMA | Fixed Attn | Fixed Attn +ARMA | FITS | iTrans-former | CATS | PatchTST | DLinear | Enc-Former |
|---|---|---|---|---|---|---|---|---|---|---|---|---|---|---|---|---|
| Weather | 0.104 | 0.101 | 0.104 | 0.100 | 0.119 | 0.105 | 0.104 | 0.103 | 0.105 | 0.104 | 0.114 | 0.117 | 0.105 | 0.107 | 0.124 | 0.135 |
| Solar | 0.134 | 0.124 | 0.122 | 0.119 | 0.148 | 0.124 | 0.136 | 0.133 | 0.142 | 0.135 | 0.152 | 0.145 | 0.122 | 0.150 | 0.149 | 0.125 |
| ECL | 0.110 | 0.106 | 0.106 | 0.104 | 0.110 | 0.108 | 0.115 | 0.114 | 0.121 | 0.118 | 0.124 | 0.106 | 0.110 | 0.111 | 0.114 | 0.201 |
| ETTh1 | 0.323 | 0.318 | 0.318 | 0.316 | 0.408 | 0.321 | 0.323 | 0.321 | 0.330 | 0.328 | 0.333 | 0.351 | 0.327 | 0.335 | 0.329 | 0.817 |
| ETTh2 | 0.192 | 0.192 | 0.193 | 0.195 | 0.217 | 0.198 | 0.193 | 0.190 | 0.200 | 0.194 | 0.197 | 0.229 | 0.194 | 0.201 | 0.198 | 0.597 |
| ETTm1 | 0.264 | 0.239 | 0.238 | 0.222 | 0.407 | 0.260 | 0.246 | 0.244 | 0.267 | 0.251 | 0.237 | 0.259 | 0.222 | 0.244 | 0.235 | 0.429 |
| ETTm2 | 0.131 | 0.128 | 0.126 | 0.121 | 0.142 | 0.128 | 0.134 | 0.128 | 0.129 | 0.127 | 0.115 | 0.135 | 0.116 | 0.119 | 0.120 | 0.311 |
| Traffic | 0.341 | 0.333 | 0.337 | 0.330 | 0.429 | 0.350 | 0.352 | 0.348 | 0.373 | 0.365 | 0.385 | 0.330 | 0.372 | 0.358 | 0.375 | 0.847 |
| PEMS03 | 0.112 | 0.100 | 0.100 | 0.096 | 0.209 | 0.101 | 0.116 | 0.112 | 0.121 | 0.116 | 0.133 | 0.096 | 0.105 | 0.140 | 0.134 | 0.111 |
| PEMS04 | 0.118 | 0.106 | 0.103 | 0.098 | 0.167 | 0.105 | 0.122 | 0.119 | 0.128 | 0.124 | 0.151 | 0.098 | 0.108 | 0.164 | 0.148 | 0.099 |
| PEMS07 | 0.092 | 0.083 | 0.087 | 0.077 | 0.093 | 0.087 | 0.101 | 0.097 | 0.106 | 0.100 | 0.132 | 0.079 | 0.094 | 0.093 | 0.129 | 0.102 |
| PEMS08 | 0.148 | 0.132 | 0.119 | 0.116 | 0.159 | 0.125 | 0.150 | 0.144 | 0.161 | 0.152 | 0.201 | 0.117 | 0.135 | 0.121 | 0.193 | 0.183 |
| AvgRank | 7.958 | 4.292 | 4.271 | 2.333 | 12.375 | 6.229 | 8.938 | 7.125 | 11.146 | 8.688 | 12.042 | 7.271 | 5.938 | 9.792 | 11.250 | 12.813 |
| #Top1 | 0 | 4 | 4 | 25 | 0 | 1 | 1 | 3 | 1 | 3 | 0 | 5 | 4 | 1 | 2 | 4 |

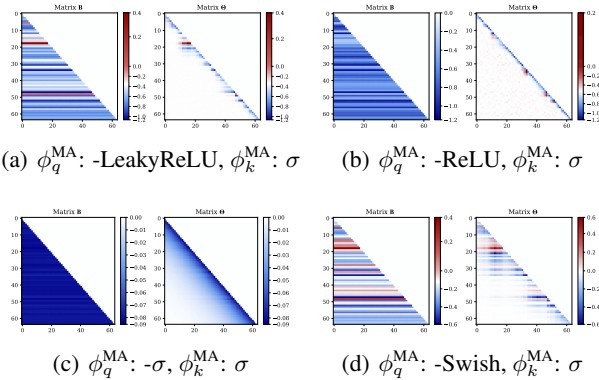

(a) $\phi_q^{\text{MA}}$: -LeakyReLU, $\phi_k^{\text{MA}}$: $\sigma$  (b) $\phi_q^{\text{MA}}$: -ReLU, $\phi_k^{\text{MA}}$: $\sigma$

(c) $\phi_q^{\text{MA}}$: -$\sigma$, $\phi_k^{\text{MA}}$: $\sigma$  (d) $\phi_q^{\text{MA}}$: -Swish, $\phi_k^{\text{MA}}$: $\sigma$

*Figure 4.* Visualization of the $\mathbf{B}$(left) $-\,\boldsymbol{\Theta}$(right) relationship with different $\phi(\cdot)$. We construct the simulated $\mathbf{B}$ matrices using randomly sampled $\boldsymbol{q}$ and $\boldsymbol{k}$ ($N = 64$, $d = 32$) from the normal distribution, and display the corresponding implicit $\boldsymbol{\Theta}$ matrices.

of the MA term, with occasional positive values to enhance modeling flexibility.

To summarize the WAVE attention process with indirect MA weight generation: First, we compute all $\boldsymbol{o}_t^{\text{AR}}$ using the selected attention mechanism. Then, we apply token shifting and compute all $\boldsymbol{r}_j$ for $j \leq N - 1$. Next, using $\phi_q^{\text{MA}}(\boldsymbol{q}_t^{\text{MA}})$, $\phi_k^{\text{MA}}(\boldsymbol{k}_j^{\text{MA}})$, and $\boldsymbol{r}_j$, we calculate $\boldsymbol{o}_t^{\text{MA}}$ with the efficient method matching AR attention, as illustrated in Fig. 3. Finally, the ARMA output is $\boldsymbol{o}_t = (\boldsymbol{o}_t^{\text{AR}} + \boldsymbol{o}_t^{\text{MA}})\mathbf{W}_o$. A summary of MA computation methods for each attention mechanism is in Table 1.

**Computational cost and model performance** The introduction of MA term adds three weight matrices $\mathbf{W}_{\{q,k,v\}}^{\text{MA}}$, increasing parameter size. To ensure fair comparison, we use weight-sharing to match the parameter sizes of ARMA and AR models. Specifically, we share $\mathbf{W}_q$ between the AR and MA terms and set $\mathbf{W}_v$ to an identity matrix, with minimal impact due to the existance of $\mathbf{W}_o$ and the MLP layer (see Eq. (1)). This reduces ARMA's trainable weights to

$\mathbf{W}_q, \mathbf{W}_k^{\text{AR}}, \mathbf{W}_k^{\text{MA}}, \mathbf{W}_o$, as shown in Fig. 3. While WAVE attention has the same time complexity in order of magnitude as efficient AR attention, its two-stage structure may increase computational costs on constant level. We compare models with different number of layer in the experiments section to show that ARMA's improved performance is due to structural enhancements, but not increased complexity.

## 3. Experiments

We conducted comprehensive experiments on 12 widely-used TSF datasets, including Weather, Solar, Electricity (ECL), ETTs, Traffic, and PEMS. See §A.2 for detailed description of datasets.

**Baselines** We built AR Transformers using the five attention mechanisms from Table 1 and added MA terms to create WAVE attention for comparison in TSF tasks. Additionally, we included five recent SOTA baselines: FITS (Xu et al., 2024), iTransformer (Liu et al., 2024b), CATS (Lu et al., 2024), PatchTST (Nie et al., 2022), and DLinear (Zeng et al., 2023). We also used a simple channel-dependent encoder-only Transformer, modified by repeating the last input value (like NLinear) to address distribution shift. This model already surpasses older architectures like Autoformer (Wu et al., 2021) and Informer (Zhou et al., 2021), so we excluded these from our comparison.

In the main experiments, both pure AR and WAVE Transformers use a consistent setup: $m = 3$ Transformer layers, 8 heads, and model dimension determined by a empirical method $d = 16\sqrt{C}$, where $C$ is the number of series. We evaluate their performance using one-step prediction for each test datapoint, aligned with the baselines. Baseline hyperparameters are set to the reported values from their original papers. For more details on hyperparameters and implementation, see §A.3.

We ran all models on all datasets for the four different $L_P$. In the main text, we report the average test set MSE for each

*Table 3.* Summary showing that pure AR/WAVE Transformers effectively utilize extended lookback $L_I$, while baselines experience performance degradation. $L_I \in \{512, 1024, 2048, 4096\}$ with $L_P \in \{12, 24, 48, 96\}$ are evaluated and averaged. Original results can be found in Table 10.

| | Model | Pure AR/WAVE Transformer | | | | | | | | | | Baseline | | | | | |
|---|---|---|---|---|---|---|---|---|---|---|---|---|---|---|---|---|---|
| | | Std Attn | Std Attn +ARMA | Lin Attn | Lin Attn +ARMA | GLin Attn | GLin Attn +ARMA | ELin Attn | ELin Attn +ARMA | Fixed Attn | Fixed Attn +ARMA | FITS | iTrans-former | CATS | PatchTST | DLinear | Enc-Former |
| Weather | $L_I = 512$ | 0.104 | 0.101 | 0.104 | 0.100 | 0.119 | 0.105 | 0.104 | 0.103 | 0.105 | 0.104 | 0.114 | 0.117 | 0.105 | 0.108 | 0.124 | 0.135 |
| | $L_I = 1024$ | 0.107 | 0.102 | 0.102 | 0.101 | 0.116 | 0.104 | 0.106 | 0.106 | 0.108 | 0.105 | 0.120 | 0.117 | 0.108 | 0.120 | 0.118 | 0.124 |
| | $L_I = 2048$ | 0.110 | 0.102 | 0.101 | 0.100 | 0.114 | 0.102 | 0.108 | 0.108 | 0.123 | 0.110 | 0.121 | 0.119 | 0.113 | 0.122 | 0.119 | 0.128 |
| | $L_I = 4096$ | 0.108 | 0.102 | 0.100 | 0.100 | 0.115 | 0.105 | 0.109 | 0.107 | 0.110 | 0.108 | 0.124 | 0.132 | 0.123 | 0.125 | 0.121 | 0.136 |
| ETTm1 | $L_I = 512$ | 0.264 | 0.239 | 0.238 | 0.222 | 0.407 | 0.260 | 0.246 | 0.244 | 0.267 | 0.251 | 0.237 | 0.259 | 0.222 | 0.244 | 0.235 | 0.429 |
| | $L_I = 1024$ | 0.280 | 0.241 | 0.239 | 0.227 | 0.423 | 0.236 | 0.265 | 0.253 | 0.281 | 0.263 | 0.240 | 0.258 | 0.238 | 0.245 | 0.239 | 0.364 |
| | $L_I = 2048$ | 0.278 | 0.239 | 0.233 | 0.223 | 0.327 | 0.232 | 0.281 | 0.252 | 0.288 | 0.268 | 0.246 | 0.248 | 0.261 | 0.250 | 0.239 | 0.415 |
| | $L_I = 4096$ | 0.275 | 0.234 | 0.237 | 0.226 | 0.324 | 0.229 | 0.282 | 0.265 | 0.287 | 0.266 | 0.252 | 0.274 | 0.340 | 0.260 | 0.250 | 0.428 |

*Table 4.* Summary showing that WAVE Transformers with $m = 3$ layers consistently outperform their AR counterparts across a wide range of $m$. The same experimental settings and data presentation method as in Table 2 are used. See Table 9 for the original results.

| | Model | $m = 3$ WAVE | $m = 1$ Pure AR | $m = 2$ Pure AR | $m = 3$ Pure AR | $m = 4$ Pure AR | $m = 5$ Pure AR | $m = 6$ Pure AR | $m = 7$ Pure AR | $m = 8$ Pure AR |
|---|---|---|---|---|---|---|---|---|---|---|
| Weather | Std Attn | 0.101 | 0.109 | 0.108 | 0.104 | 0.108 | 0.113 | 0.111 | 0.113 | 0.112 |
| | Lin Attn | 0.100 | 0.104 | 0.103 | 0.104 | 0.103 | 0.103 | 0.103 | 0.102 | 0.103 |
| | GLin Attn | 0.105 | 0.122 | 0.122 | 0.119 | 0.121 | 0.121 | 0.122 | 0.121 | 0.120 |
| | ELin Attn | 0.103 | 0.110 | 0.107 | 0.104 | 0.108 | 0.109 | 0.111 | 0.110 | 0.111 |
| | Fixed Attn | 0.104 | 0.113 | 0.109 | 0.105 | 0.110 | 0.112 | 0.110 | 0.110 | 0.110 |
| ETTm1 | Std Attn | 0.239 | 0.265 | 0.270 | 0.264 | 0.266 | 0.269 | 0.270 | 0.270 | 0.272 |
| | Lin Attn | 0.222 | 0.241 | 0.233 | 0.238 | 0.232 | 0.230 | 0.230 | 0.231 | 0.231 |
| | GLin Attn | 0.260 | 0.411 | 0.413 | 0.407 | 0.409 | 0.410 | 0.410 | 0.409 | 0.404 |
| | ELin Attn | 0.244 | 0.253 | 0.251 | 0.246 | 0.253 | 0.257 | 0.259 | 0.256 | 0.258 |
| | Fixed Attn | 0.251 | 0.269 | 0.264 | 0.267 | 0.260 | 0.258 | 0.259 | 0.258 | 0.257 |

model across different $L_P$ on each dataset and provide the full results in §A.5.

**Short-term TSF results** Table 2 highlights the significant performance gains from introducing MA terms to the AR Transformers. All WAVE attention mechanisms outperform their AR counterparts in both average test MSE and ranking, with linear and standard attention showing the best results.

**Long-term TSF results** We evaluated pure AR/WAVE Transformers with varying input lengths ($L_I$) for different prediction horizons ($L_P$): $(L_I, L_P)$ : $(1024, 96), (2048, 192), (2048, 336), (4096, 720)$. For the baseline models, we selected the best-performing results across multiple input lengths $L_I \in \{512, 1024, 2048, 4096\}$ for each prediction horizon. As shown in Table 5, AR models demonstrated comparable performance to baselines, and the incorporation of the ARMA structure consistently yielded improved results over the AR models across all prediction horizons.

**Performance of linear attention** Linear attention outperforms softmax attention in TSF, suggesting that simpler attention patterns and non-normalized input shortcuts (without denominator) can improve generalization on time-varying distributions. This aligns with earlier findings where linear models can outperform more complex Transformers in TSF (Zeng et al., 2023; Xu et al., 2024).

**Performance of gated linear attention** WAVE brought the greatest improvement to gated linear attention. In gated AR models, the decay factor helps the AR term focus on important local patterns, but it weakens the ability to capture long-term or stable cyclic patterns. By introducing the MA

term, local effects are absorbed, allowing the decay factor to function properly in the AR forgetting mechanism, leading to significant performance gains.

**Performance of fixed attention** Fixed attention, which lacks dynamic parameter generation, performs worse than other attention. However, its significant improvement with MA terms shows that WAVE enhances the model's ability structurally to capture comprehensive sequence patterns.

**Performance and complexity** The improvement of adding the MA term comes from its ability to model short-term impacts, allowing the AR term to focus on long-term and cyclic effects, not from increased computational costs. Table 4 shows that, regardless of the number of layers $m$ (1 to 8), pure AR Transformers consistently underperform compared to WAVE Transformers with a fixed $m = 3$.

**Adaptability to Longer $L_I$** Previous baseline models typically use $L_I$ between 96 and 720, as longer $L_I$ often leads to overfitting to long-term patterns, ignoring more important local effects (Zeng et al., 2023; Nie et al., 2022; Liu et al., 2024b). However, the next-step prediction and varying look-back inputs in pure AR/WAVE Transformers help the model focus on tokens closer to the next step, improving generalization. As shown in Table 3, increasing $L_I$ from 512 to 4096 improves pure AR/WAVE performance, demonstrating scalability and the ability to properly leverage long-term effects. Also, the WAVE structure consistently boosts AR model performance across different $L_I$.

**Comparison to MEGA** The MEGA structure (Ma et al., 2022) uses an exponential moving average (EMA) in gated attention to model local patterns. However, applying EMA

*Table 5.* Summary of long-term time series forecasting for $L_P \in \{96, 192, 336, 720\}$. Pure AR/WAVE Transformers uses $(L_I, L_P)$ : $(1024, 96), (2048, 192), (2048, 336), (4096, 720)$ and we choose the best results for the baselines from $L_I \in \{512, 1024, 2048, 4096\}$ for the 4 $L_P$ settings. See Fig. 11 for the original results.

| Model | Pure AR/WAVE Transformer | | | | | | | | | | Baseline | | | | | |
|---|---|---|---|---|---|---|---|---|---|---|---|---|---|---|---|---|
| | Std Attn | Std Attn +ARMA | Lin Attn | Lin Attn +ARMA | GLin Attn | GLin Attn +ARMA | ELin Attn | ELin Attn +ARMA | Fixed Attn | Fixed Attn +ARMA | FITS | iTrans-former | CATS | PatchTST | DLinear | Enc-Former |
| Weather | 0.221 | 0.218 | 0.218 | 0.215 | 0.223 | 0.216 | 0.220 | 0.219 | 0.220 | 0.218 | 0.222 | 0.232 | 0.216 | 0.221 | 0.233 | 0.251 |
| Solar | 0.198 | 0.195 | 0.196 | 0.192 | 0.204 | 0.193 | 0.198 | 0.195 | 0.199 | 0.195 | 0.209 | 0.219 | 0.206 | 0.202 | 0.216 | 0.212 |
| ETTh1 | 0.414 | 0.411 | 0.415 | 0.411 | 0.417 | 0.408 | 0.409 | 0.405 | 0.414 | 0.410 | 0.440 | 0.454 | 0.408 | 0.413 | 0.422 | 0.906 |
| ETTh2 | 0.340 | 0.339 | 0.343 | 0.339 | 0.342 | 0.340 | 0.337 | 0.332 | 0.348 | 0.344 | 0.354 | 0.374 | 0.320 | 0.330 | 0.426 | 0.877 |
| ETTm1 | 0.347 | 0.345 | 0.351 | 0.348 | 0.357 | 0.346 | 0.348 | 0.345 | 0.347 | 0.344 | 0.354 | 0.373 | 0.345 | 0.346 | 0.347 | 0.735 |
| ETTm2 | 0.249 | 0.246 | 0.247 | 0.243 | 0.250 | 0.245 | 0.246 | 0.244 | 0.245 | 0.240 | 0.247 | 0.265 | 0.243 | 0.247 | 0.252 | 0.576 |

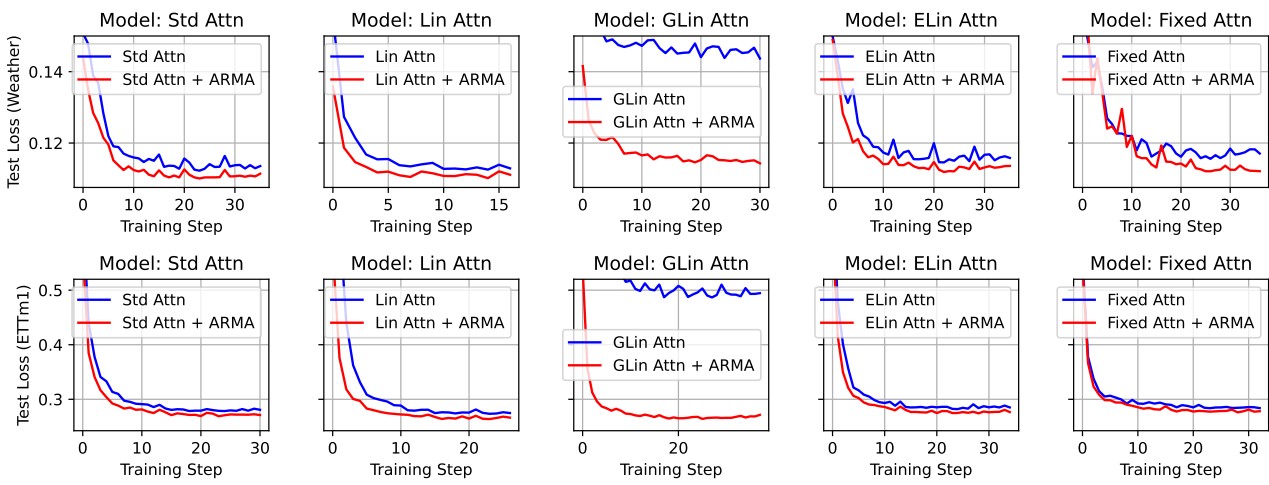

*Figure 5.* Visualization of test loss curves. We show the testing performance of five attention mechanisms using pure AR/WAVE structures on the Weather and ETTm1 datasets ($L_I = 512, L_P = 48$).

*Table 6.* Summary of the performance comparison with MEGA. See Table 12 for the original result.

| Model | Std Attn | Std Attn +ARMA | Lin Attn | Lin Attn +ARMA | GLin Attn | GLin Attn +ARMA | MEGA |
|---|---|---|---|---|---|---|---|
| Weather | 0.104 | 0.101 | 0.104 | 0.100 | 0.119 | 0.105 | 0.121 |
| Solar | 0.134 | 0.124 | 0.122 | 0.119 | 0.148 | 0.124 | 0.226 |
| ETTh1 | 0.323 | 0.318 | 0.318 | 0.316 | 0.408 | 0.321 | 0.404 |
| ETTh2 | 0.192 | 0.192 | 0.193 | 0.195 | 0.217 | 0.198 | 0.214 |
| ETTm1 | 0.264 | 0.239 | 0.238 | 0.222 | 0.407 | 0.260 | 0.412 |
| ETTm2 | 0.131 | 0.128 | 0.126 | 0.121 | 0.142 | 0.128 | 0.137 |
| PEMS03 | 0.112 | 0.100 | 0.100 | 0.096 | 0.209 | 0.101 | 0.161 |

directly to AR weights weakens the model's ability to capture long-term and stable seasonal patterns, making it less effective than ARMA at decoupling long-term and short-term effects. Table 6 shows that the performance of using MEGA as the attention mechanism is similar to using gated linear attention without the MA term. It provides less improvement compared to gated linear attention with ARMA.

**Visualization analysis** Fig. 5 shows test loss curves for different Pure AR/WAVE attention mechanisms on the Weather and ETTm1 datasets, with WAVE consistently outperforming AR in both convergence speed and final loss. Fig. 6 visualizes attention input sequence, AR weights, **B**, and $\Theta$ matrices of a test datapoint on Weather, showing how MA weights decouple local patterns, allowing AR weights to focus on cyclic and long-term patterns. Additional visu-

alizations in Figs. 9–12 reinforce that there are important long-term stable seasonal patterns for AR weights to capture that should not be disrupted by applying forget gates or EMA. This explains why gated linear attention underperforms linear attention in our experiments.

**Computational cost** Table 7 compares the computational cost of pure AR/WAVE Transformers with baselines on the ETTm1 dataset. Our tokenization method reduces the token size $N$, keeping pure AR/WAVE models' computational cost comparable to the baselines. Additionally, parameter sharing ensures the MA term doesn't increase the number of parameters, and the extra FLOPs from using WAVE are not significant.

## 4. Conclusion, limitation, and future works

We propose the WAVE attention mechanism, which integrates an MA term into existing AR attention using a novel indirect MA weight generation method. This approach maintains the same time complexity and parameter size while ensuring the validity of the implicit MA weights. Experiments demonstrate that WAVE attention successfully decouples and handles long-term and short-term effects. The WAVE Transformer, enhanced with the MA term, outperforms their

AR counterparts and achieves state-of-the-art results, offering consistent improvements in training with minimal added computational cost.

One limitation is that we have not explored combining the channel-independent WAVE Transformer with multivariate forecasting models to improve its handling of inter-series relationships. For future work, WAVE attention could be applied to general sequence modeling tasks beyond TSF. Testing on larger-scale datasets, such as using WAVE Transformers for large-scale NLP pretraining, is another promising direction.

## Impact Statement

This paper contributes to the field of Machine Learning by presenting a model that enhances the accuracy and efficiency of time series forecasting. The proposed WAVE approach has valuable applications, including improved decision-making in critical domains like transportation and healthcare. Although the societal impacts of this research are largely positive, it is important to ensure responsible implementation and careful oversight, particularly in sensitive applications, to mitigate any potential risks or negative outcomes.

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

# A. Appendix

## A.1. Related works

**Linear Attention Mechanisms** The quadratic complexity of traditional attention has motivated extensive research into efficient alternatives. Katharopoulos et al. (2020) pioneered linear attention by replacing the exponential similarity function with kernel functions, achieving linear complexity through reordered matrix operations. While this enabled Transformers to be reformulated as RNNs during inference, early implementations suffered performance degradation. Various improvements followed: Choromanski et al. (2021) introduced Performers with FAVOR+ for unbiased softmax approximation; Qin et al. (2022) addressed unbounded gradients and attention dilution in TransNormer; and Sun et al. (2023) combined linear attention with retention mechanisms in RetNet. The connection to classical concepts was explored by Mao (2022), who linked linear transformers to Fast Weight Programmers from the 1990s. Recent advances focus on practical large-scale applications. Yang et al. (2024) proposed Gated Linear Attention with hardware-efficient training, while Lightning Attention-2 achieved constant training speed regardless of sequence length through innovative tiling strategies.

**Linear attention with exponential moving-average** Beyond using a gating decay factor on the hidden state matrix of linear attention (Mao, 2022; Sun et al., 2023; Yang et al., 2024), recent studies have explored incorporating EMA mechanisms into gated linear attention by applying a smoothing factor (summing to 1) to the two terms in the state update (Ma et al., 2022). Similar EMA mechanisms have also been used in many modern RNN structures (Gu et al., 2022; Peng et al., 2023; Orvieto et al., 2023; Qin et al., 2024). Additionally, Schiele et al. (2022) attempted to introduce the ARMA structure into traditional RNNs, but their method could not ensure that the generated MA weights can properly model short-term patterns, and the final results did not significantly surpass traditional RNNs nor compare with recent attention models.

**Time Series Analysis** Time series analysis encompasses several key tasks. Beyond forecasting, tasks include anomaly detection, which involves identifying abnormal points or patterns in data sequences (Chandola et al., 2009; Malhotra et al., 2015; Yang et al., 2025); classification, which assigns time series data to predefined categories or labels (Ismail Fawaz et al., 2019); clustering, grouping similar time series without predefined labels (Liao, 2005; Aghabozorgi et al., 2015); imputation, addressing missing data points to ensure continuity (Che et al., 2018); and change-point detection, pinpointing moments of significant shifts in statistical properties (Truong et al., 2020). Each of these tasks poses distinct challenges and requires tailored methodological approaches.

**Time Series Forecasting** Time series forecasting has evolved from classical methods like ARIMA (Box et al., 1974) and exponential smoothing (Holt, 2004) to deep learning approaches. RNN-based methods (Hochreiter & Schmidhuber, 1997; Rangapuram et al., 2018; Salinas et al., 2020) captured sequential dependencies but struggled with long-range patterns.

**TSF Structures** The use of neural network structures for TSF has been widely explored (Hochreiter & Schmidhuber, 1997; Rangapuram et al., 2018; Salinas et al., 2020; Wu et al., 2022). Recently, many Transformer-based TSF models with encoder-only and encoder-decoder structures have emerged (Li et al., 2019a; Zhou et al., 2021; Wu et al., 2021; Zhang & Yan, 2023; Nie et al., 2022; Liu et al., 2024b). Transformers revolutionized the field through various adaptations: LogTrans (Li et al., 2019b) with local convolutions, Informer (Zhou et al., 2021) with ProbSparse attention, and Autoformer (Wu et al., 2021) with auto-correlation mechanisms. However, these complex Transformer architectures have not significantly outperformed simpler MLP or linear models (Zeng et al., 2023; Das et al., 2023; Xu et al., 2024; Lu et al., 2024). Additionally, these models struggle to handle short-term effects properly with longer lookback windows, where, paradoxically, longer inputs often lead to worse performance. Recent work explores novel perspectives. Nie et al. (2022) proposed PatchTST treating time series as patches, while Liu et al. (2024b) applied attention across variates rather than time. The emergence of LLMs has opened new directions, with Gruver et al. (2023) demonstrating zero-shot forecasting capabilities and Jin et al. (2024) adapting pre-trained models to temporal tasks. Key challenges remain in balancing model complexity with performance and effectively modeling both temporal and cross-variate dependencies (Zhang & Yan, 2023; Lu et al., 2024), while integrating domain-specific biases without sacrificing generality.

## A.2. Datasets

Our main MTSF experiments are conducted on 12 widely-used real-world time series datasets. These datasets are summarized as follows:

**Weather Dataset**[1]**(Wu et al., 2021)**    comprises 21 meteorological variables, including air temperature and humidity, recorded at 10-minute intervals throughout 2020 from the Weather Station of the Max Planck Biogeochemistry Institute in Germany.

**Solar Dataset**[2]**(Lai et al., 2018)**    consists of high-frequency solar power production data from 137 photovoltaic plants recorded throughout 2006. Samples were collected at 10-minute intervals.

**Electricity Dataset**[3]**(Wu et al., 2021)**    contains hourly electricity consumption records for 321 consumers over a three-year period from 2012 to 2014.

**ETT Dataset**[4]**(Zhou et al., 2021)**    The ETT (Electricity Transformer Temperature) Dataset comprises load and oil temperature data from two electricity transformers, recorded at 15-minute and hourly intervals from July 2016 to July 2018. It is divided into four subsets (ETTm1, ETTm2, ETTh1, and ETTh2), each containing seven features related to oil and load characteristics.

**Traffic Dataset**[5]**(Wu et al., 2021)**    Sourced from 862 freeway sensors in the San Francisco Bay area, the Traffic dataset provides hourly road occupancy rates from January 2015 to December 2016. This comprehensive dataset offers consistent measurements across a two-year period.

**PEMS Dataset**[6]**(Li et al., 2017)**    The PEMS dataset consists of public traffic network data collected in California at 5-minute intervals. Our study utilizes four widely-adopted subsets (PEMS03, PEMS04, PEMS07, and PEMS08), which have been extensively studied in the field of spatial-temporal time series analysis for traffic prediction tasks.

### A.3. Hyper-parameter settings and implementation details

For the hyper-parameter settings of the pure AR/WAVE Transformer, we use $m = 3$ Transformer layers, 8 heads, and set the hidden dimension $d$ based on the number of series $C$, using the empirical formula $d = 16\lfloor\sqrt{C}\rfloor$. We use $4d$ as the hidden dimension for the feedforward MLP in the Transformer layer. A dropout rate of 0.1 is applied to both the AR term and MA term. We initialize the weights of all linear layers and embedding layers using the GPT-2 weight initialization method, with a normal distribution and a standard deviation of 0.02. For the output projection layers in the attention and MLP, we additionally scale the standard deviation by a factor of $1/\sqrt{m}$, aligned with the GPT-2 setting. Normalization layer is applied both before the input to the Transformer and after the Transformer output. We experimented with both standard LayerNorm and RMSNorm as the normalization layer, finding no significant performance differences, so we opted for RMSNorm for lower computational cost. For token input projection, we use a linear layer to project the $L_P$-dimensional token to a $d$-dimensional input vector. In the output projection, we do not tie the weights between the input and output linear layers. A learnable position embedding that maps the integer labels from 1 to $N$ (the input sequence length) to the corresponding $d$-dimensional position vectors is used. At the beginning of the model, we apply RevIN to input series $\mathbf{S}_I$, subtracting the mean and dividing by the standard deviation for each series. Before outputting the final result, we multiply by the standard deviation and add the mean back. All input series are processed independently and in parallel, merging different series dimensions into the batch size for parallel computation. The random seed used in all the experiments is 2024.

All training tasks in this paper can be conducted using a single Nvidia RTX 4090 GPU. The batch size is set to 32. For larger datasets, such as Traffic and PEMS07, we use a batch size of 16 or 8, with 2-step or 4-step gradient accumulation to ensure the effective batch size for parameter updates remains 32. During training, pure AR/WAVE Transformers are trained using the next-step prediction objective with MSE loss. We use the AdamW optimizer with betas=(0.9, 0.95) and weight decay=0.1, following the GPT-2 settings. For a fair comparison, the same optimizer is used for training baseline models. It is important to note that the baseline models trained with this AdamW setup show significantly better TSF performance compared to those trained with the default Adam optimizer settings. As a result, the baseline performance presented in this

---

[1] https://www.bgc-jena.mpg.de/wetter/
[2] http://www.nrel.gov/grid/solar-power-data.html
[3] https://archive.ics.uci.edu/ml/datasets/ElectricityLoadDiagrams20112014
[4] https://github.com/zhouhaoyi/ETDataset
[5] http://pems.dot.ca.gov/
[6] http://pems.dot.ca.gov/

paper may exceed the results reported in their original papers. Since this study focuses on long-term last token prediction results, we apply an additional weight factor to the training loss for the last token, multiplying it by $N$. However, this weighting only slightly affects performance on smaller datasets with fewer data points, such as ETTs, and has little to no effect on larger datasets. Given the minimal impact of this method, the original next-token MSE loss is sufficient for most datasets, without requiring further modifications.

We use the same train-validation-test set splitting ratio as in previous studies by Zeng et al. (2023); Nie et al. (2022); Liu et al. (2024b). We also follow the same dataset standardization methods used in these studies. During training, we evaluate the validation and test losses at the end of each epoch, with an early-stopping patience set to 12 epochs. The maximum number of training epochs is 100. We apply a linear warm-up for the learning rate, increasing it from 0.00006 to 0.0006 over the first 5 epochs, and gradually decreasing it in the subsequent epochs.

### A.4. Time Complexity of WAVE Attention

**Proposition A.1.** *Let $N$ be the sequence length and $d$ the embedding dimension. Using an efficient linear-attention implementation, WAVE attention has time complexity*

$$O(N\,d^2),$$

*which is linear in $N$.*

*Proof.* We split WAVE attention into its AR (autoregressive) and MA (moving-average) parts.

**AR Component.**

- **Query, Key, Value projections:** Computing $Q, K, V \in \mathbb{R}^{N \times d}$ via $d \times d$ projections costs $O(N\,d^2)$.

- **Key–Value summary:** At each $t$, update
$$S_t = S_{t-1} + k_t\,v_t^\top,$$
costing $O(d^2)$ per step, for a total of $O(N\,d^2)$.

- **AR output:** Each output
$$o_t^{\mathrm{AR}} = q_t^\top S_{t-1}$$
costs $O(d^2)$, summing to $O(N\,d^2)$.

Thus, AR costs $O(N\,d^2)$.

**MA Component.**

- **Residuals:** For $j = 1, \dots, N-1$, compute
$$r_j = v_{j+1} - o_{j+1}^{\mathrm{AR}},$$
costing $O(N\,d)$ overall.

- **MA projections:** Form $K^{\mathrm{MA}}$ and $Q^{\mathrm{MA}} \in \mathbb{R}^{(N-1)\times d}$ via two $d \times d$ projections, costing $O(N\,d^2)$.

- **Running MA summary:** At each $t$, update
$$T_t = T_{t-1} + \phi_k^{\mathrm{MA}}(k_t^{\mathrm{MA}})\left[\phi_r^{\mathrm{MA}}(r_t)\right]^\top,$$
costing $O(d^2)$ per step, for $O(N\,d^2)$ total.

- **MA output:** Each
$$o_t^{\mathrm{MA}} = \left[\phi_q^{\mathrm{MA}}(q_t^{\mathrm{MA}})\right]^\top T_{t-1}$$
costs $O(d^2)$, summing to $O(N\,d^2)$.

Ignoring the lower-order $O(N\,d)$ term, MA also costs $O(N\,d^2)$.

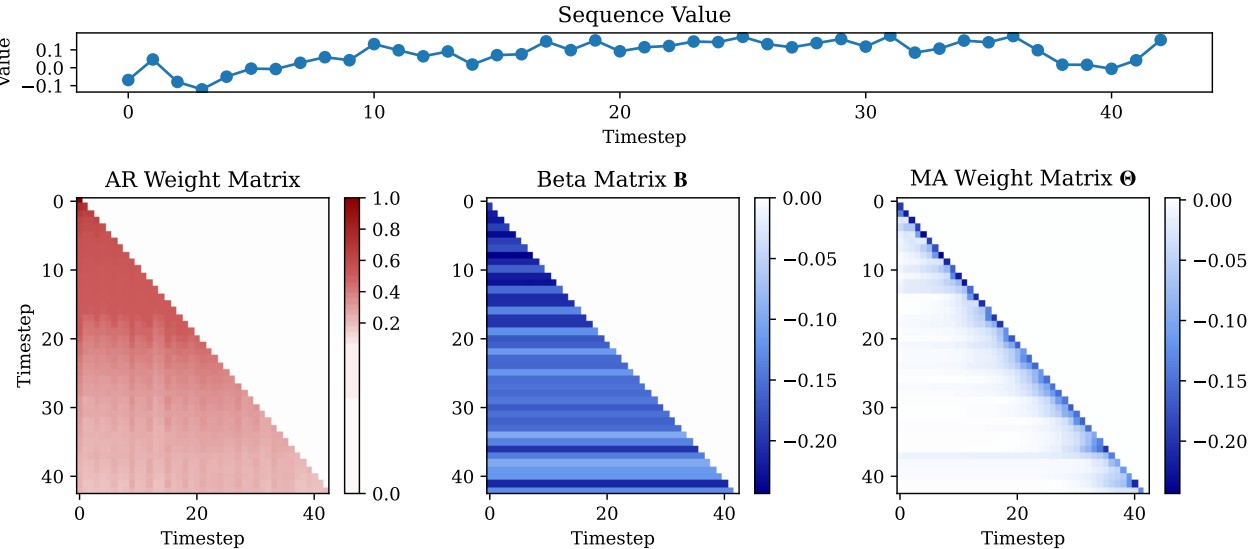

*Figure 6.* Visualization of the WAVE attention weights (first attention layer, averaged across the multiple heads or $d$-dimensional channels) for the first test set data point in the Weather dataset ($L_I = 4096$, $L_P = 96$). More weight visualization can be found in Fig. 9, 10, 11, and 12.

**Conclusion.** Combining AR and MA yields

$$O(N d^2) + O(N d^2) = O(N d^2),$$

i.e. linear in $N$ for fixed $d$. ☐

### A.5. Supplementary experiment results

In the following section, we provide the complete experimental data corresponding to the tables in the main text. Additionally, we include extra visualizations to help illustrate the actual behavior of the MA weights.

*Table 7.* Comparison of computational costs utilizing the data format of ETTm1 to build model inputs ($L_I = 512$). The hyper-parameters for models are set according to their default configurations.

| Models | EncFormer | | CATS | | PatchTST | | iTransformer | | DLinear | | FITS | |
|---|---|---|---|---|---|---|---|---|---|---|---|---|
| Metric | FLOPs | Params | FLOPs | Params | FLOPs | Params | FLOPs | Params | FLOPs | Params | FLOPs | Params |
| $L_P = 96$ | 1.442G | 1.646M | 262.9M | 1.326M | 180.9M | 1.046M | 81.96M | 1.857M | 4.337M | 98.50K | 334.0K | 24.02K |
| $L_P = 48$ | 1.328G | 1.646M | 243.5M | 1.227M | 163.6M | 652.9K | 81.69M | 1.851M | 2.174M | 49.25K | 308.1K | 22.16K |
| $L_P = 24$ | 1.271G | 1.645M | 233.9M | 1.178M | 155.0M | 456.3K | 81.56M | 1.848M | 1.093M | 24.62K | 294.2K | 21.16K |
| $L_P = 12$ | 1.242G | 1.645M | 229.0M | 1.154M | 150.7M | 358.0K | 81.49M | 1.847M | 552.5K | 12.31K | 288.3K | 20.74K |

| Model | GLin Attn | | GLin Attn +ARMA | | Lin Attn | | Lin Attn +ARMA | | ELin Attn | | ELin Attn +ARMA | |
|---|---|---|---|---|---|---|---|---|---|---|---|---|
| Metric | FLOPs | Params | FLOPs | Params | FLOPs | Params | FLOPs | Params | FLOPs | Params | FLOPs | Params |
| $L_P = 96$ | 7.403M | 45.81K | 7.431M | 45.81K | 7.387M | 45.79K | 7.415M | 45.79K | 7.258M | 45.79K | 7.266M | 45.79K |
| $L_P = 48$ | 12.63M | 43.97K | 12.77M | 43.97K | 12.60M | 43.95K | 12.74M | 43.95K | 12.36M | 43.95K | 12.37M | 43.95K |
| $L_P = 24$ | 24.30M | 45.22K | 24.70M | 45.22K | 24.25M | 45.21K | 24.64M | 45.21K | 23.77M | 45.21K | 23.80M | 45.21K |
| $L_P = 12$ | 46.58M | 49.82K | 47.45M | 49.82K | 46.46M | 49.80K | 47.34M | 49.80K | 45.54M | 49.80K | 45.60M | 49.80K |

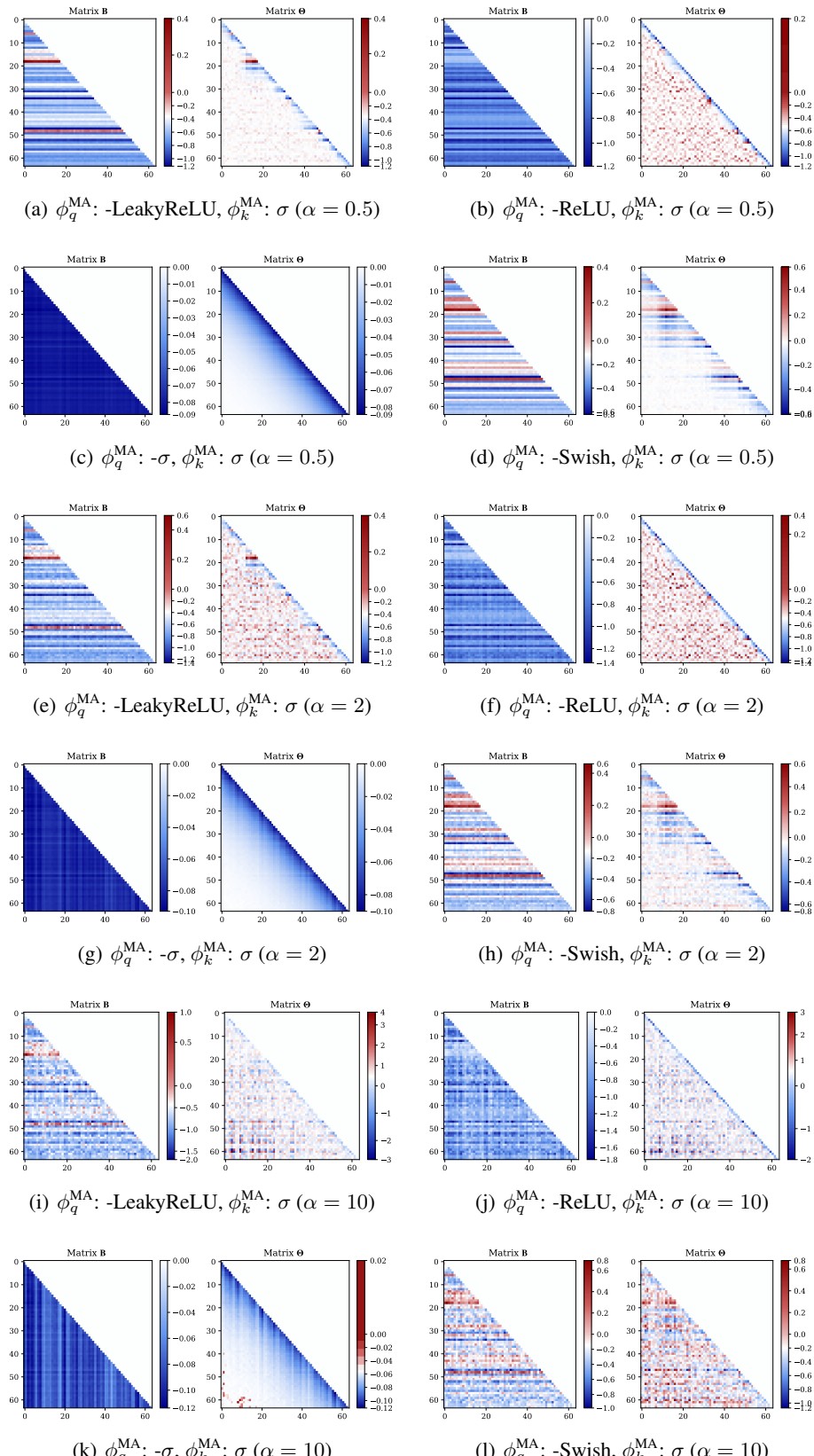

*Figure 7.* Additional visualization of $\mathbf{B} - \mathbf{\Theta}$ relationship with different $\phi(\cdot)$ and different $\alpha$. We construct the simulated $\mathbf{B}$ matrices using randomly sampled $\boldsymbol{q}$ and $\boldsymbol{k}$ ($N = 64$, $d = 32$) from the normal distribution, and display the corresponding $\mathbf{\Theta}$ matrices.

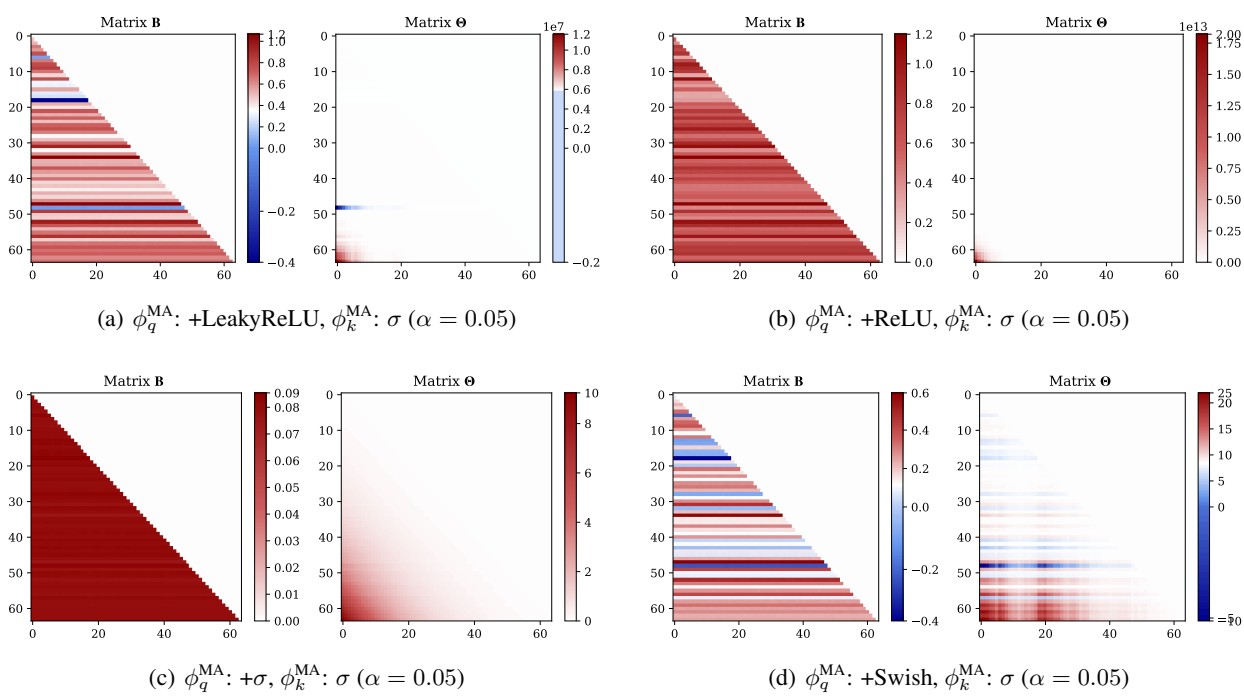

*Figure 8.* Visualization of $\mathbf{B} - \mathbf{\Theta}$ relationship with positive query activation functions $\phi_q^{\text{MA}}(\cdot)$.

*Table 8.* Detailed results of main TSF experiments with forecasting horizons $L_P \in \{12, 24, 48, 96\}$ and $L_I = 512$. Test set MSE and MAE for each model on each experiment setup are presented.

| Dataset | Metrics | Std Attn MSE | Std Attn MAE | Std Attn+ARMA MSE | Std Attn+ARMA MAE | Lin Attn MSE | Lin Attn MAE | Lin Attn+ARMA MSE | Lin Attn+ARMA MAE | GLin Attn MSE | GLin Attn MAE | GLin Attn+ARMA MSE | GLin Attn+ARMA MAE | ELin Attn MSE | ELin Attn MAE | ELin Attn+ARMA MSE | ELin Attn+ARMA MAE | Fixed Attn MSE | Fixed Attn MAE | Fixed Attn+ARMA MSE | Fixed Attn+ARMA MAE | FITS MSE | FITS MAE | iTransformer MSE | iTransformer MAE | CATS MSE | CATS MAE | PatchTST MSE | PatchTST MAE | DLinear MSE | DLinear MAE | EncFormer MSE | EncFormer MAE |
|---|---|---|---|---|---|---|---|---|---|---|---|---|---|---|---|---|---|---|---|---|---|---|---|---|---|---|---|---|---|---|---|---|---|
| Weather | 96 | 0.144 | 0.195 | 0.142 | 0.193 | 0.142 | 0.194 | 0.139 | 0.191 | 0.161 | 0.210 | 0.142 | 0.194 | 0.146 | 0.197 | 0.143 | 0.195 | 0.147 | 0.194 | 0.142 | 0.198 | 0.151 | 0.204 | 0.158 | 0.204 | 0.146 | 0.198 | 0.149 | 0.224 | 0.150 | 0.209 | 0.188 | 0.248 |
| | 48 | 0.113 | 0.157 | 0.109 | 0.151 | 0.115 | 0.158 | 0.110 | 0.153 | 0.144 | 0.191 | 0.116 | 0.159 | 0.114 | 0.158 | 0.112 | 0.156 | 0.112 | 0.155 | 0.115 | 0.159 | 0.125 | 0.177 | 0.132 | 0.176 | 0.116 | 0.160 | 0.118 | 0.161 | 0.122 | 0.177 | 0.143 | 0.199 |
| | 24 | 0.089 | 0.118 | 0.085 | 0.115 | 0.088 | 0.117 | 0.083 | 0.114 | 0.101 | 0.129 | 0.090 | 0.122 | 0.087 | 0.116 | 0.089 | 0.119 | 0.091 | 0.122 | 0.088 | 0.120 | 0.103 | 0.147 | 0.101 | 0.139 | 0.090 | 0.125 | 0.092 | 0.122 | 0.147 | 0.102 | 0.117 | 0.162 |
| | 12 | 0.071 | 0.091 | 0.067 | 0.086 | 0.069 | 0.088 | 0.067 | 0.086 | 0.070 | 0.090 | 0.070 | 0.091 | 0.069 | 0.087 | 0.069 | 0.087 | 0.071 | 0.089 | 0.069 | 0.090 | 0.078 | 0.112 | 0.078 | 0.105 | 0.069 | 0.092 | 0.070 | 0.097 | 0.078 | 0.115 | 0.093 | 0.120 |
| Solar | 96 | 0.196 | 0.263 | 0.192 | 0.257 | 0.183 | 0.247 | 0.180 | 0.244 | 0.209 | 0.283 | 0.182 | 0.243 | 0.194 | 0.258 | 0.191 | 0.257 | 0.195 | 0.261 | 0.187 | 0.256 | 0.210 | 0.254 | 0.230 | 0.257 | 0.182 | 0.239 | 0.209 | 0.251 | 0.208 | 0.274 | 0.201 | 0.225 |
| | 48 | 0.160 | 0.230 | 0.151 | 0.223 | 0.152 | 0.219 | 0.149 | 0.217 | 0.177 | 0.258 | 0.154 | 0.222 | 0.162 | 0.236 | 0.159 | 0.233 | 0.161 | 0.233 | 0.159 | 0.229 | 0.188 | 0.245 | 0.189 | 0.220 | 0.157 | 0.209 | 0.186 | 0.240 | 0.184 | 0.255 | 0.157 | 0.186 |
| | 24 | 0.112 | 0.180 | 0.098 | 0.168 | 0.098 | 0.166 | 0.095 | 0.162 | 0.143 | 0.232 | 0.099 | 0.167 | 0.115 | 0.192 | 0.113 | 0.188 | 0.131 | 0.221 | 0.122 | 0.203 | 0.132 | 0.203 | 0.108 | 0.163 | 0.161 | 0.161 | 0.129 | 0.208 | 0.128 | 0.208 | 0.092 | 0.140 |
| | 12 | 0.069 | 0.137 | 0.055 | 0.118 | 0.056 | 0.113 | 0.052 | 0.111 | 0.063 | 0.139 | 0.059 | 0.121 | 0.072 | 0.135 | 0.069 | 0.133 | 0.081 | 0.161 | 0.070 | 0.141 | 0.078 | 0.159 | 0.053 | 0.104 | 0.052 | 0.113 | 0.075 | 0.155 | 0.075 | 0.161 | 0.048 | 0.101 |
| ECL | 96 | 0.136 | 0.233 | 0.132 | 0.229 | 0.130 | 0.226 | 0.128 | 0.225 | 0.135 | 0.231 | 0.133 | 0.229 | 0.139 | 0.237 | 0.138 | 0.235 | 0.142 | 0.239 | 0.132 | 0.236 | 0.144 | 0.246 | 0.132 | 0.226 | 0.131 | 0.229 | 0.132 | 0.224 | 0.135 | 0.232 | 0.227 | 0.342 |
| | 48 | 0.117 | 0.214 | 0.113 | 0.211 | 0.110 | 0.207 | 0.110 | 0.206 | 0.113 | 0.210 | 0.113 | 0.210 | 0.124 | 0.221 | 0.121 | 0.219 | 0.125 | 0.223 | 0.121 | 0.220 | 0.129 | 0.233 | 0.111 | 0.207 | 0.115 | 0.214 | 0.112 | 0.206 | 0.120 | 0.219 | 0.198 | 0.317 |
| | 24 | 0.097 | 0.193 | 0.094 | 0.190 | 0.092 | 0.188 | 0.091 | 0.189 | 0.099 | 0.197 | 0.096 | 0.193 | 0.103 | 0.202 | 0.100 | 0.199 | 0.106 | 0.205 | 0.104 | 0.202 | 0.115 | 0.221 | 0.095 | 0.191 | 0.101 | 0.199 | 0.103 | 0.196 | 0.106 | 0.206 | 0.183 | 0.302 |
| | 12 | 0.089 | 0.188 | 0.085 | 0.184 | 0.091 | 0.191 | 0.088 | 0.189 | 0.090 | 0.190 | 0.088 | 0.188 | 0.093 | 0.197 | 0.096 | 0.201 | 0.110 | 0.218 | 0.107 | 0.215 | 0.106 | 0.214 | 0.084 | 0.186 | 0.091 | 0.192 | 0.098 | 0.199 | 0.096 | 0.196 | 0.194 | 0.313 |
| ETTh1 | 96 | 0.357 | 0.393 | 0.360 | 0.395 | 0.358 | 0.396 | 0.361 | 0.399 | 0.378 | 0.406 | 0.368 | 0.404 | 0.360 | 0.393 | 0.356 | 0.393 | 0.362 | 0.396 | 0.359 | 0.394 | 0.369 | 0.398 | 0.396 | 0.422 | 0.371 | 0.398 | 0.386 | 0.407 | 0.370 | 0.394 | 0.986 | 0.720 |
| | 48 | 0.334 | 0.375 | 0.331 | 0.374 | 0.331 | 0.375 | 0.331 | 0.376 | 0.349 | 0.386 | 0.337 | 0.381 | 0.330 | 0.372 | 0.331 | 0.375 | 0.331 | 0.373 | 0.334 | 0.378 | 0.343 | 0.379 | 0.362 | 0.396 | 0.341 | 0.379 | 0.352 | 0.389 | 0.342 | 0.379 | 0.884 | 0.670 |
| | 24 | 0.312 | 0.365 | 0.299 | 0.357 | 0.299 | 0.356 | 0.299 | 0.357 | 0.349 | 0.394 | 0.303 | 0.360 | 0.305 | 0.360 | 0.305 | 0.360 | 0.306 | 0.361 | 0.304 | 0.360 | 0.318 | 0.366 | 0.334 | 0.381 | 0.313 | 0.365 | 0.313 | 0.361 | 0.312 | 0.362 | 0.792 | 0.676 |
| | 12 | 0.290 | 0.345 | 0.280 | 0.340 | 0.285 | 0.342 | 0.272 | 0.337 | 0.554 | 0.503 | 0.277 | 0.340 | 0.296 | 0.351 | 0.293 | 0.348 | 0.320 | 0.374 | 0.316 | 0.368 | 0.301 | 0.357 | 0.310 | 0.363 | 0.281 | 0.341 | 0.290 | 0.345 | 0.291 | 0.348 | 0.607 | 0.534 |
| ETTh2 | 96 | 0.266 | 0.330 | 0.268 | 0.331 | 0.273 | 0.336 | 0.275 | 0.338 | 0.285 | 0.338 | 0.281 | 0.345 | 0.267 | 0.333 | 0.263 | 0.329 | 0.288 | 0.345 | 0.276 | 0.337 | 0.270 | 0.336 | 0.373 | 0.394 | 0.270 | 0.338 | 0.274 | 0.341 | 0.277 | 0.346 | 1.303 | 0.924 |
| | 48 | 0.213 | 0.290 | 0.216 | 0.294 | 0.215 | 0.290 | 0.217 | 0.293 | 0.233 | 0.294 | 0.220 | 0.295 | 0.212 | 0.290 | 0.210 | 0.288 | 0.215 | 0.290 | 0.208 | 0.288 | 0.217 | 0.298 | 0.226 | 0.310 | 0.212 | 0.299 | 0.222 | 0.304 | 0.217 | 0.301 | 0.568 | 0.596 |
| | 24 | 0.160 | 0.252 | 0.160 | 0.252 | 0.159 | 0.250 | 0.162 | 0.252 | 0.182 | 0.263 | 0.164 | 0.254 | 0.162 | 0.249 | 0.158 | 0.249 | 0.158 | 0.250 | 0.158 | 0.251 | 0.168 | 0.264 | 0.178 | 0.275 | 0.167 | 0.262 | 0.174 | 0.269 | 0.166 | 0.263 | 0.290 | 0.405 |
| | 12 | 0.129 | 0.229 | 0.125 | 0.224 | 0.124 | 0.224 | 0.125 | 0.224 | 0.168 | 0.263 | 0.127 | 0.224 | 0.129 | 0.228 | 0.128 | 0.230 | 0.139 | 0.240 | 0.133 | 0.235 | 0.133 | 0.239 | 0.139 | 0.248 | 0.128 | 0.235 | 0.135 | 0.242 | 0.131 | 0.237 | 0.225 | 0.354 |
| ETTm1 | 96 | 0.305 | 0.359 | 0.301 | 0.354 | 0.303 | 0.355 | 0.296 | 0.351 | 0.336 | 0.382 | 0.299 | 0.355 | 0.305 | 0.356 | 0.301 | 0.354 | 0.298 | 0.347 | 0.296 | 0.344 | 0.305 | 0.347 | 0.352 | 0.378 | 0.300 | 0.354 | 0.323 | 0.362 | 0.305 | 0.348 | 0.686 | 0.603 |
| | 48 | 0.287 | 0.344 | 0.276 | 0.333 | 0.278 | 0.336 | 0.266 | 0.328 | 0.500 | 0.472 | 0.372 | 0.334 | 0.284 | 0.346 | 0.282 | 0.342 | 0.284 | 0.338 | 0.280 | 0.340 | 0.280 | 0.331 | 0.304 | 0.346 | 0.266 | 0.328 | 0.289 | 0.341 | 0.278 | 0.329 | 0.475 | 0.474 |
| | 24 | 0.246 | 0.312 | 0.223 | 0.293 | 0.218 | 0.293 | 0.196 | 0.279 | 0.473 | 0.429 | 0.225 | 0.305 | 0.239 | 0.305 | 0.241 | 0.307 | 0.284 | 0.332 | 0.255 | 0.321 | 0.218 | 0.289 | 0.223 | 0.293 | 0.197 | 0.277 | 0.235 | 0.304 | 0.216 | 0.288 | 0.352 | 0.391 |
| | 12 | 0.218 | 0.287 | 0.156 | 0.241 | 0.151 | 0.240 | 0.128 | 0.222 | 0.320 | 0.335 | 0.144 | 0.237 | 0.157 | 0.247 | 0.153 | 0.246 | 0.203 | 0.282 | 0.174 | 0.261 | 0.144 | 0.235 | 0.155 | 0.246 | 0.125 | 0.222 | 0.128 | 0.224 | 0.140 | 0.230 | 0.204 | 0.294 |
| ETTm2 | 96 | 0.177 | 0.262 | 0.174 | 0.261 | 0.167 | 0.255 | 0.162 | 0.250 | 0.178 | 0.260 | 0.172 | 0.258 | 0.178 | 0.260 | 0.174 | 0.260 | 0.167 | 0.254 | 0.166 | 0.254 | 0.164 | 0.253 | 0.198 | 0.277 | 0.168 | 0.266 | 0.168 | 0.266 | 0.184 | 0.283 | 0.481 | 0.525 |
| | 48 | 0.139 | 0.238 | 0.137 | 0.236 | 0.145 | 0.248 | 0.143 | 0.250 | 0.167 | 0.264 | 0.148 | 0.249 | 0.153 | 0.253 | 0.139 | 0.238 | 0.146 | 0.245 | 0.138 | 0.238 | 0.126 | 0.225 | 0.147 | 0.242 | 0.127 | 0.231 | 0.131 | 0.231 | 0.125 | 0.226 | 0.483 | 0.516 |
| | 24 | 0.121 | 0.219 | 0.117 | 0.217 | 0.110 | 0.211 | 0.101 | 0.199 | 0.132 | 0.227 | 0.110 | 0.209 | 0.117 | 0.217 | 0.116 | 0.216 | 0.119 | 0.220 | 0.119 | 0.221 | 0.096 | 0.195 | 0.112 | 0.212 | 0.096 | 0.195 | 0.097 | 0.194 | 0.095 | 0.194 | 0.162 | 0.280 |
| | 12 | 0.086 | 0.175 | 0.083 | 0.174 | 0.083 | 0.174 | 0.078 | 0.169 | 0.089 | 0.178 | 0.082 | 0.172 | 0.086 | 0.174 | 0.083 | 0.174 | 0.085 | 0.177 | 0.083 | 0.174 | 0.073 | 0.168 | 0.082 | 0.181 | 0.072 | 0.165 | 0.072 | 0.165 | 0.077 | 0.195 | 0.116 | 0.231 |
| Traffic | 96 | 0.379 | 0.273 | 0.373 | 0.269 | 0.365 | 0.262 | 0.362 | 0.260 | 0.474 | 0.324 | 0.381 | 0.275 | 0.381 | 0.270 | 0.378 | 0.271 | 0.393 | 0.279 | 0.389 | 0.277 | 0.404 | 0.286 | 0.356 | 0.259 | 0.377 | 0.284 | 0.381 | 0.298 | 0.399 | 0.286 | 0.915 | 0.460 |
| | 48 | 0.352 | 0.256 | 0.342 | 0.253 | 0.349 | 0.251 | 0.339 | 0.254 | 0.569 | 0.371 | 0.375 | 0.271 | 0.363 | 0.263 | 0.359 | 0.261 | 0.374 | 0.275 | 0.365 | 0.265 | 0.393 | 0.280 | 0.341 | 0.264 | 0.369 | 0.271 | 0.373 | 0.341 | 0.385 | 0.284 | 0.857 | 0.434 |
| | 24 | 0.324 | 0.238 | 0.315 | 0.231 | 0.322 | 0.239 | 0.318 | 0.238 | 0.346 | 0.257 | 0.330 | 0.245 | 0.334 | 0.247 | 0.330 | 0.243 | 0.344 | 0.256 | 0.340 | 0.251 | 0.374 | 0.280 | 0.318 | 0.250 | 0.372 | 0.279 | 0.345 | 0.262 | 0.363 | 0.271 | 0.814 | 0.417 |
| | 12 | 0.310 | 0.230 | 0.303 | 0.228 | 0.311 | 0.232 | 0.302 | 0.227 | 0.325 | 0.250 | 0.313 | 0.236 | 0.330 | 0.255 | 0.326 | 0.252 | 0.379 | 0.285 | 0.364 | 0.281 | 0.370 | 0.282 | 0.303 | 0.238 | 0.369 | 0.268 | 0.331 | 0.253 | 0.353 | 0.270 | 0.801 | 0.403 |
| PEMS03 | 96 | 0.171 | 0.280 | 0.153 | 0.263 | 0.149 | 0.258 | 0.143 | 0.252 | 0.360 | 0.416 | 0.147 | 0.257 | 0.173 | 0.281 | 0.169 | 0.279 | 0.178 | 0.285 | 0.174 | 0.280 | 0.193 | 0.274 | 0.135 | 0.229 | 0.157 | 0.267 | 0.198 | 0.285 | 0.198 | 0.299 | 0.162 | 0.272 |
| | 48 | 0.122 | 0.246 | 0.106 | 0.230 | 0.105 | 0.248 | 0.102 | 0.250 | 0.299 | 0.380 | 0.109 | 0.216 | 0.128 | 0.235 | 0.123 | 0.248 | 0.131 | 0.243 | 0.126 | 0.237 | 0.155 | 0.247 | 0.108 | 0.215 | 0.116 | 0.230 | 0.167 | 0.266 | 0.155 | 0.260 | 0.117 | 0.234 |
| | 24 | 0.089 | 0.215 | 0.079 | 0.199 | 0.081 | 0.211 | 0.075 | 0.199 | 0.097 | 0.209 | 0.082 | 0.189 | 0.091 | 0.199 | 0.089 | 0.201 | 0.099 | 0.213 | 0.095 | 0.209 | 0.108 | 0.211 | 0.078 | 0.184 | 0.085 | 0.196 | 0.108 | 0.221 | 0.109 | 0.218 | 0.089 | 0.201 |
| | 12 | 0.067 | 0.190 | 0.063 | 0.179 | 0.065 | 0.174 | 0.062 | 0.165 | 0.072 | 0.185 | 0.067 | 0.177 | 0.071 | 0.181 | 0.068 | 0.176 | 0.073 | 0.187 | 0.070 | 0.176 | 0.076 | 0.178 | 0.062 | 0.166 | 0.061 | 0.166 | 0.088 | 0.184 | 0.075 | 0.184 | 0.075 | 0.186 |
| PEMS04 | 96 | 0.126 | 0.237 | 0.139 | 0.227 | 0.113 | 0.240 | 0.117 | 0.225 | 0.137 | 0.277 | 0.118 | 0.228 | 0.138 | 0.245 | 0.133 | 0.245 | 0.256 | 0.256 | 0.139 | 0.254 | 0.193 | 0.274 | 0.119 | 0.210 | 0.144 | 0.257 | 0.132 | 0.245 | 0.178 | 0.278 | 0.145 | 0.258 |
| | 48 | 0.091 | 0.201 | 0.082 | 0.189 | 0.084 | 0.223 | 0.071 | 0.179 | 0.093 | 0.204 | 0.085 | 0.199 | 0.109 | 0.222 | 0.106 | 0.219 | 0.114 | 0.227 | 0.109 | 0.224 | 0.152 | 0.254 | 0.072 | 0.171 | 0.102 | 0.212 | 0.091 | 0.206 | 0.155 | 0.257 | 0.104 | 0.216 |
| | 24 | 0.080 | 0.189 | 0.068 | 0.171 | 0.079 | 0.199 | 0.065 | 0.169 | 0.080 | 0.188 | 0.077 | 0.186 | 0.086 | 0.199 | 0.081 | 0.190 | 0.099 | 0.206 | 0.086 | 0.196 | 0.106 | 0.213 | 0.068 | 0.156 | 0.075 | 0.181 | 0.082 | 0.193 | 0.107 | 0.218 | 0.086 | 0.196 |
| | 12 | 0.072 | 0.180 | 0.063 | 0.168 | 0.072 | 0.180 | 0.055 | 0.152 | 0.068 | 0.184 | 0.068 | 0.181 | 0.071 | 0.181 | 0.069 | 0.178 | 0.073 | 0.184 | 0.067 | 0.174 | 0.075 | 0.178 | 0.055 | 0.150 | 0.056 | 0.154 | 0.066 | 0.162 | 0.074 | 0.179 | 0.073 | 0.180 |
| PEMS07 | 96 | 0.154 | 0.265 | 0.139 | 0.250 | 0.133 | 0.240 | 0.131 | 0.237 | 0.166 | 0.277 | 0.135 | 0.243 | 0.163 | 0.275 | 0.158 | 0.270 | 0.171 | 0.284 | 0.168 | 0.280 | 0.215 | 0.299 | 0.121 | 0.217 | 0.141 | 0.253 | 0.221 | 0.320 | 0.209 | 0.301 | 0.116 | 0.232 |
| | 48 | 0.133 | 0.246 | 0.119 | 0.230 | 0.114 | 0.223 | 0.106 | 0.217 | 0.299 | 0.389 | 0.113 | 0.225 | 0.138 | 0.235 | 0.135 | 0.248 | 0.131 | 0.253 | 0.135 | 0.251 | 0.173 | 0.270 | 0.109 | 0.205 | 0.126 | 0.247 | 0.214 | 0.340 | 0.167 | 0.268 | 0.103 | 0.216 |
| | 24 | 0.103 | 0.215 | 0.092 | 0.199 | 0.091 | 0.199 | 0.085 | 0.194 | 0.116 | 0.230 | 0.094 | 0.202 | 0.103 | 0.217 | 0.103 | 0.215 | 0.113 | 0.227 | 0.107 | 0.218 | 0.124 | 0.230 | 0.087 | 0.186 | 0.090 | 0.201 | 0.131 | 0.243 | 0.123 | 0.229 | 0.092 | 0.203 |
| | 12 | 0.082 | 0.190 | 0.075 | 0.179 | 0.075 | 0.180 | 0.070 | 0.176 | 0.087 | 0.195 | 0.078 | 0.183 | 0.085 | 0.191 | 0.081 | 0.188 | 0.089 | 0.198 | 0.085 | 0.193 | 0.092 | 0.199 | 0.073 | 0.174 | 0.074 | 0.180 | 0.089 | 0.196 | 0.091 | 0.197 | 0.085 | 0.193 |
| PEMS08 | 96 | 0.222 | 0.262 | 0.207 | 0.249 | 0.179 | 0.235 | 0.175 | 0.233 | 0.255 | 0.291 | 0.194 | 0.238 | 0.229 | 0.271 | 0.221 | 0.268 | 0.234 | 0.273 | 0.226 | 0.269 | 0.337 | 0.322 | 0.170 | 0.215 | 0.212 | 0.276 | 0.236 | 0.236 | 0.318 | 0.326 | 0.253 | 0.302 |
| | 48 | 0.166 | 0.241 | 0.141 | 0.222 | 0.128 | 0.215 | 0.125 | 0.208 | 0.191 | 0.269 | 0.138 | 0.228 | 0.172 | 0.269 | 0.164 | 0.247 | 0.188 | 0.263 | 0.164 | 0.248 | 0.232 | 0.282 | 0.131 | 0.197 | 0.142 | 0.238 | 0.136 | 0.225 | 0.223 | 0.283 | 0.203 | 0.279 |
| | 24 | 0.114 | 0.209 | 0.102 | 0.192 | 0.095 | 0.191 | 0.086 | 0.184 | 0.112 | 0.212 | 0.098 | 0.199 | 0.112 | 0.213 | 0.109 | 0.211 | 0.118 | 0.221 | 0.114 | 0.214 | 0.142 | 0.233 | 0.090 | 0.183 | 0.107 | 0.203 | 0.091 | 0.199 | 0.137 | 0.232 | 0.154 | 0.238 |
| | 12 | 0.088 | 0.189 | 0.077 | 0.173 | 0.074 | 0.171 | 0.076 | 0.175 | 0.079 | 0.182 | 0.071 | 0.170 | 0.086 | 0.185 | 0.082 | 0.186 | 0.103 | 0.225 | 0.105 | 0.221 | 0.094 | 0.196 | 0.076 | 0.171 | 0.078 | 0.177 | 0.073 | 0.176 | 0.092 | 0.195 | 0.123 | 0.216 |

*Table 9.* Results showing that WAVE Transformers with $m = 3$ layers consistently outperform their AR counterparts across a wide range of $m$. Forecasting horizons $L_P \in \{12, 24, 48, 96\}$ and $L_I = 512$ are used. Test set MSE and MAE for each model on each experiment setup are presented.

| | | Model | WAVE(m=3) | | Pure AR(m=1) | | Pure AR(m=2) | | Pure AR(m=3) | | Pure AR(m=4) | | Pure AR(m=5) | | Pure AR(m=6) | | Pure AR(m=7) | | Pure AR(m=8) | |
|---|---|---|---|---|---|---|---|---|---|---|---|---|---|---|---|---|---|---|---|---|
| | | Metrics | MSE | MAE | MSE | MAE | MSE | MAE | MSE | MAE | MSE | MAE | MSE | MAE | MSE | MAE | MSE | MAE | MSE | MAE |
| ETTm1 | 96 | Std Attn | 0.301 | 0.354 | 0.305 | 0.360 | 0.308 | 0.360 | 0.305 | 0.359 | 0.308 | 0.360 | 0.304 | 0.358 | 0.309 | 0.361 | 0.306 | 0.359 | 0.307 | 0.359 |
| | | Lin Attn | 0.296 | 0.351 | 0.310 | 0.361 | 0.301 | 0.358 | 0.303 | 0.355 | 0.303 | 0.356 | 0.299 | 0.352 | 0.301 | 0.355 | 0.299 | 0.353 | 0.300 | 0.354 |
| | | GLin Attn | 0.299 | 0.355 | 0.337 | 0.381 | 0.337 | 0.387 | 0.336 | 0.382 | 0.334 | 0.380 | 0.337 | 0.382 | 0.337 | 0.382 | 0.335 | 0.381 | 0.333 | 0.379 |
| | | ELin Attn | 0.301 | 0.354 | 0.307 | 0.363 | 0.309 | 0.361 | 0.305 | 0.356 | 0.307 | 0.360 | 0.306 | 0.359 | 0.309 | 0.362 | 0.305 | 0.359 | 0.308 | 0.361 |
| | | Fixed Attn | 0.296 | 0.344 | 0.299 | 0.346 | 0.298 | 0.349 | 0.298 | 0.347 | 0.299 | 0.347 | 0.300 | 0.348 | 0.298 | 0.347 | 0.302 | 0.348 | 0.305 | 0.351 |
| | 48 | Std Attn | 0.276 | 0.333 | 0.293 | 0.347 | 0.293 | 0.347 | 0.287 | 0.344 | 0.290 | 0.345 | 0.290 | 0.345 | 0.288 | 0.344 | 0.286 | 0.342 | 0.291 | 0.345 |
| | | Lin Attn | 0.266 | 0.328 | 0.280 | 0.336 | 0.278 | 0.337 | 0.278 | 0.336 | 0.278 | 0.336 | 0.271 | 0.331 | 0.272 | 0.333 | 0.274 | 0.332 | 0.276 | 0.334 |
| | | GLin Attn | 0.372 | 0.334 | 0.494 | 0.347 | 0.496 | 0.464 | 0.500 | 0.472 | 0.505 | 0.467 | 0.500 | 0.468 | 0.516 | 0.459 | 0.494 | 0.469 | 0.499 | 0.469 |
| | | ELin Attn | 0.282 | 0.342 | 0.293 | 0.350 | 0.289 | 0.350 | 0.284 | 0.346 | 0.292 | 0.349 | 0.294 | 0.352 | 0.295 | 0.352 | 0.292 | 0.350 | 0.299 | 0.355 |
| | | Fixed Attn | 0.280 | 0.340 | 0.286 | 0.345 | 0.283 | 0.340 | 0.284 | 0.338 | 0.283 | 0.340 | 0.284 | 0.338 | 0.277 | 0.335 | 0.282 | 0.335 | 0.279 | 0.336 |
| | 24 | Std Attn | 0.223 | 0.293 | 0.234 | 0.300 | 0.258 | 0.323 | 0.246 | 0.312 | 0.244 | 0.311 | 0.262 | 0.325 | 0.260 | 0.324 | 0.259 | 0.323 | 0.263 | 0.327 |
| | | Lin Attn | 0.196 | 0.279 | 0.226 | 0.297 | 0.210 | 0.288 | 0.218 | 0.293 | 0.211 | 0.288 | 0.210 | 0.286 | 0.208 | 0.285 | 0.212 | 0.287 | 0.209 | 0.287 |
| | | GLin Attn | 0.225 | 0.305 | 0.487 | 0.430 | 0.499 | 0.440 | 0.473 | 0.429 | 0.476 | 0.436 | 0.482 | 0.436 | 0.466 | 0.435 | 0.486 | 0.440 | 0.463 | 0.430 |
| | | ELin Attn | 0.241 | 0.307 | 0.253 | 0.328 | 0.246 | 0.317 | 0.239 | 0.305 | 0.253 | 0.318 | 0.268 | 0.327 | 0.263 | 0.322 | 0.264 | 0.325 | 0.264 | 0.326 |
| | | Fixed Attn | 0.255 | 0.321 | 0.274 | 0.317 | 0.272 | 0.334 | 0.284 | 0.332 | 0.260 | 0.326 | 0.254 | 0.320 | 0.266 | 0.329 | 0.257 | 0.322 | 0.255 | 0.322 |
| | 12 | Std Attn | 0.156 | 0.241 | 0.229 | 0.293 | 0.222 | 0.288 | 0.218 | 0.287 | 0.221 | 0.291 | 0.221 | 0.289 | 0.223 | 0.291 | 0.228 | 0.288 | 0.227 | 0.287 |
| | | Lin Attn | 0.128 | 0.222 | 0.148 | 0.239 | 0.141 | 0.232 | 0.151 | 0.240 | 0.137 | 0.228 | 0.138 | 0.234 | 0.139 | 0.231 | 0.137 | 0.228 | 0.138 | 0.229 |
| | | GLin Attn | 0.144 | 0.237 | 0.325 | 0.325 | 0.321 | 0.324 | 0.320 | 0.335 | 0.319 | 0.326 | 0.322 | 0.326 | 0.322 | 0.325 | 0.320 | 0.323 | 0.320 | 0.323 |
| | | ELin Attn | 0.153 | 0.246 | 0.160 | 0.251 | 0.160 | 0.252 | 0.157 | 0.247 | 0.159 | 0.250 | 0.160 | 0.252 | 0.169 | 0.260 | 0.163 | 0.251 | 0.160 | 0.247 |
| | | Fixed Attn | 0.174 | 0.261 | 0.215 | 0.289 | 0.204 | 0.285 | 0.203 | 0.282 | 0.199 | 0.281 | 0.194 | 0.279 | 0.195 | 0.278 | 0.192 | 0.275 | 0.189 | 0.278 |
| Weather | 96 | Std Attn | 0.142 | 0.193 | 0.156 | 0.207 | 0.153 | 0.210 | 0.144 | 0.195 | 0.152 | 0.206 | 0.156 | 0.201 | 0.156 | 0.209 | 0.156 | 0.207 | 0.156 | 0.207 |
| | | Lin Attn | 0.139 | 0.191 | 0.144 | 0.197 | 0.143 | 0.195 | 0.142 | 0.194 | 0.143 | 0.196 | 0.143 | 0.194 | 0.143 | 0.194 | 0.142 | 0.193 | 0.144 | 0.196 |
| | | GLin Attn | 0.142 | 0.194 | 0.163 | 0.213 | 0.163 | 0.212 | 0.161 | 0.210 | 0.165 | 0.213 | 0.163 | 0.212 | 0.164 | 0.213 | 0.164 | 0.216 | 0.164 | 0.213 |
| | | ELin Attn | 0.143 | 0.195 | 0.157 | 0.211 | 0.148 | 0.207 | 0.146 | 0.197 | 0.151 | 0.211 | 0.156 | 0.208 | 0.157 | 0.211 | 0.156 | 0.207 | 0.157 | 0.208 |
| | | Fixed Attn | 0.142 | 0.198 | 0.158 | 0.210 | 0.151 | 0.209 | 0.147 | 0.194 | 0.152 | 0.206 | 0.154 | 0.206 | 0.153 | 0.204 | 0.153 | 0.207 | 0.153 | 0.205 |
| | 48 | Std Attn | 0.109 | 0.151 | 0.116 | 0.161 | 0.115 | 0.159 | 0.113 | 0.157 | 0.116 | 0.160 | 0.127 | 0.177 | 0.120 | 0.168 | 0.128 | 0.181 | 0.127 | 0.177 |
| | | Lin Attn | 0.110 | 0.153 | 0.114 | 0.156 | 0.113 | 0.156 | 0.115 | 0.158 | 0.113 | 0.153 | 0.113 | 0.155 | 0.112 | 0.153 | 0.112 | 0.155 | 0.112 | 0.155 |
| | | GLin Attn | 0.116 | 0.159 | 0.144 | 0.190 | 0.144 | 0.189 | 0.144 | 0.191 | 0.145 | 0.193 | 0.145 | 0.191 | 0.144 | 0.192 | 0.143 | 0.189 | 0.143 | 0.189 |
| | | ELin Attn | 0.112 | 0.156 | 0.119 | 0.167 | 0.116 | 0.166 | 0.114 | 0.158 | 0.117 | 0.166 | 0.118 | 0.165 | 0.122 | 0.172 | 0.121 | 0.167 | 0.123 | 0.171 |
| | | Fixed Attn | 0.115 | 0.159 | 0.124 | 0.171 | 0.118 | 0.170 | 0.112 | 0.155 | 0.122 | 0.170 | 0.126 | 0.173 | 0.121 | 0.171 | 0.122 | 0.168 | 0.121 | 0.167 |
| | 24 | Std Attn | 0.085 | 0.115 | 0.091 | 0.124 | 0.091 | 0.124 | 0.089 | 0.118 | 0.093 | 0.124 | 0.096 | 0.132 | 0.095 | 0.130 | 0.095 | 0.129 | 0.094 | 0.126 |
| | | Lin Attn | 0.083 | 0.114 | 0.087 | 0.117 | 0.087 | 0.118 | 0.088 | 0.117 | 0.087 | 0.115 | 0.086 | 0.116 | 0.087 | 0.118 | 0.087 | 0.117 | 0.087 | 0.116 |
| | | GLin Attn | 0.090 | 0.122 | 0.103 | 0.132 | 0.103 | 0.132 | 0.101 | 0.129 | 0.103 | 0.132 | 0.103 | 0.129 | 0.103 | 0.134 | 0.104 | 0.132 | 0.103 | 0.132 |
| | | ELin Attn | 0.089 | 0.119 | 0.092 | 0.123 | 0.092 | 0.123 | 0.087 | 0.116 | 0.090 | 0.122 | 0.092 | 0.128 | 0.092 | 0.128 | 0.092 | 0.127 | 0.092 | 0.124 |
| | | Fixed Attn | 0.088 | 0.120 | 0.095 | 0.127 | 0.093 | 0.126 | 0.091 | 0.122 | 0.094 | 0.128 | 0.094 | 0.125 | 0.093 | 0.125 | 0.092 | 0.124 | 0.094 | 0.127 |
| | 12 | Std Attn | 0.067 | 0.086 | 0.073 | 0.091 | 0.072 | 0.091 | 0.071 | 0.091 | 0.072 | 0.093 | 0.072 | 0.091 | 0.072 | 0.091 | 0.072 | 0.094 | 0.072 | 0.091 |
| | | Lin Attn | 0.067 | 0.086 | 0.069 | 0.088 | 0.069 | 0.089 | 0.069 | 0.088 | 0.069 | 0.089 | 0.068 | 0.088 | 0.068 | 0.087 | 0.068 | 0.088 | 0.069 | 0.090 |
| | | GLin Attn | 0.070 | 0.091 | 0.078 | 0.092 | 0.077 | 0.095 | 0.070 | 0.090 | 0.072 | 0.093 | 0.071 | 0.088 | 0.077 | 0.093 | 0.071 | 0.088 | 0.071 | 0.090 |
| | | ELin Attn | 0.069 | 0.087 | 0.071 | 0.090 | 0.071 | 0.091 | 0.069 | 0.087 | 0.072 | 0.091 | 0.071 | 0.091 | 0.071 | 0.090 | 0.072 | 0.091 | 0.071 | 0.090 |
| | | Fixed Attn | 0.069 | 0.090 | 0.073 | 0.093 | 0.072 | 0.094 | 0.071 | 0.089 | 0.072 | 0.093 | 0.072 | 0.091 | 0.072 | 0.090 | 0.072 | 0.089 | 0.072 | 0.091 |

*Table 10.* Results showing that pure AR/WAVE Transformers effectively utilize extended lookback $L_I$, while baselines experience performance degradation. $L_I \in \{512, 1024, 2048, 4096\}$ with $L_P \in \{12, 24, 48, 96\}$ are evaluated. Test set MSE and MAE for each model on each setup are presented.

| Dataset | $L_I$ | $L_P$ | Std Attn | | Std Attn +ARMA | | Lin Attn | | Lin Attn +ARMA | | GLin Attn | | GLin Attn +ARMA | | ELin Attn | | ELin Attn +ARMA | | Fixed Attn | | Fixed Attn +ARMA | | FITS | | iTransformer | | CATS | | PatchTST | | DLinear | | EncFormer | |
|---|---|---|---|---|---|---|---|---|---|---|---|---|---|---|---|---|---|---|---|---|---|---|---|---|---|---|---|---|---|---|---|---|---|---|
| | | | MSE | MAE | MSE | MAE | MSE | MAE | MSE | MAE | MSE | MAE | MSE | MAE | MSE | MAE | MSE | MAE | MSE | MAE | MSE | MAE | MSE | MAE | MSE | MAE | MSE | MAE | MSE | MAE | MSE | MAE | MSE | MAE |
| Weather | 512 | 96 | 0.144 | 0.195 | 0.142 | 0.193 | 0.142 | 0.194 | 0.139 | 0.191 | 0.161 | 0.210 | 0.142 | 0.194 | 0.145 | 0.197 | 0.143 | 0.195 | 0.147 | 0.194 | 0.142 | 0.198 | 0.151 | 0.204 | 0.158 | 0.140 | 0.146 | 0.198 | 0.151 | 0.224 | 0.150 | 0.209 | 0.188 | 0.248 |
| | | 48 | 0.113 | 0.157 | 0.109 | 0.151 | 0.115 | 0.158 | 0.110 | 0.153 | 0.144 | 0.191 | 0.116 | 0.159 | 0.114 | 0.158 | 0.112 | 0.156 | 0.112 | 0.155 | 0.115 | 0.159 | 0.125 | 0.177 | 0.132 | 0.176 | 0.116 | 0.160 | 0.118 | 0.161 | 0.122 | 0.177 | 0.143 | 0.199 |
| | | 24 | 0.089 | 0.118 | 0.085 | 0.115 | 0.088 | 0.117 | 0.083 | 0.114 | 0.101 | 0.129 | 0.090 | 0.122 | 0.087 | 0.116 | 0.089 | 0.119 | 0.091 | 0.122 | 0.088 | 0.120 | 0.103 | 0.147 | 0.101 | 0.139 | 0.090 | 0.125 | 0.092 | 0.122 | 0.147 | 0.102 | 0.117 | 0.162 |
| | | 12 | 0.071 | 0.091 | 0.067 | 0.086 | 0.069 | 0.088 | 0.067 | 0.086 | 0.070 | 0.090 | 0.070 | 0.091 | 0.069 | 0.087 | 0.069 | 0.087 | 0.071 | 0.089 | 0.069 | 0.090 | 0.078 | 0.112 | 0.078 | 0.105 | 0.069 | 0.092 | 0.071 | 0.097 | 0.078 | 0.115 | 0.093 | 0.120 |
| | 1024 | 96 | 0.145 | 0.196 | 0.142 | 0.194 | 0.144 | 0.198 | 0.141 | 0.195 | 0.161 | 0.212 | 0.145 | 0.198 | 0.147 | 0.198 | 0.144 | 0.197 | 0.149 | 0.201 | 0.144 | 0.198 | 0.168 | 0.222 | 0.164 | 0.219 | 0.148 | 0.203 | 0.167 | 0.223 | 0.166 | 0.222 | 0.175 | 0.247 |
| | | 48 | 0.112 | 0.156 | 0.110 | 0.152 | 0.110 | 0.154 | 0.109 | 0.152 | 0.144 | 0.193 | 0.115 | 0.157 | 0.114 | 0.159 | 0.117 | 0.156 | 0.116 | 0.161 | 0.114 | 0.162 | 0.132 | 0.185 | 0.127 | 0.181 | 0.118 | 0.164 | 0.133 | 0.187 | 0.129 | 0.181 | 0.141 | 0.202 |
| | | 24 | 0.096 | 0.130 | 0.088 | 0.118 | 0.086 | 0.112 | 0.086 | 0.116 | 0.091 | 0.123 | 0.089 | 0.122 | 0.090 | 0.122 | 0.091 | 0.123 | 0.095 | 0.129 | 0.092 | 0.126 | 0.102 | 0.147 | 0.098 | 0.141 | 0.091 | 0.128 | 0.101 | 0.144 | 0.100 | 0.144 | 0.104 | 0.146 |
| | | 12 | 0.075 | 0.092 | 0.068 | 0.086 | 0.068 | 0.088 | 0.067 | 0.087 | 0.069 | 0.090 | 0.068 | 0.087 | 0.073 | 0.092 | 0.071 | 0.090 | 0.072 | 0.092 | 0.071 | 0.091 | 0.078 | 0.112 | 0.078 | 0.114 | 0.073 | 0.102 | 0.077 | 0.110 | 0.077 | 0.112 | 0.075 | 0.119 |
| | 2048 | 96 | 0.154 | 0.208 | 0.144 | 0.200 | 0.139 | 0.195 | 0.139 | 0.195 | 0.145 | 0.216 | 0.144 | 0.200 | 0.155 | 0.210 | 0.153 | 0.209 | 0.155 | 0.213 | 0.152 | 0.216 | 0.169 | 0.225 | 0.164 | 0.220 | 0.154 | 0.212 | 0.168 | 0.224 | 0.167 | 0.223 | 0.194 | 0.270 |
| | | 48 | 0.114 | 0.159 | 0.110 | 0.157 | 0.110 | 0.155 | 0.108 | 0.152 | 0.146 | 0.198 | 0.110 | 0.155 | 0.115 | 0.163 | 0.115 | 0.163 | 0.132 | 0.184 | 0.123 | 0.173 | 0.134 | 0.187 | 0.128 | 0.184 | 0.122 | 0.176 | 0.135 | 0.190 | 0.131 | 0.185 | 0.129 | 0.190 |
| | | 24 | 0.097 | 0.135 | 0.085 | 0.122 | 0.086 | 0.115 | 0.085 | 0.115 | 0.096 | 0.122 | 0.087 | 0.118 | 0.090 | 0.125 | 0.090 | 0.125 | 0.132 | 0.096 | 0.093 | 0.128 | 0.102 | 0.149 | 0.101 | 0.150 | 0.096 | 0.135 | 0.103 | 0.152 | 0.100 | 0.152 | 0.108 | 0.157 |
| | | 12 | 0.073 | 0.094 | 0.068 | 0.089 | 0.068 | 0.088 | 0.067 | 0.086 | 0.069 | 0.090 | 0.068 | 0.089 | 0.072 | 0.093 | 0.072 | 0.093 | 0.073 | 0.095 | 0.072 | 0.093 | 0.080 | 0.120 | 0.082 | 0.127 | 0.078 | 0.104 | 0.080 | 0.124 | 0.077 | 0.117 | 0.079 | 0.111 |
| | 4096 | 96 | 0.150 | 0.207 | 0.143 | 0.203 | 0.139 | 0.198 | 0.139 | 0.201 | 0.166 | 0.220 | 0.142 | 0.201 | 0.153 | 0.213 | 0.150 | 0.212 | 0.157 | 0.215 | 0.155 | 0.213 | 0.168 | 0.228 | 0.168 | 0.231 | 0.164 | 0.225 | 0.171 | 0.232 | 0.171 | 0.235 | 0.212 | 0.281 |
| | | 48 | 0.114 | 0.162 | 0.111 | 0.159 | 0.109 | 0.155 | 0.107 | 0.156 | 0.137 | 0.193 | 0.123 | 0.176 | 0.117 | 0.168 | 0.116 | 0.166 | 0.117 | 0.168 | 0.111 | 0.159 | 0.135 | 0.194 | 0.173 | 0.201 | 0.138 | 0.194 | 0.138 | 0.199 | 0.132 | 0.191 | 0.139 | 0.194 |
| | | 24 | 0.097 | 0.140 | 0.085 | 0.122 | 0.085 | 0.117 | 0.085 | 0.118 | 0.086 | 0.121 | 0.086 | 0.120 | 0.093 | 0.130 | 0.089 | 0.125 | 0.096 | 0.132 | 0.094 | 0.129 | 0.105 | 0.156 | 0.103 | 0.154 | 0.108 | 0.153 | 0.107 | 0.162 | 0.102 | 0.152 | 0.114 | 0.160 |
| | | 12 | 0.072 | 0.096 | 0.067 | 0.089 | 0.068 | 0.088 | 0.068 | 0.088 | 0.070 | 0.092 | 0.068 | 0.089 | 0.071 | 0.096 | 0.071 | 0.097 | 0.071 | 0.097 | 0.071 | 0.096 | 0.086 | 0.135 | 0.085 | 0.133 | 0.081 | 0.118 | 0.085 | 0.140 | 0.079 | 0.112 | 0.080 | 0.110 |
| ETTm1 | 512 | 96 | 0.305 | 0.359 | 0.301 | 0.354 | 0.303 | 0.355 | 0.296 | 0.351 | 0.336 | 0.382 | 0.299 | 0.355 | 0.305 | 0.356 | 0.301 | 0.354 | 0.298 | 0.347 | 0.296 | 0.344 | 0.305 | 0.347 | 0.352 | 0.378 | 0.300 | 0.354 | 0.323 | 0.362 | 0.305 | 0.348 | 0.686 | 0.603 |
| | | 48 | 0.287 | 0.344 | 0.276 | 0.333 | 0.278 | 0.336 | 0.266 | 0.328 | 0.500 | 0.472 | 0.372 | 0.334 | 0.284 | 0.346 | 0.282 | 0.342 | 0.284 | 0.338 | 0.280 | 0.340 | 0.280 | 0.331 | 0.304 | 0.346 | 0.266 | 0.328 | 0.289 | 0.341 | 0.278 | 0.329 | 0.475 | 0.474 |
| | | 24 | 0.246 | 0.312 | 0.223 | 0.293 | 0.218 | 0.293 | 0.196 | 0.279 | 0.473 | 0.429 | 0.225 | 0.305 | 0.239 | 0.305 | 0.241 | 0.307 | 0.284 | 0.332 | 0.255 | 0.321 | 0.218 | 0.289 | 0.223 | 0.293 | 0.197 | 0.277 | 0.235 | 0.304 | 0.216 | 0.288 | 0.352 | 0.391 |
| | | 12 | 0.218 | 0.287 | 0.156 | 0.241 | 0.151 | 0.240 | 0.128 | 0.222 | 0.320 | 0.335 | 0.144 | 0.237 | 0.157 | 0.246 | 0.153 | 0.246 | 0.203 | 0.282 | 0.174 | 0.261 | 0.144 | 0.235 | 0.155 | 0.246 | 0.125 | 0.222 | 0.128 | 0.224 | 0.140 | 0.230 | 0.204 | 0.294 |
| | 1024 | 96 | 0.318 | 0.366 | 0.308 | 0.357 | 0.305 | 0.356 | 0.304 | 0.357 | 0.339 | 0.384 | 0.309 | 0.362 | 0.317 | 0.367 | 0.308 | 0.359 | 0.306 | 0.353 | 0.303 | 0.353 | 0.310 | 0.353 | 0.340 | 0.377 | 0.314 | 0.367 | 0.321 | 0.363 | 0.308 | 0.353 | 0.493 | 0.497 |
| | | 48 | 0.308 | 0.355 | 0.281 | 0.333 | 0.292 | 0.345 | 0.272 | 0.333 | 0.532 | 0.486 | 0.277 | 0.340 | 0.315 | 0.363 | 0.296 | 0.349 | 0.303 | 0.356 | 0.300 | 0.351 | 0.285 | 0.337 | 0.310 | 0.350 | 0.293 | 0.346 | 0.290 | 0.340 | 0.287 | 0.344 | 0.442 | 0.423 |
| | | 24 | 0.267 | 0.332 | 0.231 | 0.305 | 0.221 | 0.294 | 0.202 | 0.282 | 0.504 | 0.441 | 0.218 | 0.298 | 0.266 | 0.328 | 0.252 | 0.316 | 0.292 | 0.352 | 0.271 | 0.330 | 0.222 | 0.293 | 0.228 | 0.301 | 0.210 | 0.289 | 0.219 | 0.292 | 0.219 | 0.298 | 0.338 | 0.373 |
| | | 12 | 0.228 | 0.236 | 0.145 | 0.299 | 0.138 | 0.229 | 0.130 | 0.222 | 0.317 | 0.323 | 0.140 | 0.238 | 0.160 | 0.248 | 0.157 | 0.251 | 0.221 | 0.292 | 0.179 | 0.269 | 0.144 | 0.237 | 0.153 | 0.249 | 0.133 | 0.229 | 0.151 | 0.245 | 0.140 | 0.232 | 0.182 | 0.268 |
| | 2048 | 96 | 0.314 | 0.364 | 0.302 | 0.357 | 0.301 | 0.356 | 0.301 | 0.356 | 0.342 | 0.387 | 0.304 | 0.360 | 0.313 | 0.366 | 0.305 | 0.360 | 0.304 | 0.358 | 0.304 | 0.356 | 0.311 | 0.356 | 0.330 | 0.373 | 0.335 | 0.384 | 0.318 | 0.366 | 0.311 | 0.360 | 0.503 | 0.504 |
| | | 48 | 0.302 | 0.353 | 0.281 | 0.334 | 0.282 | 0.339 | 0.266 | 0.331 | 0.529 | 0.482 | 0.272 | 0.338 | 0.309 | 0.364 | 0.298 | 0.355 | 0.312 | 0.369 | 0.314 | 0.369 | 0.287 | 0.341 | 0.290 | 0.341 | 0.307 | 0.359 | 0.305 | 0.361 | 0.281 | 0.339 | 0.529 | 0.563 |
| | | 24 | 0.270 | 0.337 | 0.228 | 0.305 | 0.214 | 0.290 | 0.198 | 0.283 | 0.272 | 0.330 | 0.211 | 0.293 | 0.273 | 0.330 | 0.244 | 0.314 | 0.312 | 0.367 | 0.279 | 0.338 | 0.221 | 0.298 | 0.225 | 0.304 | 0.253 | 0.324 | 0.217 | 0.294 | 0.220 | 0.298 | 0.409 | 0.436 |
| | | 12 | 0.226 | 0.238 | 0.146 | 0.296 | 0.136 | 0.232 | 0.128 | 0.222 | 0.164 | 0.255 | 0.140 | 0.238 | 0.229 | 0.295 | 0.162 | 0.252 | 0.225 | 0.294 | 0.173 | 0.260 | 0.163 | 0.258 | 0.147 | 0.201 | 0.148 | 0.251 | 0.160 | 0.257 | 0.142 | 0.239 | 0.218 | 0.306 |
| | 4096 | 96 | 0.305 | 0.365 | 0.296 | 0.356 | 0.299 | 0.361 | 0.298 | 0.359 | 0.338 | 0.385 | 0.296 | 0.356 | 0.306 | 0.366 | 0.298 | 0.359 | 0.303 | 0.362 | 0.299 | 0.356 | 0.319 | 0.366 | 0.357 | 0.394 | 0.454 | 0.455 | 0.317 | 0.364 | 0.324 | 0.368 | 0.619 | 0.576 |
| | | 48 | 0.298 | 0.354 | 0.276 | 0.339 | 0.292 | 0.350 | 0.274 | 0.342 | 0.551 | 0.502 | 0.272 | 0.341 | 0.318 | 0.374 | 0.293 | 0.357 | 0.332 | 0.386 | 0.316 | 0.370 | 0.299 | 0.353 | 0.315 | 0.365 | 0.387 | 0.415 | 0.321 | 0.373 | 0.296 | 0.354 | 0.481 | 0.509 |
| | | 24 | 0.266 | 0.331 | 0.227 | 0.302 | 0.220 | 0.303 | 0.203 | 0.291 | 0.252 | 0.322 | 0.211 | 0.298 | 0.269 | 0.333 | 0.252 | 0.325 | 0.343 | 0.386 | 0.279 | 0.344 | 0.231 | 0.308 | 0.245 | 0.323 | 0.306 | 0.351 | 0.225 | 0.300 | 0.230 | 0.309 | 0.368 | 0.406 |
| | | 12 | 0.230 | 0.298 | 0.138 | 0.233 | 0.135 | 0.231 | 0.129 | 0.224 | 0.156 | 0.246 | 0.137 | 0.232 | 0.233 | 0.299 | 0.217 | 0.292 | 0.170 | 0.298 | 0.168 | 0.260 | 0.160 | 0.260 | 0.178 | 0.274 | 0.213 | 0.307 | 0.176 | 0.278 | 0.148 | 0.251 | 0.243 | 0.359 |

*Table 11.* Results of long-term time series forecasting. Baselines are reported with their best-performing results. Test set MSE and MAE for each model on each setup are reported.

| Model | | Std Attn | Std Attn +ARMA | Lin Attn | Lin Attn +ARMA | GLin Attn | GLin Attn +ARMA | ELin Attn | ELin Attn +ARMA | Fixed Attn | Fixed Attn +ARMA | FITS | iTransformer | CATS | PatchTST | DLinear | EncFormer |
|---|---|---|---|---|---|---|---|---|---|---|---|---|---|---|---|---|---|
| Metrics | | MSE | MSE | MSE | MSE | MSE | MSE | MSE | MSE | MSE | MSE | MSE | MSE | MSE | MSE | MSE | MSE |
| Weather | 96 | 0.144 | 0.142 | 0.142 | 0.139 | 0.161 | 0.142 | 0.146 | 0.143 | 0.147 | 0.142 | 0.149 | 0.158 | 0.143 | 0.149 | 0.150 | 0.188 |
| | 192 | 0.193 | 0.190 | 0.190 | 0.188 | 0.191 | 0.191 | 0.192 | 0.194 | 0.188 | 0.188 | 0.189 | 0.203 | 0.188 | 0.190 | 0.211 | 0.215 |
| | 336 | 0.245 | 0.242 | 0.239 | 0.237 | 0.238 | 0.236 | 0.238 | 0.238 | 0.236 | 0.236 | 0.237 | 0.250 | 0.235 | 0.240 | 0.255 | 0.270 |
| | 720 | 0.301 | 0.299 | 0.300 | 0.295 | 0.300 | 0.296 | 0.303 | 0.302 | 0.309 | 0.307 | 0.311 | 0.316 | 0.297 | 0.306 | 0.316 | 0.332 |
| Solar | 96 | 0.196 | 0.192 | 0.183 | 0.180 | 0.209 | 0.182 | 0.194 | 0.191 | 0.195 | 0.187 | 0.189 | 0.230 | 0.182 | 0.209 | 0.208 | 0.201 |
| | 192 | 0.192 | 0.191 | 0.193 | 0.185 | 0.201 | 0.189 | 0.198 | 0.194 | 0.197 | 0.192 | 0.206 | 0.204 | 0.214 | 0.192 | 0.208 | 0.209 |
| | 336 | 0.193 | 0.191 | 0.198 | 0.197 | 0.199 | 0.196 | 0.191 | 0.191 | 0.201 | 0.196 | 0.219 | 0.222 | 0.216 | 0.200 | 0.221 | 0.221 |
| | 720 | 0.210 | 0.206 | 0.208 | 0.206 | 0.206 | 0.204 | 0.208 | 0.205 | 0.204 | 0.205 | 0.221 | 0.218 | 0.213 | 0.205 | 0.227 | 0.218 |
| ETTh1 | 96 | 0.357 | 0.360 | 0.358 | 0.361 | 0.378 | 0.368 | 0.360 | 0.356 | 0.362 | 0.359 | 0.369 | 0.396 | 0.365 | 0.370 | 0.370 | 0.986 |
| | 192 | 0.393 | 0.391 | 0.404 | 0.398 | 0.401 | 0.396 | 0.391 | 0.389 | 0.403 | 0.395 | 0.435 | 0.431 | 0.404 | 0.412 | 0.405 | 0.814 |
| | 336 | 0.418 | 0.415 | 0.428 | 0.424 | 0.419 | 0.416 | 0.42 | 0.418 | 0.423 | 0.419 | 0.468 | 0.459 | 0.423 | 0.422 | 0.439 | 0.883 |
| | 720 | 0.487 | 0.478 | 0.471 | 0.462 | 0.469 | 0.453 | 0.463 | 0.458 | 0.468 | 0.466 | 0.488 | 0.528 | 0.441 | 0.447 | 0.472 | 0.941 |
| ETTh2 | 96 | 0.266 | 0.268 | 0.273 | 0.275 | 0.285 | 0.281 | 0.267 | 0.263 | 0.288 | 0.276 | 0.270 | 0.299 | 0.259 | 0.274 | 0.277 | 1.303 |
| | 192 | 0.336 | 0.339 | 0.347 | 0.336 | 0.335 | 0.333 | 0.335 | 0.329 | 0.342 | 0.338 | 0.348 | 0.365 | 0.315 | 0.339 | 0.375 | 0.939 |
| | 336 | 0.371 | 0.366 | 0.375 | 0.373 | 0.363 | 0.366 | 0.365 | 0.357 | 0.361 | 0.363 | 0.376 | 0.407 | 0.339 | 0.329 | 0.448 | 0.551 |
| | 720 | 0.385 | 0.382 | 0.377 | 0.371 | 0.385 | 0.381 | 0.379 | 0.379 | 0.401 | 0.398 | 0.421 | 0.423 | 0.365 | 0.379 | 0.605 | 0.714 |
| ETTm1 | 96 | 0.305 | 0.301 | 0.303 | 0.296 | 0.336 | 0.299 | 0.305 | 0.301 | 0.298 | 0.296 | 0.305 | 0.325 | 0.282 | 0.290 | 0.299 | 0.686 |
| | 192 | 0.332 | 0.329 | 0.329 | 0.327 | 0.332 | 0.332 | 0.332 | 0.329 | 0.334 | 0.328 | 0.334 | 0.352 | 0.326 | 0.328 | 0.335 | 0.636 |
| | 336 | 0.355 | 0.354 | 0.363 | 0.362 | 0.360 | 0.359 | 0.358 | 0.356 | 0.355 | 0.354 | 0.363 | 0.382 | 0.358 | 0.359 | 0.359 | 0.791 |
| | 720 | 0.395 | 0.395 | 0.409 | 0.406 | 0.398 | 0.393 | 0.395 | 0.393 | 0.400 | 0.396 | 0.412 | 0.432 | 0.414 | 0.405 | 0.396 | 0.825 |
| ETTm2 | 96 | 0.177 | 0.174 | 0.167 | 0.162 | 0.178 | 0.172 | 0.178 | 0.174 | 0.167 | 0.166 | 0.164 | 0.187 | 0.158 | 0.165 | 0.184 | 0.481 |
| | 192 | 0.220 | 0.218 | 0.218 | 0.215 | 0.216 | 0.216 | 0.217 | 0.217 | 0.225 | 0.213 | 0.211 | 0.232 | 0.211 | 0.214 | 0.218 | 0.434 |
| | 336 | 0.256 | 0.256 | 0.263 | 0.259 | 0.264 | 0.258 | 0.260 | 0.254 | 0.256 | 0.253 | 0.259 | 0.281 | 0.261 | 0.266 | 0.263 | 0.461 |
| | 720 | 0.341 | 0.337 | 0.339 | 0.337 | 0.340 | 0.334 | 0.330 | 0.329 | 0.332 | 0.328 | 0.352 | 0.358 | 0.340 | 0.344 | 0.341 | 0.928 |

*Table 12.* Experiment results of the performance comparison with MEGA with $L_P \in \{12, 24, 48, 96\}$.

| Model | | Std Attn | | Std Attn +ARMA | | Lin Attn | | Lin Attn +ARMA | | GLin Attn | | GLin Attn +ARMA | | MEGA | |
|---|---|---|---|---|---|---|---|---|---|---|---|---|---|---|---|
| Metrics | | MSE | MAE | MSE | MAE | MSE | MAE | MSE | MAE | MSE | MAE | MSE | MAE | MSE | MAE |
| Weather | 96 | 0.144 | 0.195 | 0.142 | 0.193 | 0.142 | 0.194 | 0.139 | 0.191 | 0.161 | 0.210 | 0.142 | 0.194 | 0.164 | 0.212 |
| | 48 | 0.113 | 0.157 | 0.109 | 0.151 | 0.115 | 0.158 | 0.110 | 0.153 | 0.144 | 0.191 | 0.116 | 0.159 | 0.141 | 0.187 |
| | 24 | 0.089 | 0.118 | 0.085 | 0.115 | 0.088 | 0.117 | 0.083 | 0.114 | 0.101 | 0.129 | 0.090 | 0.122 | 0.102 | 0.128 |
| | 12 | 0.071 | 0.091 | 0.067 | 0.086 | 0.069 | 0.088 | 0.067 | 0.086 | 0.070 | 0.090 | 0.070 | 0.091 | 0.077 | 0.090 |
| Solar | 96 | 0.196 | 0.263 | 0.192 | 0.257 | 0.183 | 0.247 | 0.180 | 0.244 | 0.209 | 0.283 | 0.182 | 0.243 | 0.235 | 0.302 |
| | 48 | 0.160 | 0.230 | 0.151 | 0.223 | 0.152 | 0.219 | 0.149 | 0.217 | 0.177 | 0.258 | 0.154 | 0.222 | 0.342 | 0.406 |
| | 24 | 0.112 | 0.180 | 0.098 | 0.168 | 0.098 | 0.166 | 0.095 | 0.162 | 0.143 | 0.232 | 0.099 | 0.167 | 0.231 | 0.284 |
| | 12 | 0.069 | 0.137 | 0.055 | 0.118 | 0.056 | 0.113 | 0.052 | 0.111 | 0.063 | 0.139 | 0.059 | 0.121 | 0.097 | 0.170 |
| ETTh1 | 96 | 0.357 | 0.393 | 0.360 | 0.395 | 0.358 | 0.396 | 0.361 | 0.399 | 0.378 | 0.406 | 0.368 | 0.404 | 0.373 | 0.468 |
| | 48 | 0.334 | 0.375 | 0.331 | 0.374 | 0.331 | 0.375 | 0.331 | 0.376 | 0.349 | 0.386 | 0.337 | 0.381 | 0.348 | 0.386 |
| | 24 | 0.312 | 0.365 | 0.299 | 0.357 | 0.299 | 0.356 | 0.299 | 0.357 | 0.349 | 0.394 | 0.303 | 0.360 | 0.345 | 0.387 |
| | 12 | 0.290 | 0.345 | 0.280 | 0.340 | 0.285 | 0.342 | 0.272 | 0.337 | 0.554 | 0.503 | 0.277 | 0.340 | 0.551 | 0.499 |
| ETTh2 | 96 | 0.266 | 0.330 | 0.268 | 0.331 | 0.273 | 0.336 | 0.275 | 0.338 | 0.285 | 0.338 | 0.281 | 0.345 | 0.278 | 0.330 |
| | 48 | 0.213 | 0.290 | 0.216 | 0.294 | 0.215 | 0.290 | 0.217 | 0.293 | 0.233 | 0.294 | 0.220 | 0.295 | 0.230 | 0.295 |
| | 24 | 0.160 | 0.252 | 0.160 | 0.252 | 0.159 | 0.250 | 0.162 | 0.252 | 0.182 | 0.263 | 0.164 | 0.254 | 0.180 | 0.261 |
| | 12 | 0.129 | 0.229 | 0.125 | 0.224 | 0.124 | 0.224 | 0.125 | 0.224 | 0.168 | 0.263 | 0.127 | 0.224 | 0.167 | 0.263 |
| ETTm1 | 96 | 0.305 | 0.359 | 0.301 | 0.354 | 0.303 | 0.355 | 0.296 | 0.351 | 0.336 | 0.382 | 0.299 | 0.355 | 0.335 | 0.378 |
| | 48 | 0.287 | 0.344 | 0.276 | 0.333 | 0.278 | 0.336 | 0.266 | 0.328 | 0.500 | 0.472 | 0.372 | 0.334 | 0.507 | 0.469 |
| | 24 | 0.246 | 0.312 | 0.223 | 0.293 | 0.218 | 0.293 | 0.196 | 0.279 | 0.473 | 0.429 | 0.225 | 0.305 | 0.487 | 0.434 |
| | 12 | 0.218 | 0.287 | 0.156 | 0.241 | 0.151 | 0.240 | 0.128 | 0.222 | 0.320 | 0.335 | 0.144 | 0.237 | 0.318 | 0.322 |
| ETTm2 | 96 | 0.177 | 0.262 | 0.174 | 0.261 | 0.167 | 0.255 | 0.162 | 0.250 | 0.178 | 0.260 | 0.172 | 0.258 | 0.176 | 0.258 |
| | 48 | 0.139 | 0.238 | 0.137 | 0.236 | 0.145 | 0.248 | 0.143 | 0.250 | 0.167 | 0.264 | 0.148 | 0.249 | 0.157 | 0.253 |
| | 24 | 0.121 | 0.219 | 0.117 | 0.217 | 0.110 | 0.211 | 0.101 | 0.199 | 0.132 | 0.227 | 0.110 | 0.209 | 0.126 | 0.223 |
| | 12 | 0.086 | 0.175 | 0.083 | 0.174 | 0.083 | 0.174 | 0.078 | 0.169 | 0.089 | 0.178 | 0.082 | 0.172 | 0.088 | 0.178 |
| PEMS03 | 96 | 0.171 | 0.280 | 0.153 | 0.263 | 0.149 | 0.258 | 0.143 | 0.252 | 0.360 | 0.416 | 0.147 | 0.257 | 0.223 | 0.329 |
| | 48 | 0.122 | 0.234 | 0.106 | 0.217 | 0.105 | 0.215 | 0.102 | 0.210 | 0.299 | 0.380 | 0.109 | 0.216 | 0.202 | 0.317 |
| | 24 | 0.089 | 0.201 | 0.079 | 0.187 | 0.081 | 0.188 | 0.075 | 0.181 | 0.097 | 0.209 | 0.082 | 0.189 | 0.129 | 0.244 |
| | 12 | 0.067 | 0.173 | 0.063 | 0.167 | 0.065 | 0.169 | 0.062 | 0.165 | 0.078 | 0.185 | 0.067 | 0.177 | 0.089 | 0.199 |

*Table 13.* Additional ablation studies: without AR loss.

| Model | Metrics | Std Attn w/o AR Loss MSE | MAE | Std Attn Original MSE | MAE | Std Attn +ARMA w/o AR Loss MSE | MAE | Std Attn +ARMA Original MSE | MAE | Lin Attn w/o AR Loss MSE | MAE | Lin Attn Original MSE | MAE | Lin Attn +ARMA w/o AR Loss MSE | MAE | Lin Attn +ARMA Original MSE | MAE | GLin Attn w/o AR Loss MSE | MAE | GLin Attn Original MSE | MAE | GLin Attn +ARMA w/o AR Loss MSE | MAE | GLin Attn +ARMA Original MSE | MAE |
|---|---|---|---|---|---|---|---|---|---|---|---|---|---|---|---|---|---|---|---|---|---|---|---|---|---|
| ETTm1 96 | | 0.312 | 0.363 | 0.305 | 0.359 | 0.304 | 0.358 | 0.301 | 0.354 | 0.313 | 0.365 | 0.303 | 0.355 | 0.303 | 0.361 | **0.296** | **0.351** | 0.314 | 0.374 | 0.336 | 0.382 | 0.312 | 0.363 | 0.299 | 0.355 |
| ETTm1 48 | | 0.289 | 0.346 | 0.287 | 0.344 | 0.287 | 0.341 | 0.286 | 0.345 | 0.291 | 0.349 | 0.283 | 0.343 | 0.286 | 0.345 | **0.266** | **0.328** | 0.283 | 0.343 | 0.5 | 0.472 | 0.277 | 0.336 | 0.372 | 0.334 |
| ETTm1 24 | | 0.224 | 0.294 | 0.216 | 0.293 | 0.217 | 0.293 | 0.211 | 0.289 | 0.217 | 0.293 | 0.212 | 0.290 | 0.208 | 0.285 | **0.196** | **0.279** | 0.212 | 0.290 | 0.473 | 0.429 | 0.208 | 0.285 | 0.225 | 0.305 |
| ETTm1 12 | | 0.142 | 0.235 | 0.136 | 0.232 | 0.145 | 0.237 | 0.137 | 0.229 | 0.142 | 0.233 | 0.140 | 0.230 | 0.156 | 0.241 | **0.128** | **0.222** | 0.151 | 0.240 | 0.320 | 0.335 | 0.144 | 0.237 | | |
| ETTm2 96 | | 0.176 | 0.263 | 0.172 | 0.261 | 0.178 | 0.267 | 0.172 | 0.263 | 0.180 | 0.270 | 0.178 | 0.269 | 0.177 | 0.262 | 0.174 | 0.261 | 0.167 | 0.255 | **0.162** | **0.250** | 0.178 | 0.260 | 0.172 | 0.258 |
| ETTm2 48 | | 0.139 | 0.237 | 0.137 | 0.236 | 0.138 | 0.240 | **0.134** | **0.235** | 0.140 | 0.239 | 0.138 | 0.235 | 0.139 | 0.238 | 0.137 | 0.236 | 0.145 | 0.248 | 0.143 | 0.250 | 0.167 | 0.264 | 0.148 | 0.249 |
| ETTm2 24 | | 0.105 | 0.198 | 0.103 | 0.196 | 0.105 | 0.200 | 0.104 | 0.199 | 0.105 | 0.197 | 0.103 | **0.195** | 0.121 | 0.219 | 0.117 | 0.217 | 0.110 | 0.211 | **0.101** | 0.199 | 0.132 | 0.227 | 0.110 | 0.209 |
| ETTm2 12 | | 0.080 | 0.167 | 0.079 | 0.170 | 0.079 | 0.167 | 0.079 | 0.167 | 0.082 | 0.168 | 0.079 | **0.166** | 0.086 | 0.175 | 0.083 | 0.174 | 0.083 | 0.174 | **0.078** | 0.169 | 0.089 | 0.178 | 0.082 | 0.172 |

*Table 14.* Additional ablation studies: multivariate tokenization.

| Model | Metrics | Std Attn w/ Multi Tok MSE | MAE | Std Attn Original MSE | MAE | Std Attn +ARMA w/ Multi Tok MSE | MAE | Std Attn +ARMA Original MSE | MAE | Lin Attn w/ Multi Tok MSE | MAE | Lin Attn Original MSE | MAE | Lin Attn +ARMA w/ Multi Tok MSE | MAE | Lin Attn +ARMA Original MSE | MAE | GLin Attn w/ Multi Tok MSE | MAE | GLin Attn Original MSE | MAE | GLin Attn +ARMA w/ Multi Tok MSE | MAE | GLin Attn +ARMA Original MSE | MAE |
|---|---|---|---|---|---|---|---|---|---|---|---|---|---|---|---|---|---|---|---|---|---|---|---|---|---|
| ETTm1 96 | | 0.288 | 0.346 | **0.285** | **0.344** | 0.288 | 0.347 | 0.285 | 0.345 | 0.288 | 0.349 | 0.286 | 0.348 | 0.305 | 0.359 | 0.301 | 0.354 | 0.303 | 0.355 | 0.296 | 0.351 | 0.336 | 0.382 | 0.299 | 0.355 |
| ETTm1 48 | | 0.267 | 0.329 | 0.259 | 0.326 | 0.258 | 0.321 | **0.254** | **0.320** | 0.257 | 0.320 | 0.254 | 0.320 | 0.287 | 0.344 | 0.276 | 0.333 | 0.278 | 0.336 | 0.266 | 0.328 | 0.500 | 0.472 | 0.372 | 0.334 |
| ETTm1 24 | | 0.185 | **0.265** | **0.182** | 0.266 | 0.189 | 0.270 | 0.189 | 0.269 | 0.186 | 0.270 | 0.185 | 0.267 | 0.246 | 0.312 | 0.223 | 0.293 | 0.218 | 0.293 | 0.196 | 0.279 | 0.473 | 0.429 | 0.225 | 0.305 |
| ETTm1 12 | | 0.113 | **0.208** | 0.112 | 0.208 | 0.121 | 0.215 | 0.120 | 0.214 | 0.116 | 0.215 | 0.114 | 0.210 | 0.218 | 0.287 | 0.156 | 0.241 | 0.128 | 0.240 | 0.320 | 0.335 | 0.144 | 0.237 | | |
| ETTm2 96 | | 0.160 | 0.248 | 0.155 | 0.245 | 0.156 | 0.245 | **0.154** | **0.244** | 0.160 | 0.248 | 0.157 | 0.246 | 0.177 | 0.262 | 0.174 | 0.261 | 0.167 | 0.255 | 0.162 | 0.250 | 0.178 | 0.260 | 0.172 | 0.258 |
| ETTm2 48 | | 0.120 | 0.216 | 0.118 | 0.214 | 0.118 | 0.216 | **0.115** | **0.212** | 0.126 | 0.220 | 0.118 | 0.212 | 0.139 | 0.238 | 0.137 | 0.236 | 0.145 | 0.248 | 0.143 | 0.250 | 0.167 | 0.264 | 0.148 | 0.249 |
| ETTm2 24 | | 0.088 | 0.181 | 0.087 | 0.180 | 0.087 | 0.181 | **0.086** | **0.179** | 0.089 | 0.181 | 0.086 | 0.179 | 0.121 | 0.219 | 0.117 | 0.217 | 0.110 | 0.211 | 0.101 | 0.199 | 0.132 | 0.227 | 0.110 | 0.209 |
| ETTm2 12 | | 0.069 | 0.155 | 0.067 | **0.152** | 0.066 | 0.153 | **0.066** | 0.152 | 0.067 | 0.154 | 0.066 | 0.153 | 0.086 | 0.175 | 0.083 | 0.174 | 0.083 | 0.174 | 0.078 | 0.169 | 0.089 | 0.178 | 0.082 | 0.172 |

*Table 15.* Additional ablation studies: Comparison with TimesNet (Wu et al., 2022).

| Methods | TimesNet MSE | MAE | Std Attn MSE | MAE | Std Attn +ARMA MSE | MAE | Lin Attn MSE | MAE | Lin Attn +ARMA MSE | MAE | GLin Attn MSE | MAE | GLin Attn +ARMA MSE | MAE |
|---|---|---|---|---|---|---|---|---|---|---|---|---|---|---|
| ETTh1 (96) | 0.384 | 0.402 | **0.357** | **0.393** | 0.360 | 0.395 | 0.358 | 0.396 | 0.361 | 0.399 | 0.378 | 0.406 | 0.368 | 0.404 |
| ETTh1 (192) | 0.436 | 0.429 | 0.393 | 0.418 | **0.391** | **0.417** | 0.404 | 0.429 | 0.398 | 0.425 | 0.401 | 0.422 | 0.396 | 0.420 |
| ETTh1 (336) | 0.491 | 0.469 | 0.418 | 0.434 | **0.415** | **0.432** | 0.428 | 0.447 | 0.424 | 0.444 | 0.419 | 0.439 | 0.416 | 0.434 |
| ETTh1 (720) | 0.521 | 0.500 | 0.487 | 0.491 | 0.478 | 0.483 | 0.471 | 0.488 | 0.462 | 0.479 | 0.469 | 0.489 | **0.453** | **0.473** |
| ETTh2 (96) | 0.340 | 0.374 | **0.266** | **0.330** | 0.268 | 0.331 | 0.273 | 0.336 | 0.275 | 0.338 | 0.285 | 0.338 | 0.281 | 0.345 |
| ETTh2 (192) | 0.402 | 0.414 | 0.336 | 0.384 | 0.339 | 0.382 | 0.347 | 0.385 | **0.336** | **0.382** | 0.335 | 0.381 | **0.333** | **0.379** |
| ETTh2 (336) | 0.452 | 0.452 | 0.371 | 0.412 | 0.366 | **0.408** | 0.375 | 0.414 | 0.373 | 0.413 | **0.363** | 0.411 | 0.366 | 0.415 |
| ETTh2 (720) | 0.462 | 0.468 | 0.385 | 0.427 | 0.382 | 0.424 | 0.377 | 0.426 | **0.371** | **0.423** | 0.385 | 0.429 | 0.381 | 0.425 |
| ETTm1 (96) | 0.338 | 0.375 | 0.305 | 0.359 | 0.301 | 0.354 | 0.303 | 0.355 | **0.296** | **0.351** | 0.336 | 0.382 | 0.299 | 0.355 |
| ETTm1 (192) | 0.374 | 0.387 | 0.332 | 0.374 | 0.329 | **0.370** | 0.329 | 0.375 | **0.327** | 0.373 | 0.332 | 0.374 | 0.332 | 0.373 |
| ETTm1 (336) | 0.410 | 0.411 | 0.355 | **0.390** | 0.354 | 0.392 | 0.363 | 0.397 | 0.362 | 0.396 | 0.360 | 0.395 | 0.359 | 0.395 |
| ETTm1 (720) | 0.478 | 0.450 | 0.395 | 0.422 | 0.395 | 0.424 | 0.409 | 0.432 | 0.406 | 0.429 | 0.398 | 0.426 | **0.393** | **0.421** |
| ETTm2 (96) | 0.187 | 0.267 | 0.177 | 0.262 | 0.174 | 0.261 | 0.167 | 0.255 | **0.162** | **0.250** | 0.178 | 0.260 | 0.172 | 0.258 |
| ETTm2 (192) | 0.249 | 0.309 | 0.220 | 0.290 | 0.218 | 0.292 | 0.218 | 0.293 | **0.215** | **0.289** | 0.216 | 0.290 | 0.216 | 0.289 |
| ETTm2 (336) | 0.321 | 0.351 | **0.256** | 0.321 | 0.256 | **0.320** | 0.263 | 0.326 | 0.259 | 0.323 | 0.264 | 0.329 | 0.258 | 0.323 |
| ETTm2 (720) | 0.408 | 0.403 | 0.341 | 0.378 | 0.337 | **0.374** | 0.339 | 0.378 | 0.337 | 0.380 | 0.340 | 0.380 | **0.334** | 0.378 |
| Weather (96) | 0.172 | 0.220 | 0.144 | 0.195 | 0.142 | 0.193 | 0.142 | 0.194 | **0.139** | **0.191** | 0.161 | 0.210 | 0.142 | 0.194 |
| Weather (192) | 0.219 | 0.261 | 0.193 | 0.245 | 0.190 | **0.243** | 0.190 | 0.243 | **0.188** | 0.244 | 0.191 | 0.244 | 0.191 | 0.245 |
| Weather (336) | 0.280 | 0.306 | 0.245 | 0.289 | 0.242 | 0.287 | 0.239 | 0.283 | 0.237 | 0.283 | 0.238 | 0.284 | **0.236** | **0.281** |
| Weather (720) | 0.365 | 0.359 | 0.301 | 0.331 | 0.299 | 0.333 | 0.300 | 0.330 | **0.295** | **0.325** | 0.300 | 0.331 | 0.296 | 0.327 |

*Table 16.* Additional ablation studies: stability under different random seeds.

| Seed / Method | | Std Attn MSE | MAE | Std Attn +ARMA MSE | MAE | Lin Attn MSE | MAE | Lin Attn +ARMA MSE | MAE |
|---|---|---|---|---|---|---|---|---|---|
| Seed=2024 | ETTh1 (96) | 0.357 | 0.393 | 0.360 | 0.395 | 0.358 | 0.396 | 0.361 | 0.399 |
| | ETTh1 (192) | 0.393 | 0.418 | 0.391 | 0.417 | 0.404 | 0.429 | 0.398 | 0.425 |
| | ETTh1 (336) | 0.418 | 0.434 | 0.415 | 0.432 | 0.428 | 0.447 | 0.424 | 0.444 |
| | ETTh1 (720) | 0.487 | 0.491 | 0.478 | 0.483 | 0.471 | 0.488 | 0.462 | 0.479 |
| Seed=2025 | ETTh1 (96) | 0.358 | 0.394 | 0.361 | 0.396 | 0.358 | 0.398 | 0.361 | 0.399 |
| | ETTh1 (192) | 0.395 | 0.419 | 0.393 | 0.418 | 0.403 | 0.430 | 0.398 | 0.427 |
| | ETTh1 (336) | 0.417 | 0.432 | 0.416 | 0.431 | 0.427 | 0.449 | 0.425 | 0.446 |
| | ETTh1 (720) | 0.486 | 0.491 | 0.477 | 0.484 | 0.468 | 0.487 | 0.462 | 0.480 |
| Seed=2026 | ETTh1 (96) | 0.356 | 0.394 | 0.361 | 0.398 | 0.358 | 0.396 | 0.360 | 0.399 |
| | ETTh1 (192) | 0.393 | 0.419 | 0.391 | 0.418 | 0.404 | 0.429 | 0.396 | 0.426 |
| | ETTh1 (336) | 0.420 | 0.436 | 0.415 | 0.430 | 0.427 | 0.444 | 0.425 | 0.446 |
| | ETTh1 (720) | 0.489 | 0.489 | 0.477 | 0.482 | 0.472 | 0.487 | 0.460 | 0.480 |
| Seed=2027 | ETTh1 (96) | 0.355 | 0.389 | 0.360 | 0.395 | 0.357 | 0.396 | 0.361 | 0.401 |
| | ETTh1 (192) | 0.393 | 0.416 | 0.391 | 0.416 | 0.405 | 0.429 | 0.398 | 0.425 |
| | ETTh1 (336) | 0.419 | 0.431 | 0.415 | 0.431 | 0.426 | 0.446 | 0.424 | 0.445 |
| | ETTh1 (720) | 0.485 | 0.492 | 0.478 | 0.483 | 0.469 | 0.490 | 0.462 | 0.479 |
| Seed=2028 | ETTh1 (96) | 0.358 | 0.392 | 0.357 | 0.395 | 0.358 | 0.398 | 0.362 | 0.400 |
| | ETTh1 (192) | 0.393 | 0.418 | 0.388 | 0.415 | 0.403 | 0.430 | 0.397 | 0.424 |
| | ETTh1 (336) | 0.417 | 0.433 | 0.417 | 0.432 | 0.428 | 0.447 | 0.425 | 0.445 |
| | ETTh1 (720) | 0.487 | 0.491 | 0.478 | 0.482 | 0.469 | 0.486 | 0.464 | 0.478 |

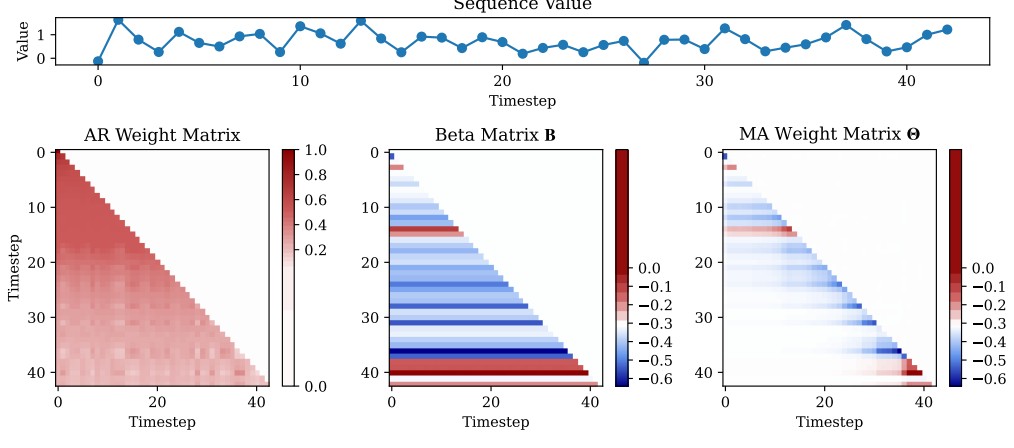

(a) Dataset: Weather, Channel: 1st, Layer: 1st

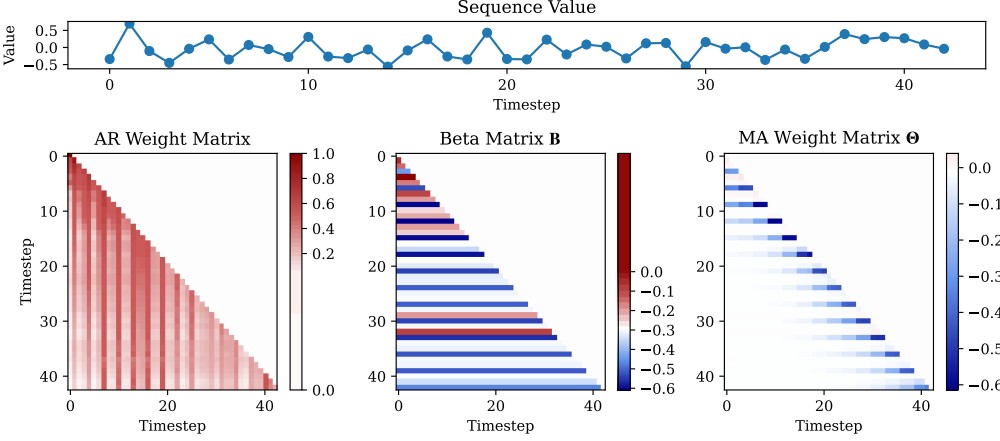

(b) Dataset: Weather, Channel: 1st, Layer: 2nd

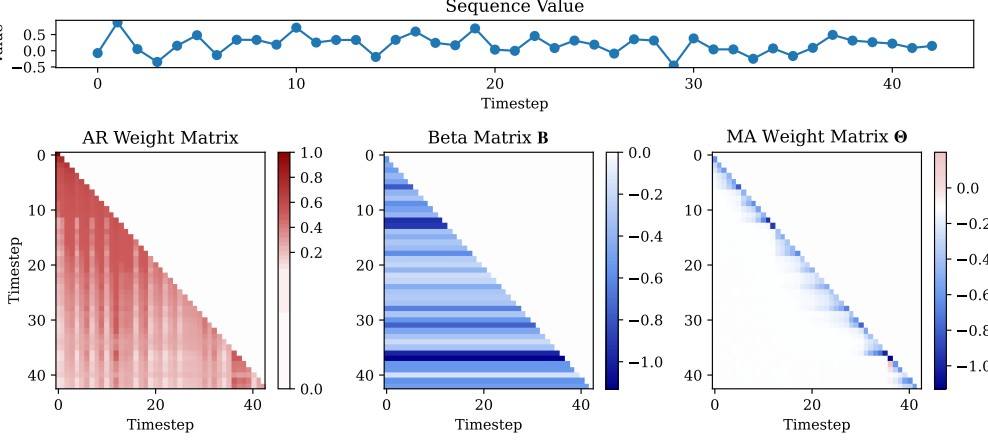

(c) Dataset: Weather, Channel: 1st, Layer: 3rd

*Figure 9.* Visualization of the WAVE attention weights of the first input channel for the first test set data point in the Weather dataset ($L_I = 4096$, $L_P = 96$).

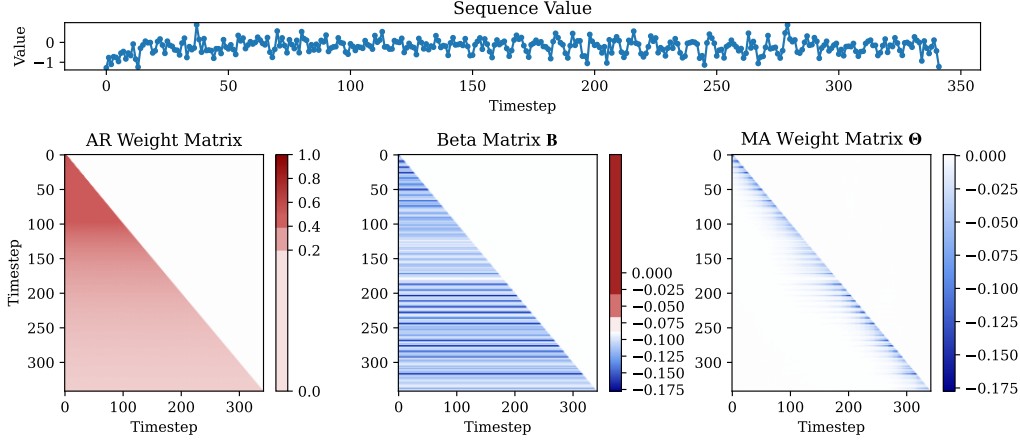

(a) Dataset: ETTm1, Channel: 1st, Layer: 1st

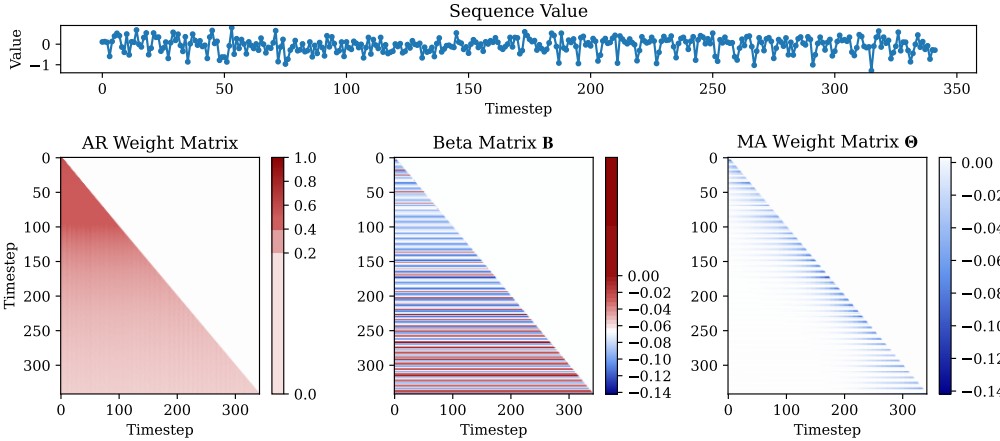

(b) Dataset: ETTm1, Channel: 1st, Layer: 2nd

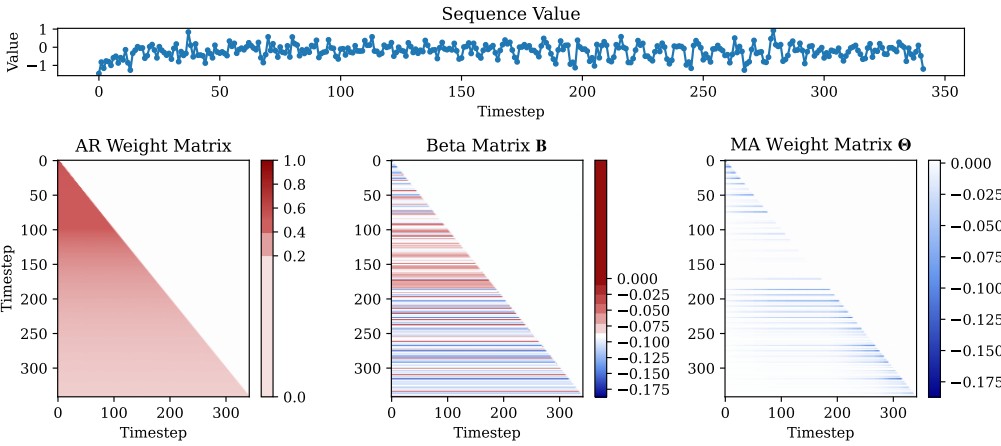

(c) Dataset: ETTm1, Channel: 1st, Layer: 3rd

*Figure 10.* Visualization of the WAVE attention weights of the first input channel for the first test set data point in the ETTm1 dataset ($L_I = 4096$, $L_P = 12$).

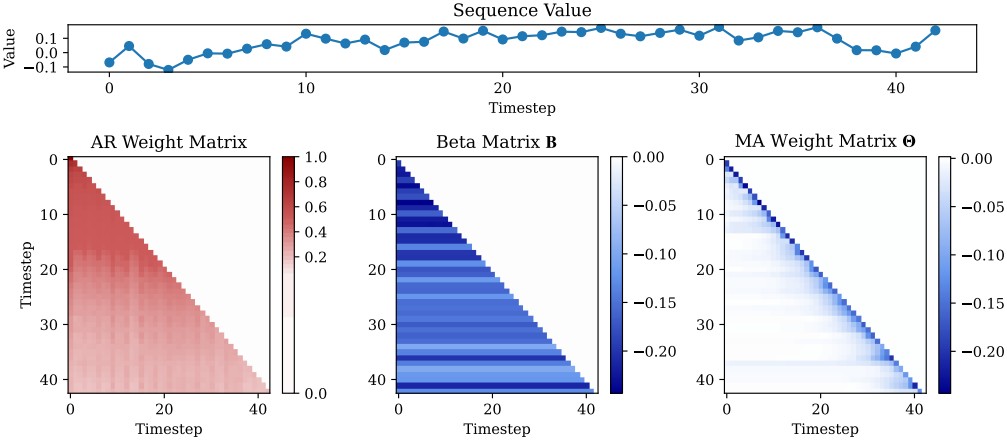

(a) Dataset: Weather, Channel: All (Averaged), Layer: 1st

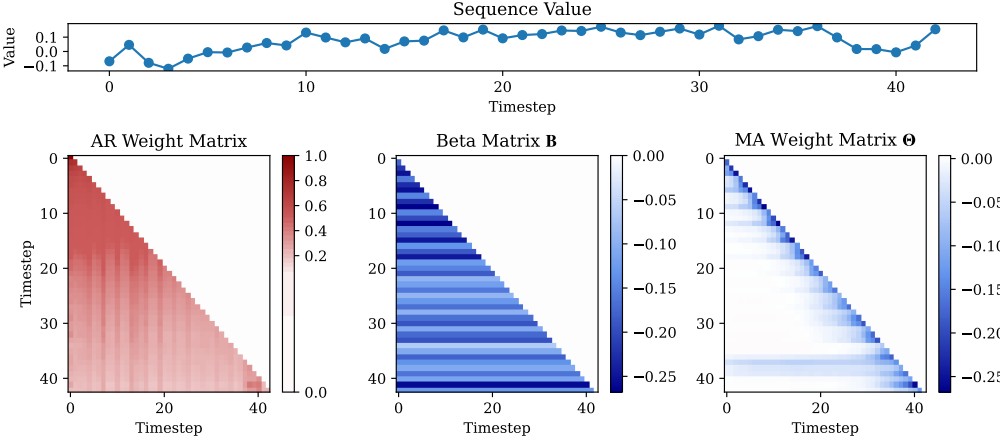

(b) Dataset: Weather, Channel: All (Averaged), Layer: 2nd

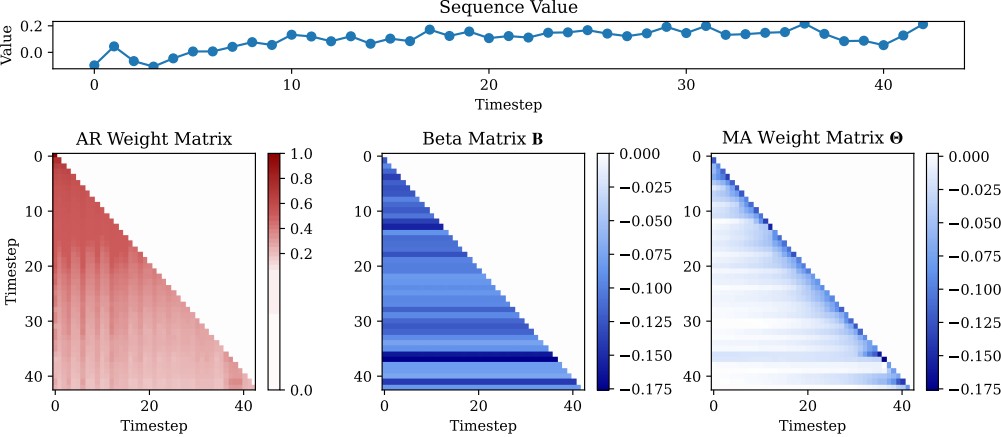

(c) Dataset: Weather, Channel: All (Averaged), Layer: 3rd

*Figure 11.* Visualization of the WAVE attention weights for the first test set data point in the Weather dataset ($L_I = 4096$, $L_P = 96$).

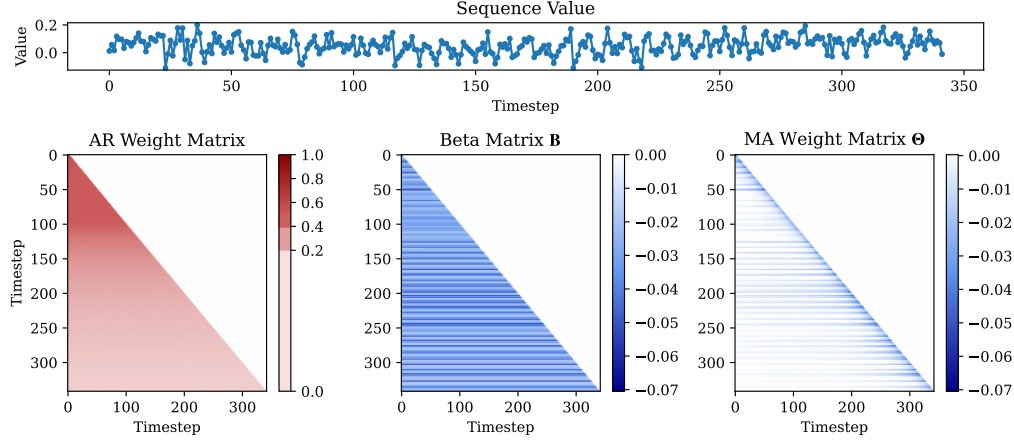

(a) Dataset: ETTm1, Channel: All (Averaged), Layer: 1st

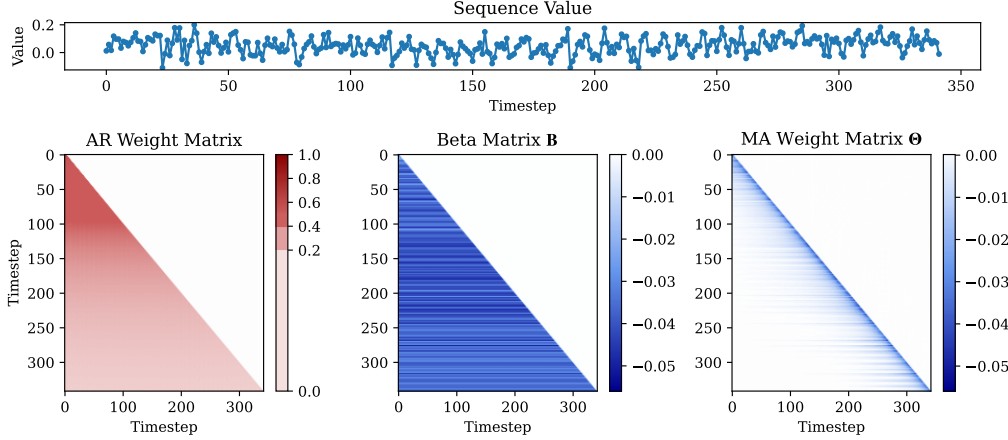

(b) Dataset: ETTm1, Channel: All (Averaged), Layer: 2nd

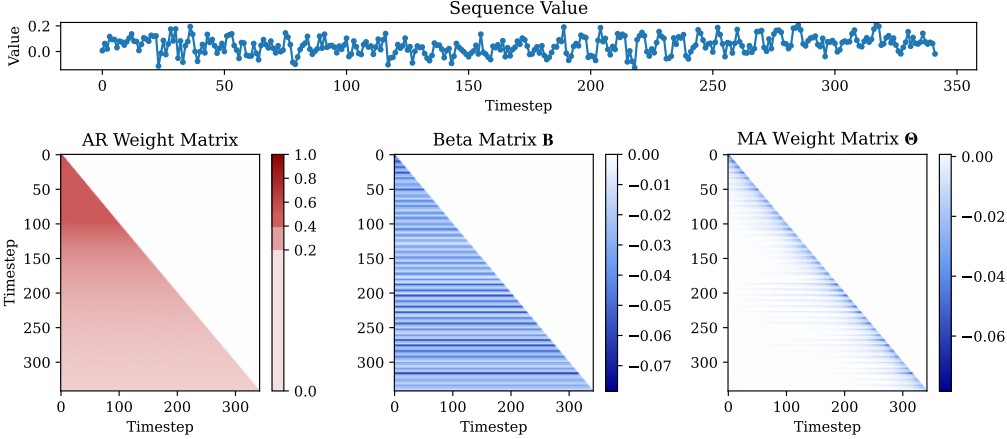

(c) Dataset: ETTm1, Channel: All (Averaged), Layer: 3rd

*Figure 12.* Visualization of the WAVE attention weights for the first test set data point in the ETTm1 dataset ($L_I = 4096$, $L_P = 12$).

