# OpenReview forum: "WAVE: Weighted Autoregressive Varying Gate for Time Series Forecasting"
_ICML.cc/2025/Conference — ICML 2025 poster_

### Official Review · Reviewer_Y1Dg · 2025-03-13

**Overall Recommendation:** 3

**Summary:**

In this paper the authors integrate the attention mechanism used for time series forecasting with the concepts of moving average in used in classic statistical ARMA models. In particular they device the indirect MA weights on top of patched tokenized time series with the emphasis on linear attention level complexities. They show via empirical study that the resulting WAVE mechanism helps improve the forecast accuracy significantly.

**Claims And Evidence:**

Yes.

**Essential References Not Discussed:**

Literature on pretrained decoder-only time series specific models have discussions on end to end evaluation usually as part of the empirical study, see e.g.,

[1] Liu, Yong, et al. "Timer: generative pre-trained transformers are large time series models." Proceedings of the 41st International Conference on Machine Learning. 2024.
[2] Das, Abhimanyu, et al. "A decoder-only foundation model for time-series forecasting." Forty-first International Conference on Machine Learning. 2024.

**Experimental Designs Or Analyses:**

Yes.

**Methods And Evaluation Criteria:**

Yes.

**Other Comments Or Suggestions:**

Math in the paper could be better organized. Consider provide more intuition, a central place for notation definitions, and more concise math writings.

**Other Strengths And Weaknesses:**

Strength:
- The WAVE mechanism, especially the patch level MA formulation, is novel among TSF methods.
- The design and the empirical study included in the paper are following common rigorous practices thus are relatively convincing.

Weakness:
- The intuition and the theory behind the proposal are relatively weak, especially compared to the original context when ARMA applies.
- The empirical study does not fully show the ARMA structure is working as intended.

**Questions For Authors:**

I am overall positive towards this submission, and plan to increase my recommendation once the following concerns / questions are addressed.

1. While ARMA has strong statistical interpretation, ARMA on the patch level in a stacked transformer becomes theoretical hard to understand. While there are examples (e.g., Fig 6) attempting to visualize the ARMA effect, what's the intuition behind WAVE when on a patched level, and how to claim it's working as intended instead of, e.g., merely introducing inductive biases in the model that work for the selected benchmark datasets?

2. Tokenization: while the authors claim an appropriate tokenization is required, the chosen tokenizer in the paper is a MLP(?) structure on top of patched time series which are common in PatchTST line of work. Any consideration of other tokenization strategies?

3. encoder-only vs decoder-only: the benefit of decoder-only mainly comes from its training efficiency and autoregressive decoding. The former is more crucial for pretrained models, so I wonder if we can see see any empirical study results on autoregressive decoding beyond a single step of the proposed method. Otherwise what's the justification of choosing decoder-only?

4. It's counterintuitive to see quadratic attention works worse than the linear attention most of the time, given the effective context length after patching is not long in the empirical study. What's the explanation, or if the quadratic models are way from optimally trained? Can we include more commonly used lookback / forecast horizon pairs in, e.g., the PatchTST paper?

**Relation To Broader Scientific Literature:**

N/A.

**Theoretical Claims:**

Yes.

---

> ### Author Rebuttal · Authors · 2025-04-01
>
> > Pretrained decoder-only models & end to end evaluation
>
> We'll clarify our introduction regarding design purpose and use this as evidence in Section 2.2 showing AR attention-based TSF models perform comparably to other structures.
>
> > Intuition behind WAVE on a patched level
> > Context of ARMA
> > Inductive biases issue
>
> Thanks for this insightful question. Due to MLPs and non-linearities, patch-level AR weights resist direct interpretation on original observations. Our parameterization provides **separate AR/MA attention weights** whose visualizations (Figures 6, 9-13) reveal **token-level cyclic patterns** captured by AR weights and **short-term effects** by MA weights.
>
> Each attention layer's value part can be viewed as a **reconstructed aggregation** of observations through a factor model. The Sequence Value section above each figure shows these layer-internal observations, essentially optimal input observations for ARMA structure learned at each layer.
>
> This optimization occurs in all multi-layer AR attention models. For non-first layers, each input sequence step differs from the original observation as each layer aggregates information from other steps. While this makes interpreting attention weights on original inputs difficult, interpreting them on the model's constructed optimal input sequence remains meaningful.
>
> In Figure 9, the three layers' AR/MA weights show distinct patterns. First-layer capture detailed **long-term/cyclic relationships** for different input lengths, while second/third layers capture stable cyclic patterns. MA weights show varying focus: first layer on distinctive **short-term dependencies**, second on shared decreasing short-term effects, and third on shorter corrective effects. This separation helps understand model mechanics better than pure AR models.
>
> Regarding inductive bias, we believe all recent networks works on inductive bias while we need to consider its generalization ability. Our ARMA attention has demonstrated good generalization across 12 widely-used TSF datasets at different scales, proving its effectiveness as a structural prior.
>
> > Other tokenization strategies
>
> We use this tokenization to maintain next-step autoregressive relationships while covering the entire forecasting horizon. This represents PatchTST-style tokenization with autoregressive loss, enabling direct benchmark comparison. Our focus is efficiently introducing **MA terms to AR attention**, and we experimented with inter-channel mixing (results at [link (Table 2)](https://anonymous.4open.science/r/WAVE-Rebuttal-B8D2)), showing multivariate relationships do improve performance.
>
> > Encoder-only vs Decoder-only
>
> Our decoder-only pure AR model extends PatchTST with **autoregressive loss**. We omitted multi-step autoregressive prediction comparisons since baselines used different training objectives. Experiments confirm AR/ARMA models outperform prior approaches in one-step forecasting. Though our model shows advantages with shorter contexts due to autoregressive training, this is a secondary benefit as longer contexts are preferred when available. Since decoder-only models operate within training windows, our evaluation uses **full contexts**, maintaining consistency with previous benchmarks.
>
> Additionally, decoder-only attention naturally performs multi-task training across varying context lengths, learning more adaptable representations that enhance generalization for TSF.
>
> > Quadratic attention works worse / are way from optimally trained? Why linear better?
>
> We agree. Research like DLinear, PatchTST, and iTransformer highlights serious overfitting in TSF Transformers. Softmax attention's non-linear capacity easily fits noise.
>
> Linear attention has a special **dual structure**:
>
> Autoregressive form of attention:
> $$
> \mathbf{o}\_t = \sum\_{i=1}^t \mathbf{w}\_{t,i} \mathbf{v}\_i \ , \mathbf{w}\_{t,i} = \mathbf{q}\_t \mathbf{k}\_i^\top \in \mathbb{R}^d
> $$
>
> Vector Autoregressive form of linear attention ($\mathbf{q}\_t \mathbf{k}\_i^\top = \mathbf{k}\_i \mathbf{q}\_t^\top$):
> $$
> \mathbf{o}\_t = \sum\_{i=1}^t \mathbf{k}\_i \mathbf{A}\_{t,i} \ , \mathbf{A}\_{t,i} = \mathbf{q}\_t^\top \mathbf{v}\_i \in \mathbb{R}^{d \times d}
> $$
>
> This dual linearity provides both **AR form and dynamic vector autoregression form**, providing **regularization**, and improving **linear relationship learning** in both channel- and token-wise dimensions.
>
> > Commonly used lookback / forecast horizon pairs
>
> For long-term forecasting, we used $L_I \in \{512, 1024, 2048, 4096\}$ with all $L_P$ values. We tested baseline setting $L_I=512$ and others. For stable token counts, we used fixed $L_I=1024$ for short-term prediction and specific combinations for long-term prediction. Table 3 demonstrates model's stability across different $L_I$ lengths. Our model works effectively with **very long lookbacks** ($L_I=4096$), showing excellent adaptability to extended contexts, which is a capability rare among existing models.

---

### Official Review · Reviewer_kz7F · 2025-03-14

**Overall Recommendation:** 3

**Summary:**

The author incorporates a moving average term into the autoregressive attention model for linear attention mechanisms, achieving state-of-the-art performance.

**Claims And Evidence:**

1. Effectiveness of the decoder-only autoregressive Transformer
   - In time series forecasting (TSF), the previously overlooked decoder-only autoregressive Transformer can achieve results comparable to top baseline methods with appropriate tokenization and training strategies.

2. Incorporating the full ARMA structure into autoregressive attention
   - Inspired by the ARMA model and recent advances in linear attention, this paper integrates the full ARMA structure into autoregressive attention mechanisms, improving long-range and local temporal modeling.

3. Proposing WAVE attention
   - WAVE attention combines autoregressive (AR) and moving-average (MA) components, enhancing adaptability to different attention mechanisms while improving their ability to model long-range dependencies and local patterns.
   - An indirect MA weight generation method introduces the MA component without increasing time complexity or parameter size in efficient attention models.

4. Effectiveness of indirect parameter generation for MA weights
   - The study shows that indirect parameter generation can implicitly produce MA weights suited for capturing local temporal effects.

5. Experimental validation of WAVE attention
   - WAVE attention consistently enhances autoregressive attention mechanisms and achieves state-of-the-art (SOTA) performance in time series forecasting tasks.

**Essential References Not Discussed:**

N/A

**Experimental Designs Or Analyses:**

N/A

**Methods And Evaluation Criteria:**

N/A

**Other Comments Or Suggestions:**

1. figure 3 is too small to understand

**Other Strengths And Weaknesses:**

s1. The WAVE attention mechanism captures short-term effects through the MA term, allowing the AR term to focus more on long-term and periodic patterns, thus balancing long-term and short-term dependencies.

s2. The WAVE attention mechanism proposed in the paper introduces the MA term while maintaining the same time complexity (O(N)) and parameter scale as the underlying efficient attention model.

s3. The experiments in the paper are comprehensive, covering both long-term and short-term time series forecasting, as well as ablation experiments.

w1. The method in this paper requires increasing the input length when the prediction length is extended. Even though, as the authors suggest, this method can be viewed as a patch for patchtst with added AR loss, patchtst itself does not require a longer input length when increasing the prediction length. Overall, while this design seems to alleviate the issue of error accumulation, it is not entirely natural. Existing decoder-only models like Autotimes do not need this design yet still remain state-of-the-art, demonstrating the true capability of decoder-only models: the ability to predict future lengths of any size.

w2. Sensitivity of parameters not analyzed.

**Questions For Authors:**

q1. Why is each row of matrix B in the visualization basically uniform?

**Relation To Broader Scientific Literature:**

N/A

**Theoretical Claims:**

contains no proofs.

---

> ### Author Rebuttal · Authors · 2025-04-01
>
> > The method in this paper requires increasing the input length when the prediction length is extended. Even though, as the authors suggest, this method can be viewed as a patch for patchtst with added AR loss, patchtst itself does not require a longer input length when increasing the prediction length. Overall, while this design seems to alleviate the issue of error accumulation, it is not entirely natural.
>
> Thank you for your suggestion. Our paper's main contribution is designing the ARMA attention mechanism and demonstrating that attention with ARMA structure significantly outperforms pure AR attention. Our tokenization method is merely a low-cost prerequisite problem introduction, aimed at incorporating AR training into the existing baseline experimental environment with minimal modifications. This allows us to isolate the improvement of adding the MA term to AR attention without introducing additional complex designs. We did this **only to control the introduction of irrelevant factors**.
>
> We use non-overlapping PatchTST-style tokenization + AR loss to maintain a **pure autoregressive relationship** between tokens. This ensures the MA term calculation better aligns with its original design. Otherwise, the MA term calculation would be affected by overlapping portions between the current and previous tokens. While this wouldn't significantly impact performance, using such results for comparison wouldn't sufficiently demonstrate that ARMA structure's performance gain comes from properly introducing the MA term. Similarly, if we mixed channel information in tokenization to model multivariate relationships, it would improve model performance to some extent (see [link (Table 2)](https://anonymous.4open.science/r/WAVE-Rebuttal-B8D2)). However, mixing channel information would place output tokens containing single series information and input tokens containing channel relationships in different spaces, breaking the AR training objective. These additional factors would weaken our experimental results showing ARMA attention's improvement over AR forecasting, requiring additional ablation studies. Therefore, to straightforwardly demonstrate our results, we used this constrained pure AR setting to prove our conclusions while controlling these extra structures.
>
> As shown in Table 3, this constrained setting provides additional insights: compared to current baselines with L_I≈512, it adapts to longer context lengths. This suggests some advantages for this **low-cost tokenization approach** when transferring AR attention to current experimental settings. However, these experiments were still designed to highlight ARMA attention's improvements over pure AR attention, which remains our primary objective.
>
> > Sensitivity of parameters not analyzed.
>
> Thank you for your suggestion. We modified the seed and ran our model five times, with results shown in [link (Table 4)](https://anonymous.4open.science/r/WAVE-Rebuttal-B8D2). We will add these additional experiments to the revision.
>
> > Figure 3 is too small to understand.
>
> Thank you for your suggestion. We will move the smaller font in Figure 3 to the caption to help readers understand it better.
>
> > Why is each row of matrix B in the visualization basically uniform?
>
> This is an intentional design choice, as we explained in detail in the footnote at the bottom right of page 5 and demonstrated in Figure 7. When matrix B lacks smoothness, the long-term components in the lower triangle dominate, preventing effective short-term modeling of MA terms.
>
> Increasing element variance in matrix $\mathbf{B}$ causes greater fluctuations in longer-term elements of $\mathbf{\Theta}$. This can be understood through the simplified mean form in Eq. 5: $\theta_{ij} = b(1 + b)^{i-j-1}$. The lower-left portion of $\mathbf{\Theta}$ with larger values of $i-j-1$ corresponds to longer-term components. Due to the larger exponents in this area, variance is more **significantly affected** by variance in $b$. This is described in the paper as:
>
> ---
> In the key activation, $ \alpha $ controls the variance of each row in the $ \mathbf{B} $ matrix, indirectly influencing the amount of long-term information (lower left) in the MA weights $ \mathbf{\Theta} $. Increasing $ \alpha $ would make the MA weights focus more on modeling long-term information. However, since we want the AR weights to handle the long-term component, we set $ \alpha $ to a relatively small value. This explains why the rows of the $ \mathbf{B} $ matrix **appear smooth** in the visualization. Refer to Fig. 7 for more details on $\alpha$, and see Fig. 8 for the effects of reversed positive $\phi_q$.

---

> > ### Comment · Reviewer_kz7F · 2025-04-02
> >
> > The author's response basically resolved my concerns, and I will keep my score.

---

### Official Review · Reviewer_ycDF · 2025-03-18

**Overall Recommendation:** 3

**Summary:**

The paper proposes WAVE, a novel attention mechanism integrating autoregressive (AR) and moving average (MA) components for time series forecasting (TSF). The key contributions include:

Demonstrating that a decoder-only autoregressive Transformer, with proper tokenization and preprocessing, achieves performance comparable to state-of-the-art (SOTA) baselines.

Introducing an ARMA structure into autoregressive attention via an indirect MA weight generation method, which maintains linear time complexity and parameter efficiency.

Validating WAVE's effectiveness across 12 TSF benchmarks, showing that it improves AR-based Transformers and achieves SOTA results. Experiments demonstrate WAVE's superiority over existing methods in both short- and long-term forecasting.

**Claims And Evidence:**

The paper's primary claims are largely supported by experiments. For instance, Figure 1 and Table 2 illustrate that AR Transformers perform competitively against baselines, while all WAVE variants outperform their AR counterparts. Table 7 further confirms that WAVE maintains computational efficiency in terms of FLOPs and parameter counts.

**Essential References Not Discussed:**

The paper proposes a decoder-only Transformer with ARMA-enhanced attention (WAVE) for time series forecasting, claiming state-of-the-art performance. However, it does not cite TimesNet: Temporal 2D-Variation Modeling for General Time Series Analysis (Wu et al., ICLR 2023), which also addresses multi-periodicity modeling and achieved SOTA results in forecasting tasks. A discussion of TimesNet's 2D temporal modeling approach and its differences from WAVE's ARMA mechanism would strengthen the technical context, and inclusion in experiments is recommended for rigorous benchmarking.

**Experimental Designs Or Analyses:**

The experimental section supports the main claims but has limitations. For example, the ablation studies could be expanded to validate the impact of AR Transformer components (e.g., tokenization strategies, weight-sharing mechanisms) on performance.

**Methods And Evaluation Criteria:**

The ARMA structure in WAVE is well-designed: the MA term models short-term impacts via error accumulation, and the indirect MA weight generation avoids explicit matrix inversion through linear attention mechanisms. These design choices are methodologically sound and align with the goal of balancing efficiency and performance.

**Other Comments Or Suggestions:**

Future work could explore the generalizability of WAVE to multivariate time series, as the current experiments focus primarily on univariate scenarios.

**Other Strengths And Weaknesses:**

Strengths:

The novel integration of the ARMA structure with efficient attention mechanisms provides a fresh perspective for time series forecasting (TSF), demonstrating the potential of AR Transformers in this domain.

The extensive experimental validation across multiple benchmarks strengthens the credibility of the results.

Weaknesses:

The absence of a dedicated related work section in the main text, which may hinder contextualizing the method within existing literature.

Limited ablation studies to isolate the contributions of key components (e.g., tokenization, AR/MA term interactions).
    The core novelty of introducing ARMA into TSF could benefit from further justification.
    Potential inconsistencies in Figure 1 (e.g., arrow directions in the schematic diagram).

**Questions For Authors:**

None.

**Relation To Broader Scientific Literature:**

This work situates itself within the broader scientific literature through three principal connections. First, it extends the foundational architecture of decoder-only Transformers (Vaswani et al., 2017) and their computationally efficient variants, establishing their applicability to time series forecasting via deliberate tokenization strategies.  Second, the study introduces a novel integration of classical autoregressive moving average (ARMA) principles (Box et al., 1974) into modern attention mechanisms, effectively bridging decades-old statistical forecasting techniques with contemporary deep learning paradigms. Third, the study resolves limitations of exponential decay in gated attention (e.g., MEGA) by explicitly separating MA terms, preserving long-range capabilities while capturing local patterns.

**Theoretical Claims:**

While the paper formalizes the ARMA structure and WAVE attention, it lacks rigorous theoretical proofs (e.g., asymptotic complexity analysis to substantiate WAVE’s linear time complexity claims). The validity of the method is primarily empirically justified.

---

> ### Author Rebuttal · Authors · 2025-04-01
>
> > It lacks rigorous theoretical proofs (e.g., asymptotic complexity analysis to substantiate WAVE’s linear time complexity claims)
>
> Thank you for your suggestion. We provide the time complexity analysis below:
>
> **Proposition** For a sequence of length $N$ and embedding dimension $d$, WAVE attention based on **efficient linear attention** maintains a time complexity of $O(Nd^2)$, which is linear in the sequence length.
>
> **Proof** We analyze the complexity of each component of WAVE attention separately:
>
> **AR Component Complexity** For linear attention variants, computing $\mathbf{o}^{\text{AR}}_t$ for all positions $t$ requires:
> - Computing query, key, value matrices: $O(Nd^2)$
> - For each position $t$, computing a running sum $\mathbf{S}\_t = \mathbf{S}\_{t-1} + (\mathbf{k}^{\text{AR}}\_t)^\top \mathbf{v}\_t$: $O(Nd^2)$
> - Computing $\mathbf{o}^{\text{AR}}\_t = \mathbf{q}\_t \mathbf{S}\_t$ for all $t$: $O(Nd^2)$
>
> Total AR component complexity: $O(Nd^2)$
>
> **MA Component Complexity** The indirect MA weight generation and computation involves:
> - Computing residuals $\mathbf{r}\_j = \mathbf{v}\_{j+1} - \mathbf{o}^{\text{AR}}\_j$: $O(Nd)$
> - Computing $\mathbf{q}^{\text{MA}}$ and $\mathbf{k}^{\text{MA}}$ matrices: $O(Nd^2)$
> - Applying activation functions $\phi^{\text{MA}}\_q$ and $\phi^{\text{MA}}\_k$: $O(Nd)$
> - For each position $t$, computing a running sum $\mathbf{T}\_t = \mathbf{T}\_{t-1} + \phi^{\text{MA}}\_k(\mathbf{k}^{\text{MA}}\_t)^\top \mathbf{r}\_t$: $O(Nd^2)$
> - Computing $\mathbf{o}^{\text{MA}}\_t = \phi^{\text{MA}}\_q(\mathbf{q}^{\text{MA}}\_{t-1}) \mathbf{T}\_{t-1}$ for all $t$: $O(Nd^2)$
>
> Total MA component complexity: $O(Nd^2)$
>
> **Final Output Complexity** Computing $\mathbf{o}_t = (\mathbf{o}^{\text{AR}}_t + \mathbf{o}^{\text{MA}}_t)\mathbf{W}_o$ for all $t$: $O(Nd^2)$
>
> Therefore, the total time complexity of WAVE attention based on linear attention is:
> $O(Nd^2 + Nd^2 + Nd^2) = O(Nd^2)$
>
> This confirms that WAVE attention maintains linear time complexity with respect to the sequence length $N$ when applied to **efficient linear attention** mechanisms.
>
> **Corollary** For WAVE attention based on **standard softmax attention**, the time complexity remains $O(N^2d + Nd^2)$, matching the underlying softmax attention mechanism.
>
>
> > 1. The ablation studies could be expanded to validate the impact of AR Transformer components (e.g., tokenization strategies, weight-sharing mechanisms)
> > 2. Limited ablation studies to isolate the contributions of key components (e.g., tokenization, AR/MA term interactions)
> > 3. Explore the generalizability of WAVE to multivariate time series
>
> Thank you for your suggestions. We provide experimental results in [link (Table 2)](https://anonymous.4open.science/r/WAVE-Rebuttal-B8D2) comparing channel-independent tokenization with channel-mixing tokenization, as well as results without autoregressive training loss. These findings support our claims. Additionally, without weight-sharing, parameter size would exceed pure AR attention, creating an unfair comparison with the same number of layers. We'll discuss this further in the revision appendix. Regarding AR/MA terms, every experiment includes comparisons between pure AR and ARMA, effectively serving as an ablation study. For interpretability, please refer to the following discussion:
>
> We illustrate using the hierarchical AR/MA weights visualization in Figure 9. The three layers exhibit different patterns. First layer AR weights capture detailed **long-term and cyclic relationships** at different input lengths, while second and third layers capture **common stable cyclic patterns**. First layer MA weights focus on distinct short-term dependencies across input lengths, second layer on shared decreasing short-term effects of fixed block length, and third layer on shorter-term correction effects. These AR/MA weights make the model's operation more interpretable compared to pure AR models.
>
> > 1. Does not cite TimesNet
> > 2. Discussion of TimesNet's 2D temporal modeling approach
> > 3. The absence of a dedicated related work section in the main text
> > 4. Potential inconsistencies in Figure 1
>
> Thank you for your suggestion. TimesNet uses 2D convolution on adjacent time series to simultaneously aggregate channel-wise and temporal-wise information, which differs conceptually from AR attention-based methods. We have provided performance comparisons with TimesNet in [link (Table 3)](https://anonymous.4open.science/r/WAVE-Rebuttal-B8D2).
>
> We will add a related work section in our revision to discuss recent TSF models beyond attention-based methods.
>
> We have verified that Figure 1 accurately reflects the ARMA structure calculation method. In our revision, we will move smaller fonts from Figure 1 to the caption for improved clarity.

---

### Official Review · Reviewer_rjH5 · 2025-03-19

**Overall Recommendation:** 3

**Summary:**

The Weighted Autoregressive Varying Gate (WAVE) attention is a new mechanism that augments Transformer attention with both an autoregressive component and a moving-average component. By combining ideas from statistical models with efficient Transformer architectures, WAVE expands the modeling capacity for time series forecasting while keeping the model lightweight and fast.

**Claims And Evidence:**

The paper claims that WAVE-attention-equipped Transformers outperform state-of-the-art TSF models while maintaining O(N) time complexity, and that its indirect MA weight generation allows efficient modeling of short-term effects without significantly increasing model size.

**Essential References Not Discussed:**

The paper adequately cites recent Transformer-based TSF models, but lacks statistical forecasting approaches (e.g., deep state-space models and mamba, etc).

**Experimental Designs Or Analyses:**

The one-step forecasting setup ensures fair comparisons between models. The long-term TSF experiments are insightful, demonstrating that WAVE Transformers scale effectively with increased lookback lengths, while other models suffer from performance degradation.

**Methods And Evaluation Criteria:**

Standard TSF benchmarks and baseline comparisons with recent Transformer architectures such as PatchTST, iTransformer, and DLinear.

**Other Comments Or Suggestions:**

Further interpretability analysis (e.g., visualization of learned MA weights over time) would strengthen the paper’s conclusions.
The notations are not clear.

**Other Strengths And Weaknesses:**

Pros

1. Good results. The proposed model consistently outperforms state-of-the-art TSF baselines across 12 datasets
2. The idea looks interesting and works well. WAVE attention successfully incorporates ARMA modeling principles into Transformer attention, improving both long-term dependency handling (AR) and short-term fluctuation modeling (MA) without increasing computational complexity.
3. The paper is well-written and well-organized.

Cons

1. The model is primarily tested on single-channel TSF tasks, and its performance on multivariate time series remains unexplored.
2. Its potential application to other sequential modeling tasks (e.g., NLP or reinforcement learning) is not discussed
3. Please elaborate more on the relation between mamba and WAVE.

**Questions For Authors:**

+ How does WAVE perform on multivariate time series forecasting, particularly when dealing with highly correlated input series?
+ Could WAVE be generalized to non-time-series applications, such as language modeling or reinforcement learning?

**Relation To Broader Scientific Literature:**

The study connects to statistical ARMA modeling, demonstrating how principles from classical time series analysis can enhance modern neural architectures.

**Theoretical Claims:**

The paper builds on ARMA modeling principles and argues that WAVE correctly extends attention mechanisms with an implicit MA term. The mathematical formulation for the indirect MA weight generation is clearly derived, showing that the method preserves the computational efficiency of linear attention while approximating a full ARMA structure.

---

> ### Author Rebuttal · Authors · 2025-04-01
>
> > The paper adequately cites recent Transformer-based TSF models, but lacks statistical forecasting approaches (e.g., deep state-space models and mamba, etc).
>
> Thank you for your suggestion. We will add a related works section in our revision to discuss statistical forecasting and recent SSM-based TSF models.
>
> > Performance on multivariate time series remains unexplored.
> > How does WAVE perform on multivariate time series forecasting.
>
> Thank you for your suggestion. Following your advice, we added multivariate channel-mixing to our tokenization: after obtaining a tensor of shape $(C, N, L_P)$ using AR tokenization (where $C$ is the number of series, $N$ is the patch token count, and $L_P$ is patch size = forecasting horizon), we applied a **linear projection to the $C$ dimension** to mix channel information at the same position. We report comparison results of pure AR vs. WAVE attention with multivariate AR tokenization in [link (Table 2)](https://anonymous.4open.science/r/WAVE-Rebuttal-B8D2). Results show that introducing multivariate effects enhances overall performance, and the improvements from ARMA structure remain effective. We will include these results in our revision.
>
> > Its potential application to other sequential modeling tasks is not discussed.
> > Could WAVE be generalized to non-time-series applications?
>
> Thank you for your suggestion. We've addressed this in the limitations section. Transformers for NLP are typically designed to learn complex, unstructured temporal relationships, unlike time series' common patterns (lag autocorrelation, cyclic patterns, seasonal effects, trends, local decay patterns). Introducing ARMA structure into attention provides **stable, interpretable inductive bias** for handling these effects, better modeling the time series generation process. We briefly explained TSF and NLP differences in section 2.5, which inspired our ARMA integration into AR attention. Further research is needed to verify if this TSF-specific inductive bias is effective for sequence tasks without fixed positional autocorrelation effects. We appreciate your understanding.
>
> > Please elaborate more on the relation between mamba and WAVE.
>
> Thank you for your suggestion. Mamba uses the following parameterization method:
>
> $$
> \mathbf{o}\_t = C \overline{B} \mathbf{x}\_t + C \overline{A}\_{t} \overline{B} \mathbf{x}\_{t-1} + C \overline{A}\_{t-2} \overline{A}\_{t-1} \overline{B} \mathbf{x}\_{t-2} + \cdots
> $$
>
> Its step-level formulation is:
>
> $$
> \mathbf{o}\_t = \sum\_{i=1}^t \prod\_{j=i+1}^{t} C \overline{A}\_{j}  \overline{B} \mathbf{x}\_{i}
> $$
>
> where $\mathbf{x}\_i$ is a $1 \times d$ input column vector. The term $\prod\_{j=i+1}^{t} C \overline{A}\_{j} \overline{B}$ can be viewed as a vector autoregressive weight matrix for step $i$. If we use a diagonalizable parameterization for $\overline{A}\_{j}$, similar to the method in linear recurrent unit (https://arxiv.org/abs/2303.06349), we get the parameterization $ C P (\prod_{j=i+1}^{t}\Lambda_j) P^{-1} \overline{B}$, where $P$ is an invertible matrix. Here, $CP$ can be seen as $W_o$ in attention, $P^{-1} \overline{B}$ as $W_v$, and each diagonal element in $\prod_{j=i+1}^{t}\Lambda_j$ as an attention score $w_{t,i}$ for each channel. The difference is that here $w_{t,i} = \prod_{j=i+1}^{t}\lambda_j$, rather than $w_{t,i} = f(x_t W_q W_k^\top x_i^\top)$ in attention. In the paper's channel-wise AR format: $o_t = w_{t,i} v_i, \ v_i = P^{-1} \overline{B} x_i$
>
> In summary: Diagonalizable Mamba or S4 (linear recurrent unit) can be viewed as **autoregression on each diagonal channel**, allowing direct incorporation of ARMA structure. However, Mamba or S4 (vanilla) can only be viewed as vector autoregression (SSM system), making it difficult to directly apply our method, which would require exploration in the VARMA domain.
>
> > Further interpretability analysis & notations are not clear.
>
> Thank you for your suggestion. We illustrate using the hierarchical AR/MA weights visualization in Figure 9. The AR/MA weights across the 3 layers exhibit different patterns. The first layer's AR weights capture detailed long-term and cyclic relationships corresponding to different input lengths, while the second and third layers' AR weights capture common stable cyclic patterns. The first layer's MA weights focus on **distinctive short-term dependencies** across various input lengths, the second layer focuses on **shared decreasing short-term effects** with fixed block lengths, and the third layer emphasizes **shorter-term correction effects**. These AR/MA weights help make the model's operating mechanism more understandable compared to pure AR models.
>
> We will add more visualization analyses in the revision and move smaller, less clear text from figures to captions for better readability.

---

### Official Review · Reviewer_nwFT · 2025-03-19

**Overall Recommendation:** 2

**Summary:**

This work introduces a decoder-only Transformer based model for time-series forecasting and introduces the WAVE attention mechanism. The WAVE attention mechanism leverages autoregressive and weighted moving averaging techniques. The authors show that coupling WAVE-based attention and a decoder-only structure outperforms other state-of-the-art models for short-term forecasting and produces comparable results for long-term forecasting. The authors also provide extensive ablation studies into WAVE-based attention mechanisms and their hyperparameters.

**Claims And Evidence:**

The claims made are supported by empirical evidence of the model's performance as well as plots showing the attention matrices. The plots of the attention matrices validate the authors' claims that the AR and MA attend to long-term and short-term phenomena, respectively. However, the authors do not provide evidence of how this affects the AR and MA predictions. I would like to see plots of oAR and oMA as well to validate the claims that oAR and oMA do in fact contain long-term cyclical and short-term dynamics, respectively.

**Essential References Not Discussed:**

The authors interweave discussion of previous works with their own methodological discussions. I find this approach helpful. However, I do think this paper lacks a sufficiently broad overview of previous works that touch on all aspects of this work.
- Currently, there appears to be one paragraph at the beginning that addresses previous works in general. This needs to be expanded to multiple paragraphs and include more topics discussed in this work. Below are two examples.
- Some mention of linear attention mechanism and the important works as well as their impact.
- Some mentions of other works that include weighted moving averages within the attention process. These references are severely lacking in these references, such as ETSformer. The authors can find a good overview of such works in the introduction and related works of "Powerformer: A Transformer with Weighted Causal Attention for Time-series Forecasting" and "TOTEM: TOkenized Time Series EMbeddings for General Time Series Analysis"

**Experimental Designs Or Analyses:**

- This work omits important experimental details making it unclear how exactly the experiments were performed. Please provide a section in the supplementary material providing such details. Below I list a few.
- How many experiments were run?
- What is the train/val/test split?
- Which hyperparameters (e.g. Li and Lp) are dataset dependent or dataset and prediction length dependent?

**Methods And Evaluation Criteria:**

This work evaluates multiple attention methods on standard benchmarks. However, it is unclear to me why the authors did not provide long-term prediction results for all the datasets used in the short-term experiments.
- Please report long-term experimental results for the same datasets as in the short-term experiments or provide sufficient reasons for not doing so.
- The authors do not show any time-series forecasts. It is very odd that there are no visualizations of the forecast results.

**Other Comments Or Suggestions:**

I consider each bullet point (outside of the listed strengths) as a concern that needs addressing. Below are some further writing and organization concerns. In general I think this is a compelling work ultimately worthy of publication after addressing the concerns and clarifications throughout this review.

- I cannot read the text on most figures
- Figure 3 seems very important but is never mentioned in the text. If it's not important enough to mention in the text it should not exist in the main body. The description of the attention mechanisms in the methods section would greatly benefit from pointing to the corresponding panel in Fig. 3. This figure should also be made much larger, it is too busy for how small it is.
- Table 1 needs to take up more space or be reconfigured, I can't separate the equations in Elin-attn
- Not all tables have bolded and underlined results, why? This would be very helpful
- Tables 4, 5, and 6 are mentioned in the paper in reverse order, their order needs to be reversed.

**Other Strengths And Weaknesses:**

**Strengths**
- The introduction of well estabilished theory into the attention mechanism through the AR and MA methods.
- Extensive experiments showing the benefits of the MA methods.
- Extensive experiments showing how ARMA improves upon multiple types of attention

**Weaknesses**
- Missing experiments and experimental details (described above).
- Poor use of figures and tables
- The claim of how the authors avoid error accumulation is very unclear even thought this seems important
- In general it seems like this paper needs to be read over by a more experienced writer, there are also random periods after some of the symbols which to me indicates that this manuscript was not thoroughly read over by other authors.

**Questions For Authors:**

1) Can you please describe how you avoid error accumulation?
2) Does this method predict the entire prediction sequence all at once, like PatchTST, or does it predict one time-step and then the sequence in an autoregressive fashion?

**Relation To Broader Scientific Literature:**

Time-series forecasting is a ubiquitous problem in scientific and industrial tasks. The authors introduce attention mechanisms based on well-established theory within these fields and expand upon them. Their work is broadly applicable to both science and industry.

**Theoretical Claims:**

- The authors attempt to avoid error accumulation by employing non-overlapping patching. However, when linking this patching method to their goal they briefly state "This ensures that each out-of-sample prediction token covers the entire forecasting length LP, thereby avoiding error accumulation." Basically the authors simply state that this works without providing insight or arguments that it should. Specifically, it's not clear how non-overlapping patches in the input mitigate error accumulation in the output, which I interpret as the point of this statement at this stage of the paper.

---

> ### Author Rebuttal · Authors · 2025-04-01
>
> > Describe how you avoid error accumulation.
>
> Thank you for this question. To clarify, this approach is not a contribution but a **prerequisite setting** demonstrating that **pure AR attention** can match previous models. Traditional one-step AR models accumulate significant errors during iterative prediction when $L_P>1$. By making each token contain $L_P$ steps, we **eliminate iterative prediction**, naturally avoiding these errors. This allows AR attention to integrate with existing baseline environments that also directly predict all $L_P$ steps. Essentially, this is similar to PatchTST tokenization with AR loss added between non-overlapping tokens, as illustrated in Figure 1(a).
>
> > Please report long-term experimental results for the same datasets
>
> Thank you for this valuable question. We prioritized short-term forecasting as our main experiment, with long-term as supplementary. This choice stems from **baseline limitations** rather than our model structure - previous baselines struggle to efficiently use longer lookback information needed to showcase ARMA attention's effectiveness in long-term forecasting.
>
> Our AR tokenization requires $L_P$ as patch length. For $L_P=720$, we need $L_I=4096$ to maintain sufficient tokens to demonstrate ARMA's benefits (Table 8). However, previous research typically used shorter input lengths.
>
> For fair comparison, we reran all baselines with $L_I \in \{512, 1024, 2048, 4096\}$ and selected the best results. Some baselines (Encformer, PatchTST) become **computationally infeasible** with high $L_I$, $L_P$, and input series count $C$. For large datasets like electricity ($C=321$) and traffic ($C=862$), using $L_I=4096$ makes training impossible even with batch size = 1. Thus, our long-term setting only included datasets up to Solar ($C=137$). Our code repository's baseline.sh verifies this limitation. We'll add a footnote to explain these constraints.
>
> > The authors do not show any time-series forecasts
>
> We'll add figures [link (PDFs)](https://anonymous.4open.science/r/WAVE-Rebuttal-B8D2) in the appendix. Time series visualizations typically show just one test datapoint from one series, providing limited information compared to comprehensive evaluation metrics. Papers in this field rarely include such figures in the main text, occasionally placing them in the appendix without using them to support claims.
>
> > Experimental details, train/val/test split, hyperparameters
>
> This information is detailed in Appendix B.2. Our setup matches DLinear, PatchTST, and iTransformer, using **identical train/val/test splits**. For short-term experiments, we fixed $L_I=1024$; for long-term, we used fixed combinations: $(1024,96)$, $(2048,192)$, $(2048,336)$, $(4096,720)$ for $(L_I,L_P)$. For baselines, we ran all combinations of $L_I \in \{512, 1024, 2048, 4096\}$ for each $L_P$ and selected the best results, ensuring **fair comparison** against optimally-performing baselines, similar to Table 3. Our complete code implementation supports these details.
>
> > Lacks a sufficiently overview of previous works
>
> Thank you for this suggestion. Our overview currently focuses on autoregressive attention mechanisms and attention for TSF. We'll expand the background content in the first section and add a related works section in the appendix discussing WMA and EMA applications in attention-based TSF and their relationship with ARMA.
>
> > 1. I cannot read the text on most figures
> > 2. Figure 3 never mentioned in the text
> > 3. Table 1 needs to take up more space
> > 4. bolded and underlined results
> > 5. Tables are in reverse order
>
> Thank you for these suggestions. We'll make these corrections:
>
> 1. We'll move small text from figures to captions.
> 2. We referenced Figure 3 in section 2.8 (line 262), though admittedly late. We'll reposition Figures 3 and 4 to address this ordering issue.
> 3. We'll add line breaks for Std and ELin names to provide more space for formulas.
> 4. We'll add light color formatting to all tables to highlight performance differences.
> 5. We'll reorder tables in ascending order.
>
> > Random periods after some of the symbols ... manuscript was not thoroughly read over by other authors
>
> We apologize for this confusion. What appeared as periods were actually `\cdot` **placeholders** in subscripts, particularly in sections 2.7 and 2.8. While normally displayed as $\phi(\cdot)$, in subscripts it appears as $\beta_{\cdot}$, resembling a period. In our revision, we'll use $\beta_{(\cdot)}$ in subscripts to clearly distinguish from periods.
>
> > Plots of oAR and MA.
>
> While decomposing AR/MA contributions to forecasting would be informative, our **stacked ARMA attention layers** make this challenging, as components mix with counterparts from previous layers. For understanding AR/MA behavior in each layer, our **visualizations of weights** are more appropriate. Please refer to Figure 9-12, which display AR and MA weights for each layer, providing detailed insights into how ARMA functions within layers.

---

### Decision · Program_Chairs · 2025-05-01

**Decision:**

Accept (poster)

**Comment:**

This paper proposes a new weighted attention mechanism specifically designed for time series forecasting, motivated by some classic ideas. The approach is novel and aims to better capture both long-term and short-term dependencies, supported by promising experimental results. Overall, the reviews are positive and detailed.

That said, the reviewers raised several valid concerns, including: (i) limited intuition and theoretical grounding behind the proposed approach; (ii) missing experimental details; and (iii) missing experiments on multivariate time series. The authors made good use of the rebuttal phase to address these and other issues, and one reviewer even increased their score in response.

Overall, I find this to be an interesting and promising paper, with sound methodology and results. My main concern is that the paper did not feel fully ready for publication, as evidenced by the number of concerns raised during the review process. Integrating all this feedback into the manuscript will require substantial revision. I strongly encourage the authors to incorporate all the discussed clarifications and improvements if given the opportunity to submit a camera-ready version. In particular, the presentation and discussion would benefit from further refinement. Moreover, strengthening the theoretical connection to classical time series models, such as ARIMA, would also add valuable context and depth to the work, and possibly help to motivate future work.

I recommend accepting the paper.